# Controls of outbursts of moraine-dammed lakes in the greater Himalayan region

Melanie Fischer[1], Oliver Korup[1,2], Georg Veh[1], Ariane Walz[1]

[1]Institute of Environmental Science and Geography, University of Potsdam, Potsdam, 14476, Germany
5   [2]Institute of Geosciences, University of Potsdam, Potsdam, 14476, Germany

*Correspondence to*: Melanie Fischer (melaniefischer@uni-potsdam.de)

**Abstract**

Glacial lakes in the Hindu-Kush Karakoram Himalaya Nyainqentanglha (HKKHN) have grown rapidly in number and area in past decades, and some dozens have drained in catastrophic glacial lake outburst floods (GLOFs). Estimating regional susceptibility of glacial lakes has largely relied on qualitative assessments by experts, thus motivating a more systematic and quantitative appraisal. Before the backdrop of current climate-change projections and the potential of elevation-dependent warming, an objective and regionally consistent assessment is urgently needed. We use an inventory of 3,390 moraine-dammed lakes and their documented outburst history in the past four decades to test whether elevation, lake area and its rate of change, glacier-mass balance, and monsoonality are useful inputs to a probabilistic classification model. We implement these candidate predictors in four Bayesian multi-level logistic regression models to estimate the posterior susceptibility to GLOFs. We find that mostly larger lakes have been more prone to GLOFs in the past four decades, regardless of elevation band in which they occurred. We also find that including the regional average glacier-mass balance improves the model classification. In contrast, changes in lake area and monsoonality play ambiguous roles. Our study provides first quantitative evidence that GLOF susceptibility in the HKKHN scales with lake area, though less so with its dynamics. Our probabilistic prognoses offer improvement compared to a random classification based on average GLOF frequency. Yet they also reveal some major uncertainties that have remained largely unquantified previously and that challenge the applicability of single models. Ensembles of multiple models could be a viable alternative for more accurately classifying the susceptibility of moraine-dammed lakes to GLOFs.

# 1 Introduction

Glacial lake outburst floods (GLOFs) involve the sudden release and downstream propagation of water and sediment from naturally impounded meltwater lakes (Costa and Schuster, 1987; Emmer, 2017). About one third of the 25,000 glacial lakes in the Hindu-Kush Karakoram Himalaya Nyainqentanglha (HKKHN) are dammed by moraines and some of these are potentially unstable (Maharjan et al., 2018). Such impounded meltwater can overtop or incise dams rapidly, with catastrophic consequences downstream (Costa and Schuster, 1987; Evans and Clague, 1994). High Mountain Asian countries are among the most affected by these abrupt floods, if considering both damage and fatalities (Carrivick and Tweed, 2016). For example, in June 2013, a GLOF from Chorabari Lake in the Indian state of Uttarakhand, caused >6,000 deaths in what is known as the "Kedarnath disaster" (Allen et al., 2016). The peak discharges of GLOFs can be orders of magnitude higher than those of seasonal floods. GLOFs can move large amounts of sediment, widen mountain channels, undermine hillslopes, and thus increase the hazard to local communities (Cenderelli and Wohl, 2003; Cook et al., 2018). Still, GLOFs in the HKKHN are rare and have occurred at an unchanged rate of about 1.3 per year in the past four decades (Veh et al., 2019). Ice avalanches and glacier calving are the most frequently reported triggers of GLOFs in the HKKHN. Most dated outbursts have occurred

between June and October) and might be linked to high lake levels fed by monsoonal precipitation and summer ablation of glaciers (Richardson and Reynolds, 2000). The Kedarnath GLOF is the only case attributed to a rain-on-snow event early in the monsoon season (Allen et al., 2016). This particularly destructive GLOF underlines the need for understanding better how and why meltwater lakes can be susceptible to sudden outburst triggered by rainstorms, especially given projected impacts of atmospheric warming on the high-mountain cryosphere.

Current scenarios entail that atmospheric warming may change the susceptibility of HKKHN glacial lakes to sudden outburst floods: IPCC's most recent projections attribute the decay of low-lying glaciers and permafrost to increases in lake number and area because of rising air temperatures, more frequent rain-on-snow events at higher elevations, and changes in precipitation seasonality (Hock et al., 2019). Air surface temperature in the HKKHN rose by about 0.1 °C per decade from 1901 to 2014 (Krishnan et al., 2019), likely having reduced snowfall, altered permafrost distribution, and accelerated glacier melt at lower elevations (Hock et al., 2019). Ice loss in the Himalayas has significantly increased in the past four decades, from $-0.22 \pm 0.13$ m w.e. y$^{-1}$ (meters of water equivalent per year) between 1975 and 2000 to $-0.43 \pm 0.14$ m w.e. y$^{-1}$ between 2000 and 2016 (Maurer et al., 2019). Parts of this meltwater have been trapped in glacial lakes that have expanded by approximately 14% between 1990 and 2015 (Nie et al., 2017). The notion of elevation-dependent warming (EDW) posits that increases in air temperature are most pronounced at higher elevations (Hock et al., 2019; Pepin et al., 2015). EDW has affected cold temperature metrics, including the number of frost days and minima of near-surface air temperature in the HKKHN in the past decades (Krishnan et al., 2019; Palazzi et al., 2017). Essentially, all scenarios of atmospheric warming concern aspects of elevation, glacial lake size and dynamics, and local climatic variability. Yet whether and how these aspects affect GLOF hazard still awaits more quantitative support.

Previous work on GLOF susceptibility and hazard in the region focused on identifying or classifying potentially unstable glacial lakes, including local case studies largely informed by fieldwork, dam-breach models (Koike and Takenaka, 2012; Somos-Valenzuela et al., 2012, 2014), and basin-wide assessments (Bolch et al., 2011; Mool et al., 2011; Rounce et al., 2016; Wang et al., 2011). GLOF hazard appraisals for the entire HKKHN, however, remain rare (Veh et al., 2020). Most basin-wide studies proposed qualitative to semi-quantitative decision schemes using selective lists of presumed GLOF predictors (Table 1; Rounce et al., 2016). Yet researchers have used subjective rules to choose these variables and associated thresholds, leading to diverging hazard estimates (Rounce et al., 2016). Expert knowledge has thus been essential in GLOF hazard appraisals, despite an increasing amount of freely available climatic, topographic, and glaciological data. Statistical models can help to estimate the occurrence probability of GLOFs, and thus reduce the inherent subjective bias (Emmer and Vilímek, 2013). For example, Wang et al. (2011) classified the outburst potential of moraine-dammed lakes on the southeastern Tibetan Plateau by applying a fuzzy consistent matrix method. They used as inputs the size of the parent glacier, the distance and slope between lake and glacier snout, and the mean steepness of the moraine dam and the glacier snout to come up with different nominal

hazard categories. This and many similar qualitative ranking schemes are accessible to a broader audience and policy makers,

70 but are difficult to compare and potentially oversimplify uncertainties.

One way to deal with these uncertainties in a more objective way involves a Bayesian approach. Here, we used this probabilistic reasoning with data-driven models. Specifically, we tested how well some of the more widely adopted predictors of GLOF susceptibility and hazard fare in a multi-level logistic regression that is informed more by data rather than by expert opinion. We checked how well this approach identifies glacial lakes in the HKKHN that had released GLOFs in the past four decades.

75 Our method estimates the probability of correctly detecting historic GLOFs from a set of predictors, which act as proxies subsuming various physical processes described as being relevant to GLOFs. Triggering mechanisms of these GLOFs are rarely reported, however Thus, we discuss what we can learn more about how these historic GLOFs were linked to readily available measures of topography, monsoonality, and glaciological changes. Our model results provide a posterior probability of outburst conditioned on detection, and this may be used as a relative metric of GLOF release from a given lake. Therefore,

80 our approach is an alternative to a formal assessment of moraine-dam stability, which is (geo)technically feasible only at selected sites and at scales much finer than our regional and decadal focus.

**Table 1: Frequently used predictors of GLOF susceptibility and hazard in the HKKHN.**

| Predictor groups | GLOF susceptibility and hazard predictors | Tested in this study | Reference |
|---|---|---|---|
| Lake characteristics and dynamics | Glacial lake elevation | ✓ | Mergili and Schneider, 2011 |
| | Catchment area | ✓ | Allen et al., 2019; GAPHAZ, 2017 |
| | Glacial lake area | ✓ | Aggarwal et al., 2016; Allen et al., 2019; Bolch et al., 2011; GAPHAZ, 2017; Ives et al., 2010; Khadka et al., 2021; Mergili and Schneider, 2011; Prakash and Nagarajan, 2017; Wang et al., 2012; Worni et al., 2013 |
| | Lake-area change (growth and shrinkage, absolute change) | ✓ | Aggarwal et al., 2016; Bolch et al., 2011; Ives et al., 2010; Khadka et al., 2021; Mergili and Schneider, 2011; Prakash and Nagarajan, 2017; Rounce et al., 2016; Wang et al., 2012 |
| Potential downstream impact | Lake volume | - | Aggarwal et al., 2016; Bolch et al., 2011; GAPHAZ, 2017; Kougkoulos et al., 2018; Mergili and Schneider, 2011 |
| Dam stability | Moraine-wall steepness | - | Allen et al., 2019; Bolch et al., 2011; Dubey and Goyal, 2020; GAPHAZ, 2017; Ives et al., 2010; Khadka et al., 2021; Prakash and Nagarajan, 2017; Rounce et al., 2016; Wang et al., 2011; Worni et al., 2013 |
| | Width-to-height ratio | - | Aggarwal et al., 2016; Bolch et al., 2011; GAPHAZ, 2017; Ives et al., 2010; Prakash and Nagarajan, 2017; Worni et al., 2013 |
| | Lake freeboard | - | Bolch et al., 2011; GAPHAZ, 2017; Kougkoulos et al., 2018; Mergili and Schneider, 2011; Prakash and Nagarajan, 2017; Worni et al., 2013 |
| | Existence of a buried ice core | - | Bolch et al., 2011; Dubey and Goyal, 2020; GAPHAZ, 2017; Ives et al., 2010; Rounce et al., 2016 |
| | Dam type | ✓ | GAPHAZ, 2017; Kougkoulos et al., 2018; Mergili and Schneider, 2011; Wang et al., 2011; Worni et al., 2013 |

| | | | |
|---|---|---|---|
| | Moraine lithology | | GAPHAZ, 2017 |
| Potential triggering mechanisms (geomorphic) | Seismic activity | - | GAPHAZ, 2017; Ives et al., 2010; Kougkoulos et al., 2018; Mergili and Schneider, 2011; Prakash and Nagarajan, 2017 |
| | Distance from parent glacier snout | - | Aggarwal et al., 2016; Ives et al., 2010; Khadka et al., 2021; Kougkoulos et al., 2018; Prakash and Nagarajan, 2017; Wang et al., 2011, 2012; Worni et al., 2013 |
| | Steepness parent glacier snout | - | Bolch et al., 2011; Ives et al., 2010; Kougkoulos et al., 2018; Prakash and Nagarajan, 2017; Wang et al., 2011 |
| | Parent glacier calving potential (width, crevasse density) | - | GAPHAZ, 2017; Ives et al., 2010; Mergili and Schneider, 2011 |
| | Regional or parent glacier-mass balance | ✓ | Bolch et al., 2011; Ives et al., 2010 |
| | Mass movements (traces, trajectories, probabilities) | - | Allen et al., 2019; Bolch et al., 2011; Dubey and Goyal, 2020; GAPHAZ, 2017; Ives et al., 2010; Khadka et al., 2021; Mergili and Schneider, 2011; Prakash and Nagarajan, 2017; Rounce et al., 2016; Worni et al., 2013 |
| | Permafrost conditions | - | GAPHAZ, 2017 |
| | Upstream lake (with GLOF potential) | - | Dubey and Goyal, 2020; GAPHAZ, 2017; Khadka et al., 2021 |
| Potential triggering events (climatic) | Annual mean temperature | - | GAPHAZ, 2017; Liu et al., 2014; Wang et al., 2008 |
| | Temperature seasonality | - | Ives et al., 2010; Kougkoulos et al., 2018 |
| | Temperature extremes (intensity, frequency) | - | GAPHAZ, 2017 |
| | Annual precipitation | - | Wang et al., 2008, 2012 |
| | Precipitation seasonality | - | Ives et al., 2010; Kougkoulos et al., 2018 |
| | Precipitation extremes (intensity, frequency) | - | GAPHAZ, 2017; Prakash and Nagarajan, 2017 |
| | Summer precipitation or proxy of monsoonality | ✓ | Wang et al., 2008, 2012 |

# 2 Study area, data, and methods

## 2.1 Study area and data

We studied glacial lakes of the Hindu-Kush Karakoram Himalaya Nyainqentanglha (HKKHN) region that we defined here as the Asian mountain ranges between 16º to 39ºN and 61º to 105ºE, i.e. from Afghanistan to Myanmar (Fig. 1; Bajracharya and Shrestha, 2011). Following the outlines of glacier regions in High Mountain Asia used in the Randolph Glacier Inventory version 6.0 (RGI 6.0, Pfeffer et al., 2014) with slight modifications, we subdivided our study area into the following seven

mountain ranges: the Hindu Kush, the Karakoram, the Western Himalaya, the Central Himalaya, the Eastern Himalaya, the Nyainqentanglha, and the Hengduan Shan. Meltwater from the HKKHN's extensive snow and ice cover, often referred to as "Third Pole", feeds ten major river systems to provide water for some 1.3 billion people (Molden et al., 2014). There, glaciers have had an overall negative mass balance historically and lost $150 \pm 110$ kg m$^{-2}$ yr$^{-1}$ on average from 2006 to 2015, though with balanced trends in the Karakoram (Bolch et al., 2019; Hock et al., 2019). Since the 1970s, some Karakoram glaciers also accelerated in flow, whereas glaciers stalled elsewhere in the HKKHN (Dehecq et al., 2019). In the RCP8.5 scenario the HKKHN glaciers could lose $64 \pm 5\%$ of their total mass until 2100 compared to estimated glacier volumes for the interval 1995 to 2015 (Kraaijenbrink et al., 2017). How much of this melting of glaciers is due to EDW remains debated (Palazzi et al., 2017; Rangwala and Miller, 2012; Tudoroiu et al., 2016). Snowfall at lower elevations is also likely to decrease (Hock et al., 2019; Terzago et al., 2014), judging from snowfall and glacier-mass balances of past decades (Kapnick et al., 2014; King et al., 2019). Monsoon precipitation is likely to become more episodic and intensive (Palazzi et al., 2013).

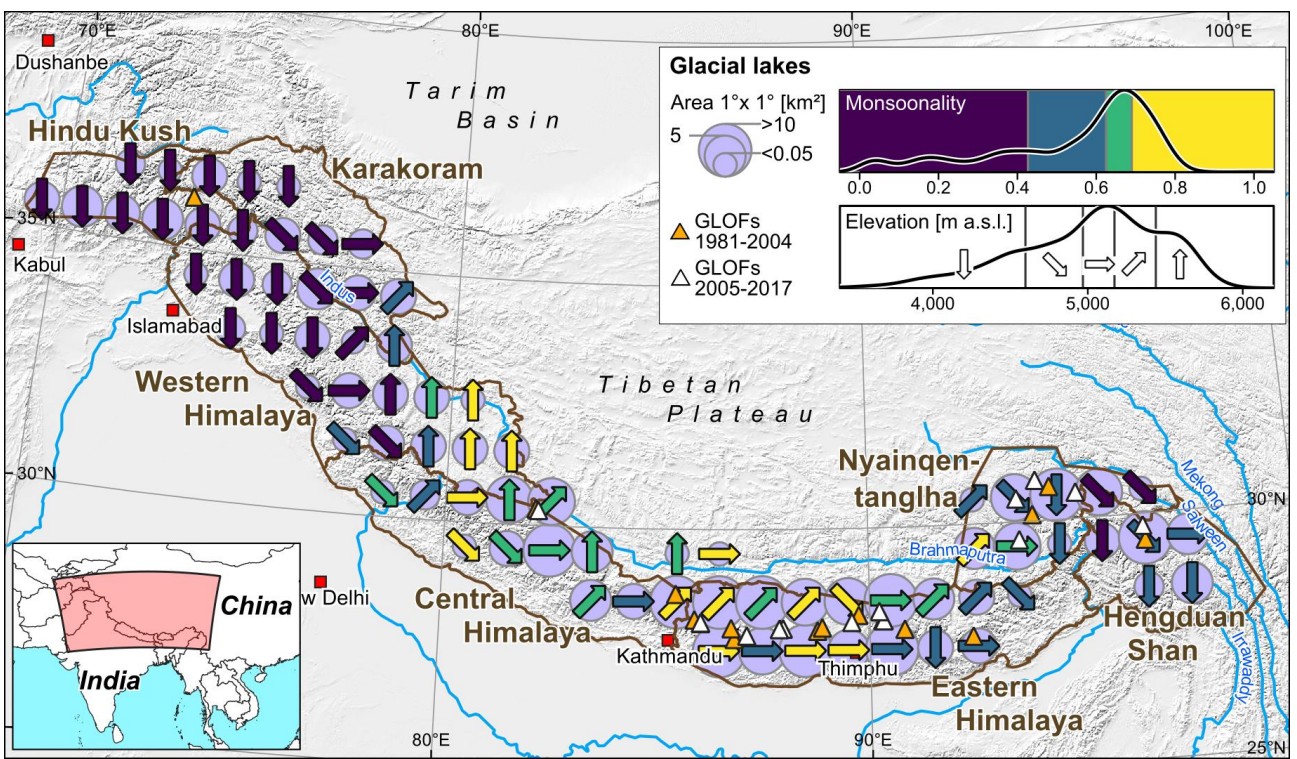

**Figure 1: Overview map of the HKKHN showing the distribution of moraine-dammed lakes in 1° x 1° bins (blue bubbles scaled by area), their elevation (expressed as quantiles coded by arrows; see inset for elevation distribution); and average monsoonality (colour coded; see inset for monsoonality distribution), defined here as the fraction of total annual precipitation falling in the summer months. Orange and white triangles indicate reported moraine-dam failures before and after 2005, respectively (Veh et al., 2019). Background hillshade is from the GTOPO30 global 30'' elevation dataset (https://doi.org/10.5066/F7DF6PQS).**

Guided by these projections, we selected several widely used glacial lake susceptibility predictors (Table 2).

We used **lake elevation** $z$ (m a.s.l.) as a proxy for the standard lapse rate of tropospheric air temperature (Rolland, 2003; Yang and Smith, 1985). This elevation-dependent thermal gradient is also a major control on the distribution of alpine permafrost (Etzelmüller and Frauenfelder, 2009) and precipitation. Mean annual rainfall along the Himalayan front can exceed 4,000 mm at elevations some 4,000 m high, where ~25% of all glacial lakes occur (Fig. 1; Bookhagen and Burbank, 2010). Lake elevation should also represent to first order topographic effects of EDW. For example, the stability of low-lying moraine dams may be

compromised by the loss of permafrost and commensurate increases in permeability in the moraine barrier and adjacent valley slopes (Haeberli et al., 2017).

Glacial **lake area** $A$ (m²) and its rate of **change** $\Delta A$ (net change) and $A^*$ (relative change, %) are other common predictors of susceptibility and hazard in GLOF studies (Allen et al., 2019; Bolch et al., 2011; Prakash and Nagarajan, 2017; see Table 1 for full list of references) that we considered here. Due to a general lack in available bathymetric data on a regional scale, a

number of studies used the frequently observed phenomenon that lake area scales with lake volume and depth (Huggel et al., 2002; Iribarren Anacona et al., 2014). Growing lake depths increase the hydrostatic pressure acting on moraine dams, thus raising the potential of failure (Iribarren Anacona et al., 2014; Rounce et al., 2016). Since 1990, lake areas have grown largest in the Central Himalayas (+23%), and lowest in the northwestern Himalayas (+5.0%) (Nie et al., 2017), and many studies have emphasised the role of growing lakes on GLOF susceptibility (e.g. GAPHAZ, 2017; Prakash and Nagarajan, 2017; Rounce et

al., 2016). Many previous GLOF assessment schemes included lake area or lake area growth as a proxy for the volume of water that could be potentially released by an outburst and, thus, the resulting downstream hazard (e.g. Allen et al., 2019; Bolch et al., 2011). However, a number of studies also stress that lake area and its growth define the exposure to external and internal triggers of moraine dam breach: larger and growing lakes offer more area for impacts from mass flows such as avalanches, rockfalls, and landslides originating from adjacent valley slopes (GAPHAZ, 2017; Haeberli et al., 2017; Prakash

and Nagarajan, 2017; Rounce et al., 2016). Some authors also link growing lake areas to an increase in hydrostatic pressure acting on its moraine dam, thus, making the letter more susceptible to sudden failure (Iribarren Anacona et al., 2014; Mergili and Schneider, 2011).

We also tested the impact of upstream **catchment area** $C$ (m²) on GLOF susceptibility. A larger upstream catchment area has been associated with an increased susceptibility to GLOFs as runoff from intense precipitation as well as glacier and snow

melt can lead to sudden increases in lake volume (Allen et al., 2019; GAPHAZ, 2017). We find that catchment area $C$ correlates with lake area A (Pearson's $\rho = 0.45$) and we, thus, preferred $C$ over $A$ in two of our models, as $C$ is invariant at the timescale of our study and we use these two models to explicitly test whether runoff by glacier melt or monsoonal precipitation had an effect on GLOFs in our study area.

Similarly to changes in lake area, glacier dynamics are frequently mentioned, though rarely incorporated quantitatively in

susceptibility appraisals (Bolch et al., 2011; Ives et al., 2010). This motivated us to consider the average changes in **regional glacier-mass balances** between 2000 and 2016 $\Delta m$ (m water equivalent yr$^{-1}$) from Brun et al. (2017). These readily available data on regional glacier-mass balances are proxies for other, less accessible, physical controls on GLOF susceptibility such as glacial meltwater input, either directly from the parent glacier or from glaciers upstream, as well as permafrost decay in slopes fringing the lake (see Table 2 for full list).

Meteorological drivers entered previous qualitative GLOF hazard appraisals mostly as (the probability of) extreme monsoonal precipitation events: the Kedarnath GLOF disaster, for example, was triggered by intense surface runoff (Huggel et al., 2004; Prakash and Nagarajan, 2017). Heavy rainfall may also trigger landslides or debris flows from adjacent hillslopes followed by displacement waves that overtop moraine dams (Huggel et al., 2004; Prakash and Nagarajan, 2017). Elevated lake levels during the monsoon season also raise the hydrostatic pressure acting onto moraine dams (Richardson and Reynolds, 2000).

Furthermore, different precipitation regimes and climatic preconditions may also influence moraine dam-failure mechanics (Wang et al., 2012). Intense precipitation occurs in our study region largely during the summer monsoon, so that we derived a synoptic measure of **monsoonality** $M$ (%). We define monsoonality $M$ in terms of the annual proportion of summer, i.e. the warmest quarter's, precipitation, which is highest in the southeast HKKHN, where it is linked to monsoonal low-pressure systems (Krishnan et al., 2019).


**Table 2: Details on tested predictors and our reasoning for selection based on their commonly reported physical links to GLOF susceptibility**

| GLOF susceptibility predictor | Symbol | | Unit | Data source | Selection reasoning |
|---|---|---|---|---|---|
| Glacial lake elevation | $z$ | | m a.s.l. | SRTM DEM | - strong link between elevation and temperature in high altitudes (standard lapse rate of tropospheric air temperature) <br> → elevation dependence of permafrost and precipitation patterns |
| Catchment area | $C$ | | m² | SRTM DEM | - potential for surface runoff into lake from precipitation and snow melt |
| Glacial lake area | $A$ | | m² | SRTM DEM | - proxy for lake volume and depth and, thus, hydrostatic pressure acting onto moraine dam |
| Lake-area change | $\Delta A$ | net change | - | Wang et al., 2020 | - increasing lake area commonly reported as scaling with increasing lake depth <br> → potentially increased hydrostatic pressure acting on the moraine dam <br> - increased proximity to steep valley slopes <br> → increased potential of mass movements entering the lake |
| | $A^*$ | relative change (between) | % | | |
| | | $A^{*a}$ (1990-2005) <br> $A^{*b}$ (2005-2018) <br> $A^{*c}$ (1990-2018) | | | |
| Glacier-mass balance | $r$ | glacier-mass balance region | - | Brun et al., 2017 | |

| | | | | | |
|---|---|---|---|---|---|
| | $\Delta m_r$ | average glacier-mass balance | m w.e. (water equivalent) yr$^{-1}$ | | - proxy for direct or surface runoff glacier meltwater input, calving potential of parent glacier front, and permafrost distribution in lake surroundings<br>- link between regional glacier-mass balance and synoptic regime (winter westerlies versus monsoon dominated) |
| Monsoonality (annual proportion of summer precipitation) | $M$ | | % (mm) | CHELSA (Karger et al., 2017) | - high intensity precipitation events during monsoon season might lead to increased surface runoff into glacial lakes (cloudburst event)<br>- seasonal increases in lake levels and, hence, lake depths increase hydrostatic pressure acting onto moraine dam<br>- link between regional glacier-mass balance and synoptic regime (winter westerlies versus monsoon dominated) |


We extracted information on these characteristics for glacial lakes recorded in two inventories. First, we used the ICIMOD database of 25,614 lakes manually mapped from Landsat imagery acquired in 2005 (± two years) (Maharjan et al. 2018), from which we extracted 7,284 lakes dammed mostly by lateral and end moraines. Second, we identified from an independent regional GLOF inventory (Veh et al. 2019) 31 lakes that had at least one outburst between 1981 and 2017 and that are listed in the ICIMOD inventory. The triggering mechanism of these studied GLOFs is reported in only seven cases, four of which are attributed to ice avalanches entering the lake (e.g. Tam Pokhari, Nepal or Kongyangmi La Tsho, India; Ives et al., 2010; Nie et al., 2018). Other triggers of the GLOFs studied here include piping (Yindapu Co, China; Nie et al., 2018) and the collapse of an ice-cored moraine (Luggye Tsho, Bhutan; Fujita et al., 2008). We focused on lakes >10,000 m² to ensure comparability between the two inventories, thus acquiring a final sample size of 3,390 lakes. Given the sparse network of weather stations in the HKKHN, we computed the monsoonality averaged for each lake from the 1-km resolution CHELSA bioclimatic variables (Karger et al., 2017). These variables are correlated with elevation because of the same underlying interpolation technique so that we limited our models to those with poorly correlated predictors. This meant omitting other predictors such as mean annual temperature, annual precipitation totals and annual temperature and precipitation variability. We extracted topographic data from the void-free 30-m resolution SRTM (Shuttle Radar Topographic Mission of 2000) DEM, and use approximate lake-area changes for two intervals (1990 to 2005 and 2005 to 2018) by Wang et al. (2020). We discarded newer, higher resolved DEMs to minimise data gaps and artefacts. Overall, we considered six topographic, synoptic, and glaciological predictors (Fig. 2, Table 2).

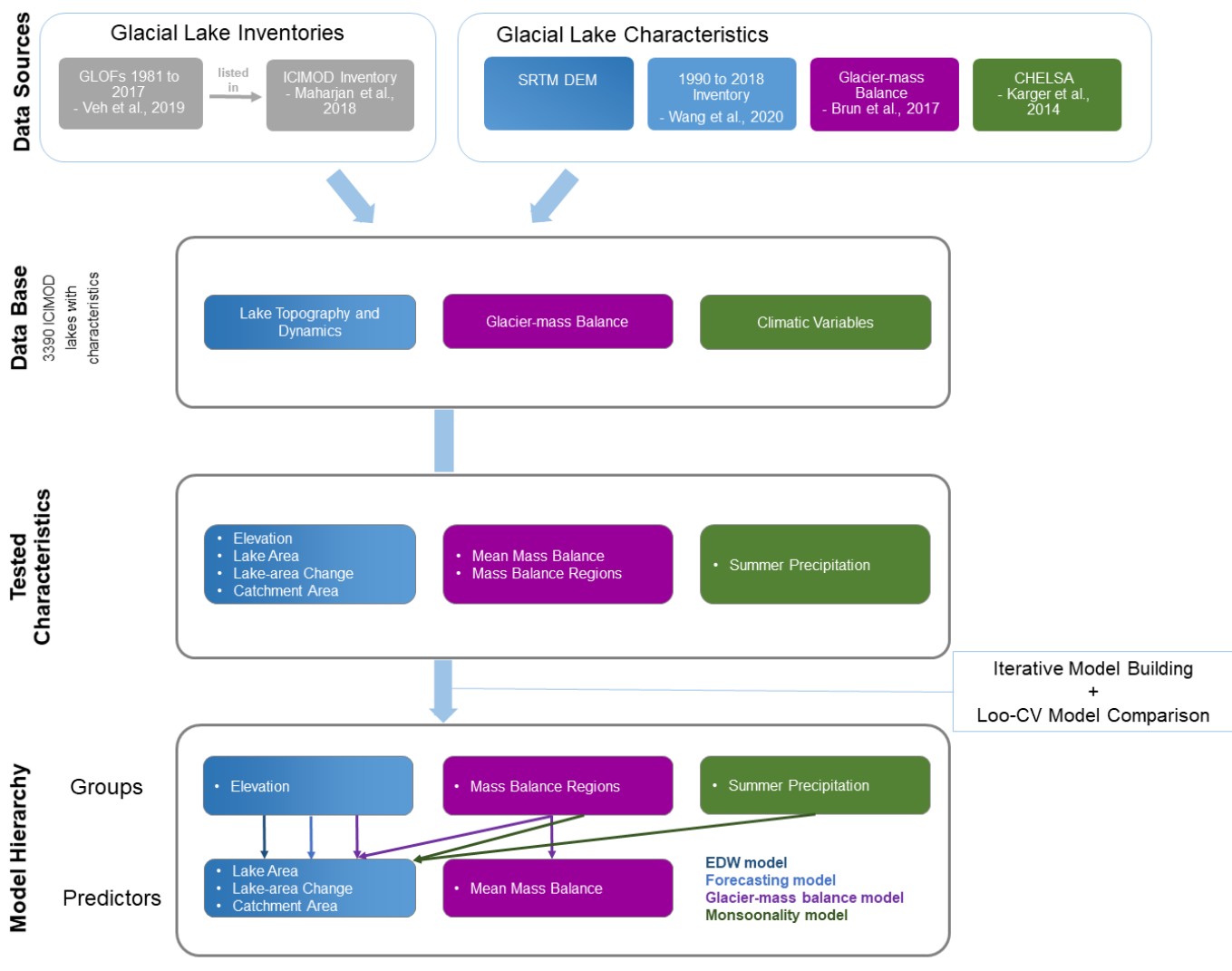

**Figure 2: Data sources and workflow; EDW = elevation-dependent warming.**

## 2.2 Bayesian multi-level logistic regression

We used logistic regression to learn the probability of whether a given lake in the HKKHN had a reported GLOF in the past four decades. This method was pioneered for moraine-dammed lakes in British Columbia (McKillop and Clague, 2007).

Logistic regression estimates a binary outcome $y$ from the optimal linear combination of $p$ weighted predictors $\mathbf{x} = \{x_1, \ldots, x_p\}$. The probability $y = P_{\text{GLOF}}$ that lake $i$ had released a GLOF is expressed as:

$$y_i \sim \text{Bernoulli}(\mu_i) \tag{1}$$

$$\mu_i = \text{S}\left(\alpha_0 + \beta_1 x_{i,1} + \beta_2 x_{i,2} + \cdots + \beta_p x_{i,p}\right) \tag{2}$$

where

$$\text{S}(x) = \frac{1}{1+\exp(-x)} \tag{3}$$

Here $\alpha_0$ is the intercept and $\boldsymbol{\beta} = \{\beta_1, \ldots, \beta_p\}^{\text{T}}$ are the $p$ predictor weights (Gelman and Hill, 2007). The logit function $\text{S}^{-1}(x)$ describes the odds on a logarithmic scale (the log-odds ratio) such that a unit increase in predictor $x_m$ raises the log-odds ratio

by an amount of $\beta_m$, with all other predictors fixed. We used standardised data to ensure that the weights measure the relative contributions of their predictors to the classification, whereas the intercept expresses the base case for average predictor values. Our strategy was to explore commonly reported predictors of GLOF susceptibility and dam stability as candidate predictors (Fig. 2, Table 1, Table 2). We further acknowledged that data on moraine-dammed lakes in the HKKHN are structured, reflecting, for example, the variance in topography and synoptic regime such as the summer monsoon in the eastern HKKHN

and westerlies in the western HKKHN. Different data sources, collection methods, and resolutions also add structure. This structure is routinely acknowledged, often raised as a caveat, but rarely treated, in GLOF studies. Ignoring such structure can lead to incorrect inference by bloating the statistical significance of irrelevant or inappropriate model parameter estimates (Austin et al., 2003). To explicitly address this issue, we chose a multi-level logistic regression as a compromise between a single pooled model and individual models for each group in the data ( Fig. 3; Gelman and Hill, 2007; Shor et al., 2007).


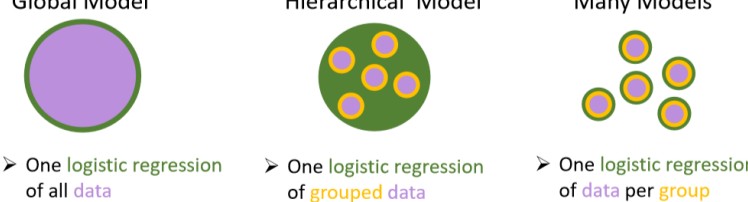

**Figure 3: Schematic comparison of global vs. multi-level logistic regression models.**

We recast Eq. (2) using a group index $j$:


$$\mu_i = \text{S}\left(\alpha_j + \beta_1 x_{i,1} + \beta_2 x_{i,2} + \cdots + \beta_p x_{i,p}\right) \tag{4}$$

$$\alpha_j \sim N(\mu_\alpha, \sigma_\alpha), \tag{5}$$

where $\mu_\alpha$ is the mean and $\sigma_\alpha$ is the standard deviation of the group-level intercepts $\alpha_j$ that are learned from all data and inform
each other via the model hierarchy. We used a Bayesian framework (Kruschke and Liddell, 2018) by combining the likelihood of observing the data with prior knowledge from previous GLOF studies (Fischer et al., 2020). The small number of reported GLOFs introduces strong imbalance to our data, given that some regions, and hence levels, had few or no reported GLOFs. Although this would be problematic in most other modelling approaches, Bayesian multi-level models are well suited for this kind of imbalanced training data (Gelman and Hill, 2007; Shor et al., 2007; Stegmueller, 2013).

We used the statistical programming language **R** with the package `brms`, which estimates joint posterior distributions using a Hamiltonian Monte Carlo algorithm and a No-U-Turn Sampler (NUTS) (Bürkner, 2017). We ran four chains of 1500 samples after 500 warm-up runs each, and checked for numerical divergences or other pathological issues. We only considered models with all values of $\hat{R} <$1.01, a measure of numerical convergence of sampling chains, to avoid unbiased posterior distributions (Nalborczyk et al., 2019).

Unless stated otherwise, we used a weakly informative half Student-t distribution with three degrees of freedom and a scale parameter of 10 for the standard deviations of group-level effects (Table 3; Bürkner, 2017; Gelman, 2006). At the population level, we chose weakly informative priors for the intercept and coefficients for which we had no other prior knowledge. We encoded this lack of knowledge with a prior Cauchy distribution centred at zero and with scale 2.5, following the recommendation by Gelman et al. (2008). Rapidly growing moraine-dammed lakes are a widely used predictor of high GLOF
susceptibility (Aggarwal et al., 2016; Allen et al., 2019; Bolch et al., 2011; Ives et al., 2010; Mergili and Schneider, 2011; Prakash and Nagarajan, 2017). We encoded this notion in a prior Gaussian distribution with one unit mean and standard deviation, hence shifting more probability mass towards positive regression weights without excluding the possibility of negative weight estimates (Table 3).

**Table 3: Prior distributions for group- and population-level effects.**

| Level | Model coefficient | Probability Density Function |
|---|---|---|
| Group-level effects | Standard deviation σ of group model variables | $\sigma_\alpha \sim HalfStudentT(3,10)$ |
| Population-level effects | Intercept | $\alpha_j \sim Cauchy(0,2.5)$ |
| | Weight of predictors with weak prior knowledge | $\beta_p \sim Cauchy(0,2.5)$ |
| | Weight of predictor lake area $\beta_A$ | $\beta_A \sim Normal(1,2)$ |

We estimated the predictive performance of all models with leave-one-out (LOO) cross-validation as part of the `brms` package (Bürkner, 2017). LOO values like the expected log predictive density (ELPD) summarise the predictive error of Bayesian

models, similar to the Akaike Information Criterion or related metrics of model selection (Vehtari et al., 2017). They are based
on the log-likelihood of the posterior simulations of parameter values (Vehtari et al., 2017).

## 3 Results

*Elevation-dependent warming model*

Our first model addresses the notion of elevation-dependent warming (EDW) by considering lake elevation as a grouping
structure in the data. The model further assumes that the GLOF history of a given lake is a function of its area $A$ and net change
$\Delta A$. This dependence differs up to a constant, i.e. the varying model intercept, across elevation bands $z$ that we define here in
five quantile grouping levels (Fig. 1). The model intercept may vary across these elevation bands, whereas lake area (in 2005)
and its net change remain fixed predictors. In essence, this varying-intercept model acknowledges that glacial lakes in the same
elevation band may have had a common baseline susceptibility to GLOFs in the past four decades. The indicator variable $\Delta A$
records whether a given lake had a net growth or shrinkage between 1990 and 2018:

$$\mu_i = S(\alpha_z + \beta_A A_i + \beta_{\Delta A} \Delta A_i) \tag{6}$$
$$\alpha_z \sim N(\mu_z, \sigma_z), \tag{7}$$

where index $z$ identifies the elevation band.

We obtain posterior estimates of $\beta_A = 0.79^{+0.27}/_{-0.27}$ and $\beta_{\Delta A} = 0.48^{+0.73}/_{-0.72}$ (95% highest density interval, HDI) that indicate
that larger lakes are more likely classified as having had a GLOF, whereas net growth or shrinkage has ambivalent weight as
its HDI includes zero (Fig. 4, Fig. 5, Table 4). On the population level, the low spread of intercepts ($\sigma_z = 0.29^{+0.68}/_{-0.28}$) estimated
for each of the five elevation bands shows that elevation effects modulate the pooled model only minutely. These posterior
effects are positive for the lower elevation bands, but negative for the higher elevation bands. Thus, the mean posterior
probability of a GLOF history, $P_{GLOF}$, under this model increases slightly for lakes in lower elevations and with larger surface
area in 2005. We also observe that $P_{GLOF} < 0.5$ regardless of reported lake elevation, and that the associated uncertainties are
higher for larger lakes.

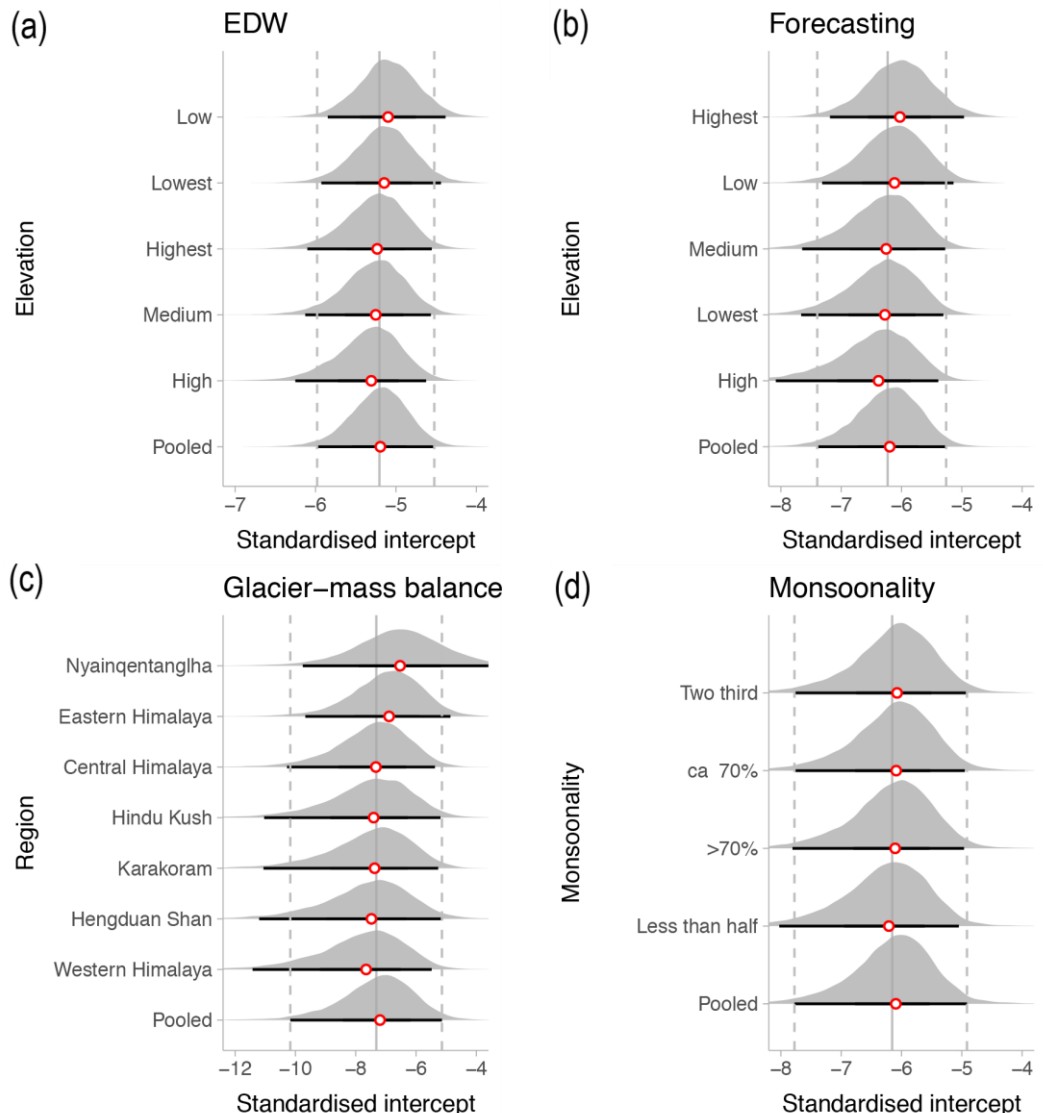

**Figure 4: Posterior pooled and group-level intercepts for the four models considered; EDW = elevation-dependent warming; see Fig. 1 for a summary of the quantiles of elevation and monsoonality. Black horizontal lines delimit 95% HDI, red circles indicate posterior medians. Vertical continuous (dashed) grey lines are posterior means (95% HDI) of the pooled intercept of each model. Intercepts are standardised and thus refer to lakes with average predictor values.**

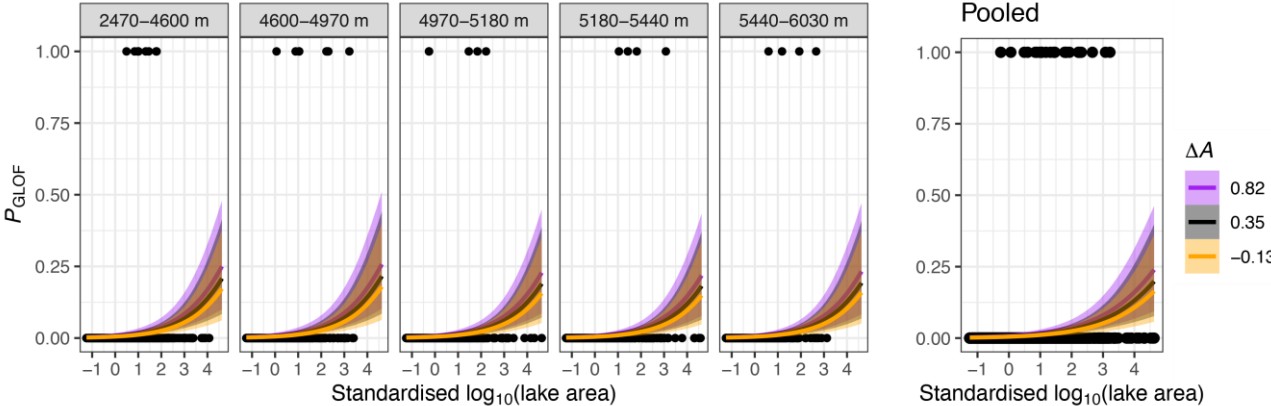


**Figure 5: Elevation-dependent warming model: posterior probabilities $P_{GLOF}$ as a function of standardised lake area $A$ (in 2005) and the sign of standardised lake-area change $\Delta A$ (i.e. net growth or shrinkage), grouped by quantiles of elevation (defined in Fig. 1; lowest: 2470-4600 m a.s.l., low: 4600-4970 m a.s.l., medium: 4970-5180 m a.s.l., high: 5180-5440 m a.s.l., highest: 5440-6030 m a.s.l.). Black dots are lake data with (no) reported GLOF records. Thick coloured lines are mean fits, and colour shades encompass the**
**associated 95% HDIs.**

*Forecasting model*

Our second model refines our approach by including only relative changes in lake area before the reported GLOFs happened. We can use this model to fore- or hindcast historic GLOFs in our inventory. Here we use lake area $A$ (in 2005) and its relative
change $A^{*a}$ from 1990 to 2005 as predictors of eleven GLOFs that occurred between 2005 and 2018 across the five elevation bands. We assume that larger and deeper lakes are more robust to relative size changes and thus also include a multiplicative interaction term between lake area and its change:

$$\mu_i = S(\alpha_z + \beta_A A_i + \beta_{A^{*a}} A_i^{*a} + \beta_{A \times A^{*a}} A_i \times A_i^{*a}) \tag{8}$$


We find that lake area has a credible positive posterior weight of $\beta_A = 0.86^{+0.44}/_{-0.43}$, hence greater lakes are more likely to having had a GLOF between 2005 and 2018. The weight of relative lake-area change in the 15 years before is ambiguous ($\beta_{A^{*a}} = -0.04^{+0.76}/_{-0.67}$) and so is the interaction ($\beta_{A \times A^{*a}} = -0.16^{+0.41}/_{-0.51}$). On average, however, relative increases in lake area between 1990 and 2005 slightly decrease $P_{GLOF}$. Unlike in the elevation-dependent warming model, the effects of elevation
bands are less clear, while the uncertainties are more pronounced and highest for larger and shrinking lakes (Fig. 4, Fig. 6).

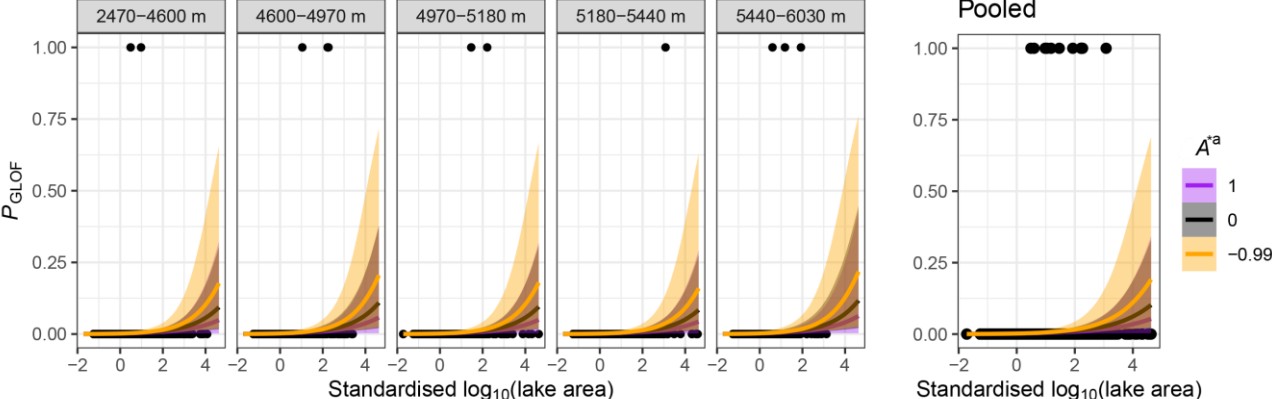

**Figure 6: Forecasting model: posterior probabilities $P_{GLOF}$ as a function of standardised lake area $A$ (in 2005) and standardised lake-area change $A^{*a}$ between 1990 and 2005, grouped by quantiles of elevation (defined in Fig. 1; lowest: 2470-4600 m a.s.l., low: 4600-4970 m a.s.l., medium: 4970-5180 m a.s.l., high: 5180-5440 m a.s.l., highest: 5440-6030 m a.s.l.). Black dots are lake data with (no) reported GLOF records for the interval 2005 to 2018. Thick coloured lines are mean fits, and colour shades encompass the associated 95% HDIs.**

*Glacier-mass balance model*

Besides elevation, our third model considers the average historic glacier-mass balances across the HKKHN. The model assumes that mean ice losses $\Delta m$ add a distinctly regional structure to the susceptibility to GLOFs in the past four decades, given that accelerated glacier melt may raise GLOF potential (Emmer, 2017; Richardson and Reynolds, 2000). We use the seven RGI regions as defined by Brun et al. (2017) as group-levels $r$ and their average glacier-mass balance as a group-level predictor $\Delta m_r$. Our pooled predictors are the relative change of lake area $A^{*b}$ from 2005 to 2018 (to ensure a comparable time interval) and the catchment area $C$ upstream of each lake. We replace lake area by its upstream catchment area, which is less prone to change, but well correlated to lake area.

$$\mu_i = S(\alpha_z + \alpha_r + \beta_{A^{*b}} A_i^{*b} + \beta_C C_i), \tag{9}$$

$$\alpha_r \sim N(\mu_r + \gamma_r \Delta m_r, \sigma_r). \tag{10}$$

This model returns a positive weight for catchment area ($\beta_C = 0.85^{+0.50}/_{-0.50}$) and a negative weight for relative lake-area changes ($\beta_{A^{*b}} = -0.69^{+0.64}/_{-0.61}$), whereas the effect of the mean glacier-mass balance remains inconclusive ($\gamma_r = -2.98^{+4.87}/_{-6.70}$). On the basis of higher standard deviations, we learn that effects of glaciological regions vary more than those of elevation bands ($\sigma_r = 0.81^{+1.60}/_{-0.78}$ and $\sigma_z = 0.48^{+1.19}/_{-0.47}$). When training this model on a subset of glacial lakes with documented GLOFs that happened after 2000 (i.e. including only those in the interval covered by glacier-mass balance data), posterior estimates of $\sigma_r$

increase to $1.11^{+1.77}/_{-1.03}$, further underlining our result that glacier-mass balance credibly affects $P_{GLOF}$. This is also reflected in the posterior distributions across the glacier-mass balance regions (Fig. 4) as well as the calculated group-level effects. This model has the highest values of $P_{GLOF}$ for average lakes (i.e. all average predictor values combined) in the Nyainqentanglha Mountains and the Eastern Himalaya (Fig. 4). In contrast to the forecasting model, we observe that increases in lake area now

credibly depress $P_{GLOF}$ (Fig. 7).

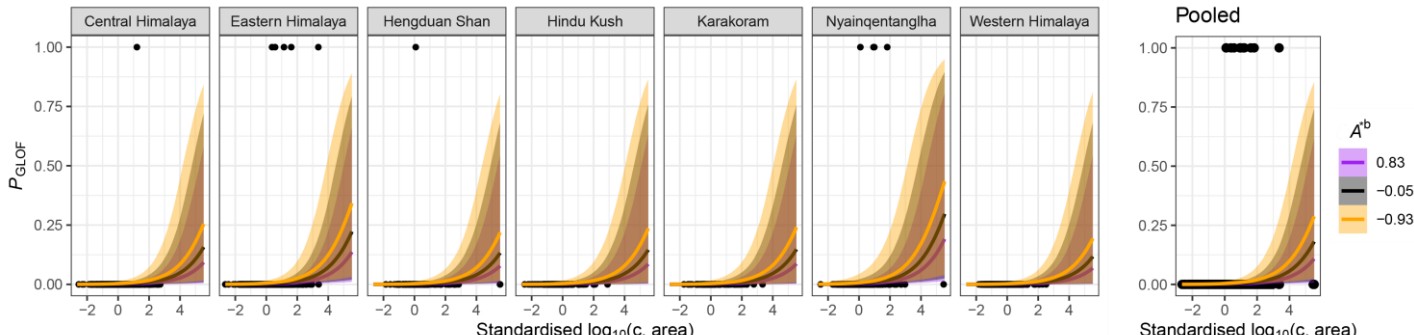

**Figure 7: Glacier-mass balance model: posterior probabilities $P_{GLOF}$ as a function of standardised catchment area $C$ and standardised lake-area change $A^{*b}$ between 2005 and 2018, grouped by regions of average glacier-mass balance (see Fig. 1). Black**
**dots are lake data with (no) reported GLOF records for the interval 2005 to 2018. Thick coloured lines are mean fits, and colour shades encompass the associated 95% HDIs.**

*Monsoonality model*

Our last model explores a synoptic influence on GLOF susceptibility by grouping the data by the summer proportion of mean
annual precipitation and thus by approximate monsoonal contribution. We defined five monsoonality levels based on quantiles of the annual proportions of summer precipitation (Fig. 1). We use relative lake-area change $A^{*c}$ between 1990 and 2018, and catchment area $C$ as population-level predictors, as well as the additional grouping by regional glacier-mass balance:

$$\mu_i = S(\alpha_M + \alpha_r + \beta_{A^{*c}}A_i^{*c} + \beta_C C_i), \tag{11}$$
$$\alpha_M \sim N(\mu_M, \sigma_M), \tag{12}$$

where index $M$ identifies the monsoonality group. We find that larger catchment areas ($\beta_C = 0.82^{+0.46}/_{-0.48}$) and lakes with relative shrinkage ($\beta_{A^{*c}} = -0.63^{+0.59}/_{-0.59}$) credibly raise $P_{GLOF}$ (Fig. 4, Fig. 8). Higher standard deviations show that regional effects vary more for the mean glacial-mass balance than for monsoonality ($\sigma_r = 0.79^{+1.59}/_{-0.76}$ and $\sigma_M = 0.40^{+1.04}/_{-0.39}$), although
both hardly change the pooled model trend.

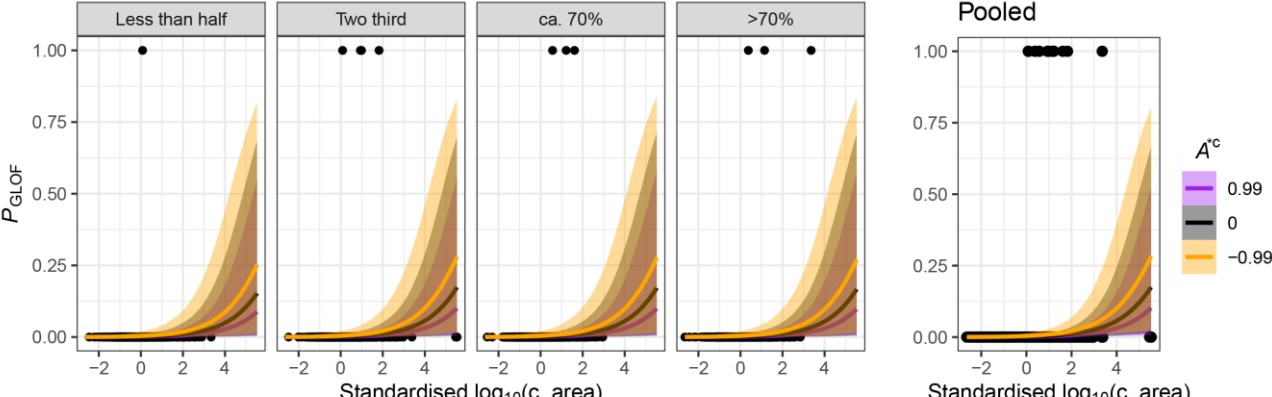

**Figure 8: Monsoonality model: posterior probabilities $P_{GLOF}$ as a function of standardised catchment area $C$ and standardised lake-area change $A^{*c}$ between 1990 and 2018, grouped by quantiles of the annual proportion of precipitation falling during summer (defined in Fig. 1). Black dots are lake data with (no) reported GLOF records for the interval 1990 to 2018. Thick coloured lines are mean fits, and colour shades encompass the associated 95% HDIs.**

**Table 4: Summary of the results of our four models.**

| Model | Model parameter | Estimate | Estimation error | Lower 95% CI boundary | Upper 95% CI boundary |
|---|---|---|---|---|---|
| Elevation-dependent warming model | $\alpha_z$ | -5.22 | 0.36 | -5.96 | -4.56 |
| | $\beta_A$ | 0.79 | 0.14 | 0.52 | 1.06 |
| | $\beta_{AA \, (1990 \, to \, 2018)}$ | 0.49 | 0.38 | -0.28 | 1.24 |
| | $\sigma_z$ | 0.28 | 0.27 | 0.01 | 0.99 |
| Forecasting model | $\alpha_z$ | -6.23 | 0.54 | -7.39 | -5.26 |
| | $\beta_A$ | 0.87 | 0.22 | 0.44 | 1.31 |
| | $\beta_{A^{*a} \, (1990 \, to \, 2005)}$ | -0.04 | 0.38 | -0.71 | 0.73 |
| | $\beta_{A \times A^{*a} \, *}$ | -0.16 | 0.24 | -0.67 | 0.26 |
| | $\sigma_z$ | 0.43 | 0.41 | 0.01 | 1.49 |
| Glacier-mass balance model | $\alpha_{z,r}$ | -7.31 | 1.26 | -10.15 | -5.19 |
| | $\beta_{A^{*b} \, (2005 \, to \, 2018)}$ | -0.69 | 0.32 | -1.31 | -0.06 |
| | $\beta_C$ | 0.85 | 0.26 | 0.35 | 1.36 |
| | $\gamma_r$ | -2.90 | 2.80 | -9.27 | 1.80 |
| | $\sigma_z$ | 0.47 | 0.44 | 0.01 | 1.61 |
| | $\sigma_r$ | 0.83 | 0.66 | 0.03 | 2.47 |
| Monsoonality model | $\alpha_{M,r}$ | -6.14 | 0.70 | -7.70 | -4.91 |
| | $\beta_{A^{*c} \, (1990 \, to \, 2018)}$ | -0.63 | 0.31 | -1.23 | -0.02 |
| | $\beta_C$ | 0.82 | 0.24 | 0.34 | 1.28 |

| | | 0.40 | 0.42 | 0.01 | 1.49 |
|---|---|---|---|---|---|
| | $\sigma_M$ | 0.40 | 0.42 | 0.01 | 1.49 |
| | $\sigma_r$ | 0.78 | 0.62 | 0.03 | 2.31 |

*Model performance and validation*

We estimate the performance of our models in terms of the posterior improvement of our prior chance of finding a lake with known outburst in the past four decades in our inventory by pure chance. We compare the posterior predictive mean $P_{GLOF}$ with a mean prior probability that we estimate from the ~1% proportion of lakes with known GLOFs in our training data. We measure what we have learned from each model in terms of the log-odds ratio that readily translates into probabilities using

Eq. (3). A positive log-odds ratio means that we obtain a higher posterior probability of attributing a historic GLOF to a given lake compared to a random draw. Negative log-odd ratios indicate lakes for which the posterior probability of a reported GLOF is lower than the prior probability. Based on this metric, all models have higher true positive than true negative rates. For a prior probability informed by the historic frequency of GLOFs, the models have at least about 80% true positives, and at least 70% true negatives on average (Fig. 9, Table 5).

The values of the LOO cross-validation of the predictive capabilities show that the EDW model formally has the least favourable, i.e. higher, values for both LOO metrics (Table 5). This is potentially due to the different true positives counts in the training data sets. However, the range of estimated ELPD values between the remaining three models is small ($\Delta$ELPD = 1.9).


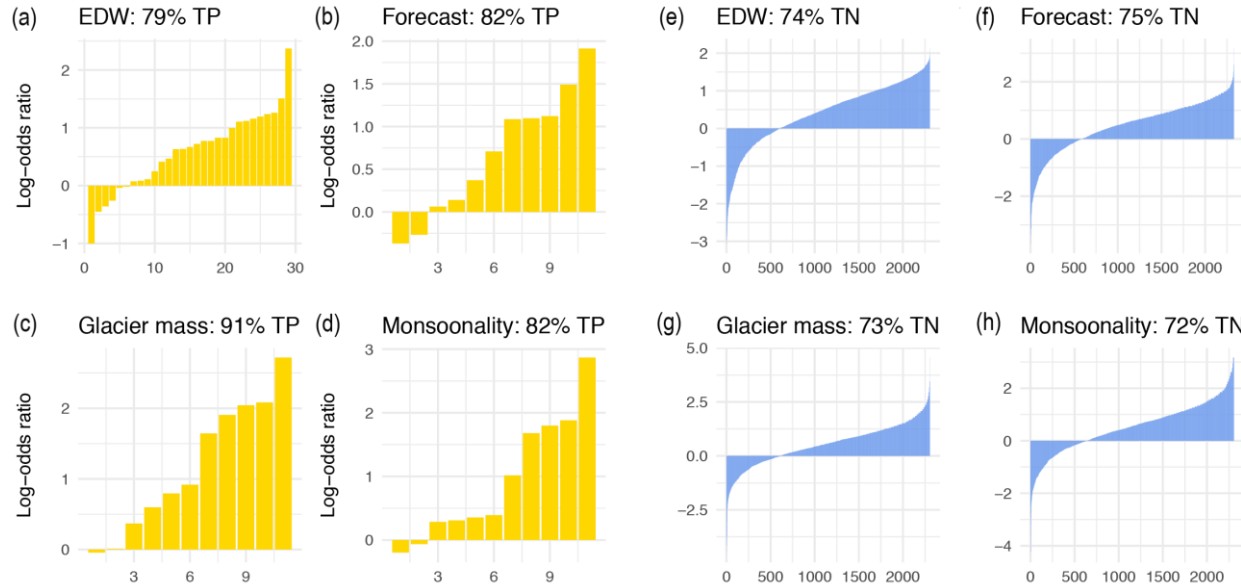

**Figure 9: Average posterior log-odds ratios for true positives TP (negatives, TN), i.e. lakes with (without) a GLOF in the period 1981 – 2018 (a and e) and 2005 – 2018 (b-d and f-h) on the x axis for the four different models. The log-odds ratios describe here the ratio of the mean posterior over the mean prior probability of classifying a given lake as having had a GLOF. We estimate the mean prior probability from the relative frequency of GLOFs in the datasets; EDW = elevation-dependent warming model.**


**Table 5: Overview of model validation measures for the predictive capabilities of our models.**

| Model | Prior vs. posterior knowledge | | LOO cross-validation metrics | |
|---|---|---|---|---|
| | % true positives / % true negatives correctly identified | % false positives / % false negatives incorrectly identified | ELPD | LOOIC |
| Elevation-dependent warming model | 79% / 74% | 21% / 26% | -144.2 | 288.3 |
| Forecasting model | 82% / 75% | 18% / 25% | -66.5 | 132.9 |
| Glacier-mass balance model | 91% / 73% | 9% / 27% | -64.6 | 129.1 |
| Monsoonality model | 82% / 72% | 18% / 28% | -65.6 | 131.2 |

## 4 Discussion

### 4.1 Topographic and climatic predictors of GLOFs

We used Bayesian multi-level logistic regression to test whether several widely advocated predictors of GLOF susceptibility and glacial lake stability are credible predictors of at least one outburst in the past four decades. All four models that we considered identify **lake area** and **catchment area** as predictors with weights that credibly differ from zero with 95% probability. Our model results quantitatively support qualitative notions of several basin-wide studies in the HKKHN (Bolch et al., 2011; Ives et al., 2010; Mergili and Schneider, 2011) and elsewhere (McKillop and Clague, 2007), which proposed that

larger moraine-dammed lakes have a higher potential for releasing GLOFs.

We also found that **changes in lake area** have partly inconclusive influences in the models. Two exceptions are the negative weight of lake-area changes $\beta_{A*b}$ and $\beta_{A*c}$ in the glacier-mass balance model and in the monsoonality model, regardless of the differing intervals that these changes were determined for (Table 4). While this result formally indicates that shrinking lakes are more likely to be classified as having had a historic GLOF, the period over which these lake-area changes are valid (2005

to 2018) overlaps with the timing of eleven recorded GLOFs (Eq. 9). In other words, the lake shrinkage could be a direct consequence of these GLOFs instead of vice versa. Nonetheless, our results indicate that lake-area changes, either absolute or directional, are somewhat inconclusive in informing us whether a given lake has a recent GLOF history. One advantage of our Bayesian approach is that we can express the role of lake-area changes in GLOF susceptibility by choosing different highest density intervals. For example, if we adopted a narrower, say 80% HDI for $\Delta A$, we could be 80% certain that net lake-area

growth increased $P_{GLOF}$ under the elevation-dependent warming model (Eq. 6). However, in the forecasting model, in which we tested whether differing data observation periods have any credible effects, the influence of lake-area change remains negligible even for <50% HDIs. We thus conclude that relative lake-area change before outburst is an inconclusive predictor. This result contradicts the assumptions made in many previous studies that argued that rapidly growing lakes are the most prone to sudden outburst (Aggarwal et al., 2016; Bolch et al., 2011; Ives et al., 2010; Mergili and Schneider, 2011; Prakash

and Nagarajan, 2017; Rounce et al., 2016; Wang et al., 2012).

The role of **elevation** in GLOF predictions is also less pronounced than that of lake or catchment area, at least as a group level. The weights of the elevation-dependent warming model indicate that lower (higher) lakes are slightly more (less) likely to have had a historic GLOF (Fig. 4), but hardly warrant any better model performance compared to the pooled (or elevation-independent) model. In the forecasting model, however, the contributions of lake elevation to $P_{GLOF}$ are devoid of any

systematic pattern and likely reflect several, potentially combined, drivers (Fig. 4). This model was trained on fewer GLOFs and the imbalance in the data introduces more uncertainties in terms of broad 95% HDIs. Clearly, the role of elevation may need more future investigation. In terms of elevation bands, it hardly seems to aid GLOF detection with the models used here. Similarly, Emmer et al. (2016) reported that lake elevation was hardly affecting GLOF hazard in the Cordillera Blanca, Peru.

Judging from the regionally averaged **glacier-mass balances**, our models predict the highest GLOF probabilities in the Nyainqentanglha Mountains and the Eastern Himalaya, which have had the highest historic GLOF counts (Fig. 1). The timing and seasonality of snowfall affects how glaciers respond to rising air temperatures. Observed frequencies and predicted probabilities of historic GLOFs are lowest for several glaciers with positive mass balance in the Karakoram and Western Himalayas (Fig. 1, Fig. 10). Most moraine-dammed lakes in the HKKHN, however, are fed by glaciers with negative mass balances that likely help to elevate GLOF potential through increased meltwater input and glacier-tongue calving rates (Emmer, 2017; Richardson and Reynolds, 2000). More than 70% of all lakes that burst out in the past four decades were in contact to their parent glaciers (Veh et al., 2019). However, systematically recorded time series of glacier fronts are even harder to come by when compared to systematic measurements of changes in glacial-lake areas. Given that the regional glacier-mass balance is linked to synoptic precipitation patterns (Kapnick et al., 2014; King et al., 2019; Krishnan et al., 2019), our glacier-mass balance model highlights that the regional ice loss outweighs the role of monsoonality in terms of higher changes to the group-level intercepts for comparable mean $P_{GLOF}$ and associated uncertainties (Fig. 4, Fig. 7, Fig. 8).

Our results offer insights into the links between historic GLOFs and the **synoptic precipitation patterns**. Richardson and Reynolds (2000) presumed that seasonal floods and GLOFs are both caused by high monsoonal precipitation and summer ablation. In contrast, our results indicate that the fraction of summer precipitation changes the predictive probabilities of historic GLOFs only marginally, at least at the group level, so that deviations from a pooled model for the HKKHN are minute when compared to the spread of posterior group-level intercepts in the other models (Fig. 4). In essence, our results underline the need for exploring more the interactions of both precipitation and temperature as potential GLOF triggers. It may well be that seasonal timing of heavy precipitation events and type (rain or snow) at a given lake may be more meaningful to GLOF susceptibility than annual totals or averages. Whether our finding that glacier-mass balances driven by superimposed synoptic regimes credibly influence regional GLOF susceptibility in the HKKHN is applicable to other regions, for example the Cordillera Blanca in the South American Andes (Emmer et al., 2016; Emmer and Vilímek, 2014; Iturrizaga, 2011), also needs further investigation.

**4.2 Model Assessment**

We consider our quantitative and data-driven approach as complementary to existing qualitative and basin-wide GLOF hazard appraisals. Our models cannot replace field observations that deliver local details on GLOF-disposing factors such as moraine or adjacent rock-slope stability, presence of ice cores, glacier calving rates, or surges. Our selection of predictors is a compromise between widely used predictors of GLOF susceptibility and hazard and their availability as data covering the entire HKKHN. To this end, we used lake (or catchment) area and lake-area changes as predictors, and elevation, regional glacier-mass balance, and monsoonality as group levels of past GLOF activity of several thousand moraine-dammed lakes in the HKKHN. Among the many possible combinations of predictors and group levels we focused on those few combinations

with minimal correlation among the input variables. We minimised the potential for misclassification by using a purely remote-sensing-based inventory of GLOFs, which reduces reporting bias for GLOFs too small to be noticed or happening in unpopulated areas: more destructive GLOFs are recorded more often than smaller GLOFs in remote areas (Veh et al., 2018, 2019). We are thus confident that we trained our models on lakes with a confirmed GLOF history at the expense of discarding known outbursts predating the onset of Landsat satellite coverage in 1981. We acknowledge that climate products such as precipitation can have large biases because of orographic effects or climate circulation patterns and interpolation using topography (Karger et al., 2017; Mukul et al., 2017). Cross-validation of CHELSA precipitation estimates with station data has a global mean coefficient of determination $R^2$ of 0.77, with regional variations between 0.53 and 0.90 (Karger et al., 2017). By accounting for orographic wind effects, CHELSA products outperform previous global datasets such as the WorldClim (Hijmans et al., 2005), especially in the rugged HKKHN topography. We stress that we therefore used all climatic data as aggregated group-level variables to avoid spurious model results. At the level of individual lakes, we thus resorted only to size, elevation, and upstream catchment area as more robust predictors.

Due to strong imbalance in our training data, we opted for prior vs. posterior log-odd comparison instead of commonly applied Receiver Operating Characteristics (ROC) in estimating the predictive capabilities of our models (Saito and Rehmsmeier, 2015). In our models, only few posterior estimates of $P_{GLOF}$ are >0.5 and they, thus, offer very conservative estimates of a GLOF history (Fig. 10). All models have wide 95% HDIs that attest a high level of uncertainty. This observation may be sobering, but nevertheless documents objectively the minimum amount of accuracy that these simple models afford for objectively detecting historic outbursts.

The low fraction of lakes with a GLOF history (~1%) curtails a traditional logistic regression model and favours instead a Bayesian multi-level approach that can handle imbalanced training data and collinear predictors (Gelman and Hill, 2007; Hille Ris Lambers et al., 2006; Shor et al., 2007). We prefer the straight-forward interpretation of posterior regression weights to random forest classifiers, neural networks or support vector machines (Caniani et al., 2008; Falah et al., 2019; Kalantar et al., 2018; Taalab et al., 2018). While these methods may perform better, they disclose little about the relationship between model inputs and outputs (Blöthe et al., 2019; Dinov, 2018); much of their higher accuracy is also linked to the overwhelming number of true negatives. Yet so far, multi-criteria decision analysis or decision-making trees have been the method of choice in GLOF hazard assessments, both in High Mountain Asia (Bolch et al., 2011; Prakash and Nagarajan, 2017; Rounce et al., 2016; Wang et al., 2012) and elsewhere (Emmer et al., 2016; Emmer and Vilímek, 2014; Huggel et al., 2002; Kougkoulos et al., 2018). While these methods strongly rely on expert judgement (Allen et al., 2019), a Bayesian logistic regression encodes any prior knowledge or constraints explicitly and reproducibly as probability distributions. Still, inconsiderate or inappropriate prior choices can introduce bias (Van Dongen, 2006; Kruschke and Liddell, 2018). Therefore, we carefully considered our choice of weakly informative priors for predictors with limited prior knowledge, following the guidelines concerning regression

models by Gelman (2006) and Gelman et al. (2008). We also cross-checked our results when applying varying prior choices and found negligible differences in the resulting posterior distributions.

To summarise, our simple classification models hardly support the notion that elevation or changes in lake area are straightforward predictors of a GLOF history, at least for the moraine-dammed lakes that we studied in the HKKHN. Lake size and regional differences in glacier-mass balance are items that future studies of GLOF susceptibility may wish to consider further. The performance of these models is moderate to good if compared to a random classification, yet associated with high uncertainties in terms of wide highest density intervals. We underline that these uncertainties have rarely been addressed, let alone quantified, in previous work. One way forward may be to create ensembles of such models to improve their predictive capability instead of relying on any single model.

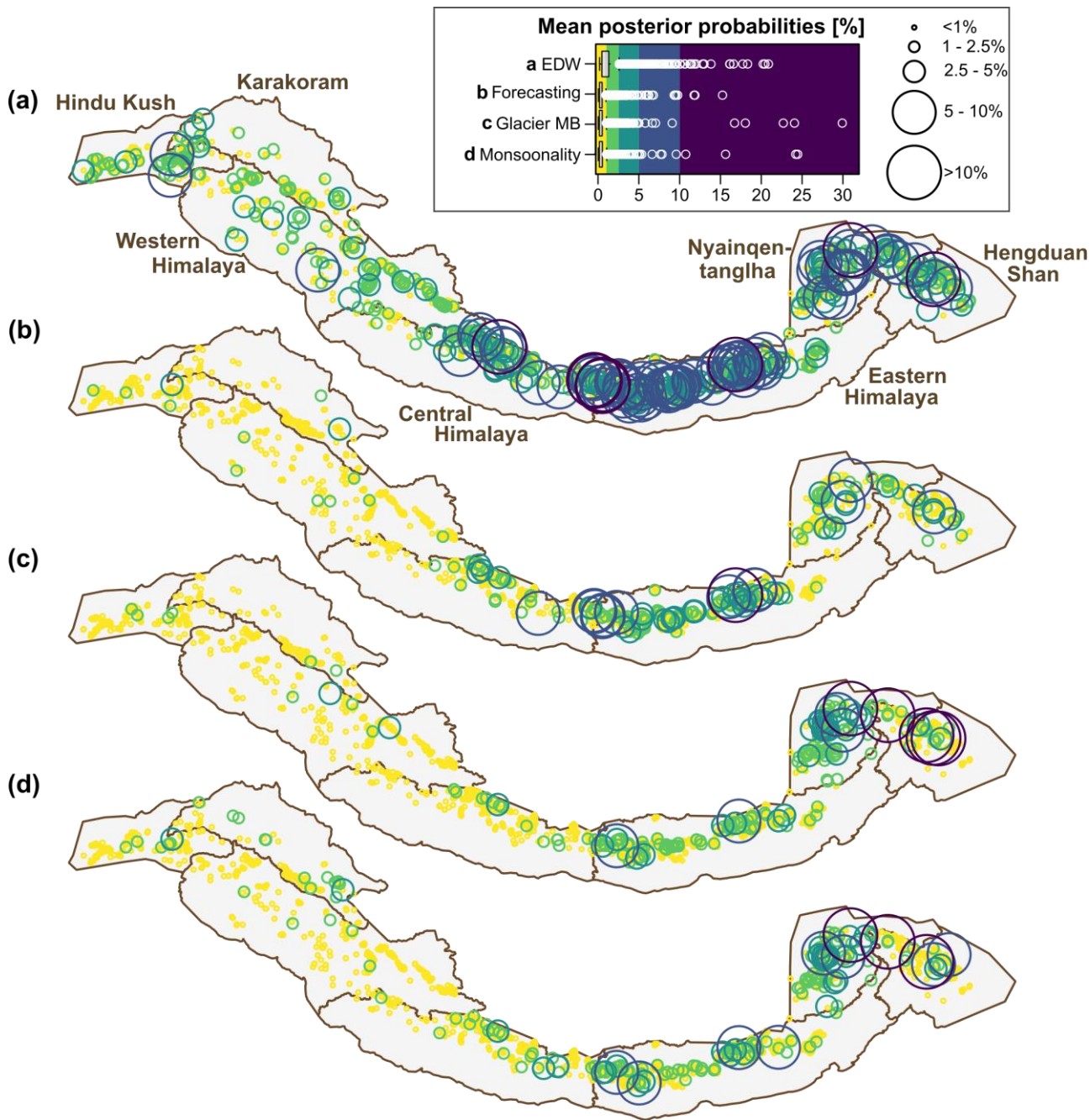

**Figure 10: Mean posterior probabilities of HKKHN glacial lakes for having had a GLOF history ($P_{GLOF}$) in the past four decades as estimated in the (a) elevation-dependent warming model, (b) forecasting model, (c) glacier-mass balance model, and (d) monsoonality model. Size and colours of bubbles are scaled by posterior probabilities.**


## 5 Conclusions

We quantitatively investigated the susceptibility of moraine-dammed lakes to GLOFs in major mountain regions of High Asia. We used a systematically compiled and comprehensive inventory of moraine-dammed lakes with documented GLOFs in the past four decades to test how elevation, lake area and its rate of change, glacier-mass balance, and monsoonality perform as predictors and group levels in a Bayesian multi-level logistic regression. Our results show that larger lakes in larger catchments have been more prone to sudden outburst floods, as have those lakes in regions with pronounced negative glacier-mass balance.

While elevation-dependent warming (EDW) may control a number of processes conducive to GLOFs, grouping our classification by elevation bands adds little to a pooled model for the entire HKKHN. Historic changes in lake area, both in absolute and relative values, have an ambiguous role in these models. We observed that shrinking lakes favour the classification as GLOF-prone, although this may arise from overlapping measurement intervals such that the reduction in lake size arises from outburst rather than vice versa. In any case, the widely adapted notion that (rapid) lake growth may be a predictor of

impending outburst remains poorly supported by our model results. Our Bayesian approach allows explicit probabilistic prognoses of the role of these widely cited controls on GLOF susceptibility, but also attests to previously hardly quantified uncertainties, especially for the larger lakes in our study area. While individual models offer some improvement with respect to a random classification based on average GLOF frequency, we recommend considering ensemble models for obtaining more accurate and flexible predictions of outbursts from moraine-dammed lakes.

## Data and code availability


This study is based on freely available data. Shuttle Radar Topography Mission (SRTM) data are available from the US Geological Survey (https://www.earthexplorer.usgs.gov). We derived climatic variables from the CHELSA Bioclim data set (https://chelsa-climate.org/bioclim/) described by Karger et al. (2017) and regional glacier-mass balances from Brun et al. (2017). We extracted glacial lake information from inventories published by Maharjan et al. (2018), Veh et al. (2019), and

Wang et al. (2020). We processed our data with free **R** statistical software (https://cran.r-project.org/), including the `brms` package by Bürkner (2017) (https://CRAN.R-project.org/package=brms). The model code to this article by Fischer et al. (2020) is published in a GitHub repository and is available online at: https://doi.org/10.5281/zenodo.4161577.

## Author contributions

This study was conceptualised by all authors. While formal analysis and methodology were conducted by MF and OK, data

curation was mainly carried out by GV. Visualisations of data and results, including maps, were prepared by GV, OK and MF. MF prepared the original manuscript; OK, GV, and AW reviewed and edited the writing.

## Competing interests

The authors declare that they have no conflict of interest.

## Acknowledgements

This research was funded by the Deutsche Forschungsgemeinschaft (DFG) via the graduate research training group NatRiskChange (GRK 2043/1) at the University of Potsdam (https://www.natriskchange.de). We thank Adam Emmer and Holger Frey for their helpful reviews of an earlier version of this article.

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
