# Peer review of "Controls of outbursts of moraine-dammed lakes in the greater Himalayan region"

_The Cryosphere, 2020_

## Referee Comment (RC1) · Adam Emmer (Referee) · 13 Jan 2021

General comments: The authors employ Bayesian multi-level regression to quantitatively investigate possible GLOF indicators (controls) in the HKKHN region, building on the inventory of 3,390 moraine-dammed lakes and 31 historical GLOFs. The study is well-structured and well-written, employed methods are statistically sound. I found this study of potential interest for readers of The Cryosphere.

The authors present interesting results, some of which are novel in a sense that contradict assumptions of previous GLOF hazard assessment studies (e.g. the assumption that fast-growing lakes are more susceptible to GLOF), but this is only one part of the story (so far pretty much model-oriented) in my opinion. If the overall aim is enhanced

identification of potential future GLOF sites or so, stronger linkages of investigated GLOF indicators to physical processes behind as well as (at least brief) character- ization of documented GLOFs (in terms of triggers, mechanisms) are missing. For instance, how (process-wise) is the EDW, glacier-mass balance or lake (catchment) area linked to documented GLOF? What are triggers of historic GLOFs considered in this study? In fact, I'd expect this to be taken into consideration in the very first step – selection and justification of GLOF indicators.

It would be interesting at least discuss how many of documented GLOFs were actually triggered by processes associated with investigated GLOF indicators? This is briefly touched in the introduction (L36-39) or study area section (L108), but I'm convinced that bit deeper and more comprehensive elaboration (e.g. a separate discussion section) would be beneficial for readers. Another example - on L244-245 it is mentioned that 'greater lakes are more likely to having had a GLOF . . .'. I wonder what do primary data say about this – what proportion of these 31 GLOF-producing lakes would be classified as large at the time of GLOF and what this proportion is in the population of 3,390 moraine-dammed lakes? And in the other way around - can a specific combination of values of GLOF indicators infer about possible (likely) GLOF trigger and mechanism (if not known)?

Let me also critically comment on some of the selected GLOF susceptibility indicators (in general, I'm convinced it would be useful presenting these indicators in a separate table with more detailed and comprehensive description than stated in the overview Tab. 1, and in places of the text):

- Lake area change – I'm aware this indicator is always tricky to define and em- ploy; according to what is written on L134-135, two intervals are used for lake are change (1990-2005 and 2005-2018); considering GLOFs occurring throughout the pe- riod 1981-2017, it means than these intervals may be pre-GLOF, post-GLOF or the GLOF occurred somewhen during one of these intervals – please comment on how this inconsistency was treated and whether it can explain that no link was observed
between lake area change and the occurrence of GLOF

- Glacier mass balance – similarly to my comment on lake area change - how can 2000-2016 glacier mass balance be used to explain GLOFs occurring throughout the period 1981-2017? These characteristics (mass balance as well as lake area change) are dynamic in nature and I'm wondering how can a static information from available datasets possibly blur a GLOF signal, especially for pre-2000 GLOFs?

- Monsoonality – using climate indicators in GLOF research is promising, but proportion of summer precipitation doesn't tell you about the extremity; for instance, the proportion will be lower in areas where extreme rainfalls occur in summer, but also some precipitation in winter, but will be super-high in generally dry areas with some precipitation during the summer and no precipitation in winter. But process-wise, the first area will have much higher potential to trigger GLOF in my opinion

I'm aware that these comments are somewhat tricky to deal with, but I'd appreciate some reflection in methods / discussion section.

Minor comments: L11: yes, the approach is quantitative, but selection of GLOF indicators in this study is also expert judgement-based as the authors are GLOF experts

L34: see also Cook et al., 2018, Science

L36-37: this needs deeper elaboration in relation to selected GLOF susceptibility indicators (see also my general comment)

L103: I suggest to use 'GLOF susceptibility indicators' instead of 'diagnostics of GLOF potential' or 'diagnostics of GLOF hazard' (L125); similarly, 'controls' and 'predictors' are used throughout the manuscript, please define a difference or unify

L111-112: lake deepening increases hydrostatic pressure, not areal or volumetric growth

L115-116: the authors usually argue that larger lakes are more susceptible because

large lake areas are more exposed to slope movements potentially triggering GLOFs; large area is also correlated with larger depth (and so hydrostatic pressure acting on a dam)

L130: how is different date of GLOF and input data for model treated? (how possibly different environmental conditions at the time of GLOF and at the time of datasets acquisition can influence your results?) see also my general comments

L136-139: I suggest to move this to L133

Fig. 2: three lake inventories are mentioned (ICIMOD, Veh et al., 2019 and Wang et al., 2020); please make clear how these were integrated; these 3,390 lakes (L131) are from which inventory?

L166: delete 's'

Fig. 3: how about green color in Many Models part?

L176: delete ','

Tab 2: what is PDF?

L207: what is meant by 'common susceptibility'?

L262-263: this step is not clear to me? Please explain

L263: please provide details about this correlation

L272: what is meant by 'average lake'?

Tab. 4: please also consider presenting false positives and false negatives

L352-354: this can be true for a specific period in long-term evolution of a mountain range (considering gradual glacier retreat and overall shift of all rapid processes including GLOFs to higher elevation zones; i.e. the general shift of morphoclimatic zones)

L365: so why not to consider this indicator in your model?

L373: are 'minute'? Please check

Fig. 10: please consider highlighting GLOF-producing lakes; switch a-d in the panel (e)

- - - To sum up, I'm convinced this is an interesting study worthy publishing, but I recommend some moderate to major revisions to be done first. I invite the authors to confront and synthesize their predominantly model-oriented study with processes behind past GLOFs and provide some insights into issues I raised. Thank you.

---

## Referee Comment (RC2) · Holger Frey (Referee) · 20 Jan 2021

GENERAL COMMENTS:

Based on an inventory of moraine-dammed glacial lakes and an inventory of 31 glacial lake outburst floods (GLOFs) from 1981-2018 in the Hindu-Kush Karakoram Himalaya Nyainqentanglha (HKKHN), the authors apply four Bayesian multi-level logistic regression models to estimate the susceptibility of these lakes for GLOFs. As predicting factors they combine lake elevation, lake area, lake area change rate, glacier-mass balance, and monsoonality, factors that are often used in regional-scale GLOF hazard assessments. They find that lake area is a useful predictor of GLOF susceptibility, as well as glacier mass balance. In contrast, lake area changes do not significantly

improve GLOF susceptibility estimates, which contradicts several existing lake hazard assessment schemes, where lake growth sometimes is an important hazard indicator.

As the authors point out, a large number of regional GLOF hazard assessments and assessment approaches is available for the HKKHN and other glaciated mountain ranges, many of them using a combination of weighted parameters, with a related number of different inventories and differing lists of potentially dangerous glacial lakes. This is a major challenge for decision makers and authorities responsible for hazard and risk management and planning. This study now presents for the first time a data driven, quantitative evidence on the ability of different parameters for posterior predictions of GLOF events, i.e. to estimate GLOF susceptibility.

Such quantitative assessments are highly needed and of major interest to the scientific community. The article is well written and structured and meets the requirements of a scientific publication (for instance regarding citing relevant works etc.), and the topic fits perfectly to the scope of The Cryosphere. However, there are a few aspects, detailed below, that need to be improved before publication.

SPECIFIC COMMENTS:

Hazard concept: The article is strictly focusing on GLOF susceptibility and using this term consistently throughout the manuscript. Nevertheless, I think regarding some aspects of the study, concepts and terminologies are mixed at some places. According to international standards from UNISDR, IPCC etc., hazard is a function of probability (of occurrence) and intensity (or magnitude). Susceptibility in turn 'is a relative measure of the likelihood (or probability) that a hazard will occur or initiate from a given site, based on intrinsic properties and dynamic characteristics of that site' and 'has an inverse relationship with stability' (GAPHAZ, 2017). I.e., susceptibility can be considered as probability of occurrence and is one factor of hazard. It is determined by conditioning factors (=inherent and more or less static factors) on the one hand, and triggering factors (=factors that directly initiate an outburst) on the other. The factors (predictors) analyzed in this paper, are limited (for good reasons!) to conditioning factors. In other words, the result of an analyses based on the parameters used in the present study, is mainly a lake stability assessment. In contrast to this, most of the mentioned regional glacial lake assessment approaches with more expert based, and probably subjective, parameter weightings, follow a hazard assessment approach, rather than a stability/susceptibility assessment. The reasons why these other studies consider factors like lake area or volume, or regional glacier mass balance, is not mainly because these factors directly influence lake stability, but because they have an impact on hazard potentials. Larger lake volumes (area is often used as a proxy for volume) and lake growth imply higher potential flood volumes, and therefore increase the hazard due to higher intensities, without affecting GLOF susceptibility. For similar reasons glacier masse balance is included in such models: Negative regional mass balances lead to glacier retreat and the formation of new and growth of existing lakes. Both processes increase the GLOF hazard potential in a region, but only have minor effects on GLOF susceptibility of individual lakes.

Further, unfavorable conditioning factors do not lead to a lake outburst immediately. It of course increases GLOF susceptibility, but requires still a triggering event to initiate an outburst. Clague and Evans (2000) and Emmer et al. (2020) present concepts about the timing of the causal chain of climate change, glacier retreat, glacial lake formation, and glacial lake outburst and conclude, based on empirical data from British Columbia and the Cordillera Blanca, that there is a lag between lake formation and outburst of up to several decades. The fact that a lake did not have an outburst event in the periods investigated in this study, does not automatically imply that the lake has a low GLOF susceptibility. It is indeed possible, that the lake is actually unstable (i.e. has a high susceptibility) but an outburst simply has not been triggered (yet).

Used data and parameters: Data availability for the entire study region is of course an important criterion for the selection of predicting parameters. But in addition to the parameters investigated in this study, there are candidates for other parameters which

are often and successfully applied in other regional assessment approaches cited in the study, such as the Steep Lakefront Area (SLA) developed by Fujita et al. (2013) and used by Rounce et al. (2016), or the topographic potential for rock or ice avalanches (cf. Allen et al., 2019), one of the most frequent GLOF triggers in High Mountain Asia. Considering this, I suggest to include more details about the selection of the predicting parameters.

Then, the influence of overlapping time periods of the different data sets used should be discussed in more detail, as also mentioned in the review of A. Emmer (Emmer, 2021). In particular the fact that the lake area change period overlaps the period which is investigated for GLOF occurrence, in my view disqualifies this parameter to be considered, as actually discussed in L340-344.

Statistical significance: Bayesian approaches are certainly most suitable for this type of research question where a large number of lakes (3,390) had relatively few (31) GLOF events. But still this is a very limited data basis, in particular since for the Forecasting, the Glacier-mass balance, and the Monsoonality models, only 11 GLOF events were recorded in the relevant 2005 to 2018 period. Even more, these 11 events are split over four to seven groups, depending on elevation, region, or monsoonality. Over the western half of the study region, only 3 GLOFs are found. This leads to very few (often only 1 or 2) or even zero GLOF events per subgroup (cf. boxes for Hindu Kush, Karakoram and Western Himalayas in Fig. 7). I wonder, how any predictor weights can be found in these cases. A very recent study from Zheng et al. (2021) on a slightly larger study region found evidence for a total of 215 GLOF events that presumable have happened since 1900, 176 thereof so far unreported. This does not contradict any of the data used here, but offers at least a potential alternative of a database with much more GLOF evidences (in turn posing challenges on the predictor data of course).

DETAILLED COMMENTS:

L11: I suggest to include something like 'regional-scale' (hazard estimations), because

at the level of individual lakes, there are many quantitative assessments available, including numerical modeling, geophysical measurements etc.

L21: Maybe change 'with respect to' to 'compared to'?

L81: Indicate the version number of the RGI

L112: Hydrostatic pressure acting on the dam depends mainly on lake depth, not area.

L263: The statement that upstream catchment area is well correlated with lake area is not clear to me. This needs further explanations of references. Also I do not understand why lake area is replaced by upstream catchment area in these model (Glacier-mass balance and Monsoonality), but not in others. This requires some more explanation.

L285: In the Forecasting and Glacier-mass balance models, A* represents lake-area change between 2005 and 2018. Is A* here also referring to this period (and not 1990 – 2018, as written)? If so, please correct, if not, another symbol should be used ($\triangle$A?).

L321/Fig. 9: Why are the log-odds ratios negative for the first (few) lakes? Would be interesting to describe in the text.

L323/324 (Caption Fig. 9): '. . .in the past four decades' applies only to the lakes in the x-axes of (a) and (c), for the other panels it's 2005-2018. I suggest to replace this with 'in the period 1981 – 2018 (a and c) and 2005 – 2018 (b and d-h). (Or change panel letters, see suggestion below).

Figures and Tables: Fig. 1: According to the caption, white triangles represent GLOFs since 1935. But as the study only deals with GLOFs that have occurred on the periods 1981-2018 and 2005-2018, respectively, only these should be shown here. Preferably with two colors, one for 1981-2005 and another for 2005-2018 to discriminate these to reference data sets. Please also indicate the spacing of the lake bubbles.

Table 1: This is a pretty large table for only presenting the 6 predictor parameter selected for this study. I suggest to present the 6 parameters used here in separate table,

giving some more details as well. (By the way, I think dam type could be ticked as well, at least a tick in brackets. As only moraine-dammed lakes are investigated here, this criteria is inherently considered). If the authors wish to keep having a table with other potentially relevant parameters for GLOF hazard assessment, this could be done in a more compressed format. But in this case, further geotechnical and geomorphic parameters would need to be included, such as permafrost conditions, lithology, seismicity, etc. The annex tables of the GAPHAZ guidelines (GAPHAZ, 2017) might give some indications for this.

Fig. 3: Blue-green combinations are hardly readable in the bubbles. I can see it in the text, slightly see it in the middle ('mulit-level') bubble, but do not see any green in the right ('many models'). Colors to be adjusted.

Figure 4: I suggest to sort sub-groups from highest (top) to lowest (bottom) in (a) and (b), West to East (or East-West) in (c) and highest monsoonality on top to lowest monsoonality in (d). (Same as ordering in Figs. 5-8).

Figs. 5-8 (general): In none of the figures I can see the middle (blueish) line. I only see the purple and orange lines. Similar for the color shades, I guess I only see the purple and the orange and the overlap of the two. Is this middle line represented in the Figures? If so, please adjust coloring, if not please add (or remove from the legend). For all figures it would be very nice to also have a panel for the pooled data, similar to Fig. 4.

Figs. 5 and 6: It would be helpful to indicated elevation bands in m a.s.l.

Fig. 9: To me it would make more sense to number the TP a-d and the TN e-h.

Fig. 10: In the legend (e) (the letter e is not needed in my view) swap ordering, that a is on top and d at the bottom, as in the main panels. Ad '%' to the numbers at the bottom. In the panels it would be helpful to include the locations with a recorded GLOF (for 1990-2018 in (a) and 2005-2018 in (b), (c) and (d)).

FINAL REMARKS:

I am well aware that several of the comments above are difficult to consider for the data analyses. However, I hope these aspects can be reflected somehow in the manuscript, either in the discussion or in the introduction, when describing the scope of the study. As a concluding remark I want to repeat my evaluation of this study, made under the general comments in the beginning, that I think this is an innovative approach and a needed analysis. I encourage the authors to revise this manuscript accordingly, and I am sure this paper will make an important contribution to the field of glacial lake hazard assessments.

REFERENCES CITED IN THIS REVIEW:

Allen, S. K., G. Zhang, W. Wang, T. Yao, and T. Bolch (2019), Potentially dangerous glacial lakes across the Tibetan Plateau revealed using a large-scale automated assessment approach, Science Bulletin, 64, 435–445, doi:10.1016/j.scib.2019.03.011.

Clague, J., and S. Evans (2000), A review of catastrophic drainage of moraine-dammed lakes in British Columbia, Quarternary Science Reviews, 19, 1763–1783.

Emmer, A., S. Harrison, M. Mergili, S. Allen, H. Frey, and C. Huggel (2020), 70 years of lake evolution and glacial lake outburst floods in the Cordillera Blanca (Peru) and implications for the future, Geomorphology, 365, 107178, doi:10.1016/j.geomorph.2020.107178.

Emmer, A. (2021), Interactive comment on "Controls of outbursts of moraine-dammed lakes in the greater Himalayan region" by Melanie Fischer et al., The Cryosphere Discussions, https://doi.org/10.5194/tc-2020-327-RC1,

Fujita, K., A. Sakai, S. Takenaka, T. Nuimura, A. B. Surazakov, T. Sawagaki, and T. Yamanokuchi (2013), Potential flood volume of Himalayan glacial lakes, Natural Hazards and Earth System Science, 13(7), 1827–1839, doi:10.5194/nhess-13-1827-2013.

GAPHAZ (2017), Assessment of Glacier and Permafrost Hazards in Mountain Regions, edited by S. K. Allen, H. Frey, and C. Huggel, Joint Standing Group on Glacier and Permafrost Hazards in High Mountains (GAPHAZ), Zurich, Lima.

Rounce, D. R., D. C. McKinney, J. M. Lala, A. C. Byers, and C. S. Watson (2016), A new remote hazard and risk assessment framework for glacial lakes in the Nepal Himalaya, Hydrology and Earth System Sciences, 20(9), 3455–3475, doi:10.5194/hess-20-3455-2016.

Zheng, G., A. Bao, S. Allen, J. Antonio Ballesteros-Cánovas, Y. Yuan, G. Jiapaer, M.Stoffel, (2021), Numerous unreported glacial lake outburst floods in the Third Pole revealed by high-resolution satellitedata and geomorphological evidence, Science Bulletin (online first), doi: https://doi.org/10.1016/j.scib.2021.01.014

---

## Author Comment (AC1) · 16 Mar 2021

Please see the supplementary PDF-document for a formatted version (incl. Table R1) of this Reply Letter.

General comment #1:

The authors present interesting results, some of which are novel in a sense that contradict assumptions of previous GLOF hazard assessment studies (e.g. the assumption that fast-growing lakes are more susceptible to GLOF), but this is only one part of the story (so far pretty much model-oriented) in my opinion. If the overall aim is enhanced identiinĂcation of potential future GLOF sites or so, stronger linkages of investigated

GLOF indicators to physical processes behind as well as (at least brief) characterization of documented GLOFs (in terms of triggers, mechanisms) are missing. For instance, how (process-wise) is the EDW, glacier-mass balance or lake (catchment) area linked to documented GLOF? What are triggers of historic GLOFs considered in this study? In fact, I'd expect this to be taken into consideration in the very first step – selection and justification of GLOF indicators.

Reply General comment #1:

We appreciate the reviewer's comment and elaborated in detail the choice of our predictors in a new table (now Table 2). We rewrote large parts of Section 2.1 to make clear how we selected each predictor. For example, we note that: "Larger and growing lakes offer more area for impacts from mass flows originating from adjacent valley slopes such as avalanches, rockfalls, and landslides (Haeberli et al., 2017)."; - "A larger upstream catchment area has been associated with an increased susceptibility to GLOFs as more runoff from intense precipitation, together with glacier and snow melt, can lead to sudden increases in lake volume (Allen et al., 2019; GAPHAZ, 2017; Worni et al., 2012). "; "These readily available data on regional glacier-mass balances are proxies for other, less accessible, physical controls on GLOF susceptibility such as glacial meltwater input, either directly from the parent glacier or from glaciers upstream, as well as permafrost decay in slopes fringing the lake."; "Meteorological drivers entered previous gualitative GLOF hazard appraisals mostly as (the probability of) extreme monsoonal precipitation events: the Kedarnath GLOF disaster, for example, was triggered by intense surface runoff (Huggel et al., 2004; Prakash and Nagarajan, 2017). [...] Elevated lake levels during the monsoon season also raise the hydrostatic pressure acting onto moraine dams (Richardson and Reynolds, 2000)." Background information on triggers is scant or conjectural for most GLOFs in our study region. We now offer a more thorough discussion on possible triggers: "The triggering mechanism of these studied GLOFs is reported in only seven cases, four of which are attributed to ice avalanches entering the lake (e.g. Tam Pokhari, Nepal or Kongyangmi La Tsho, India; Ives et al.,

TCD
2010; Nie et al., 2018). Other triggers of the GLOFs studied here include piping (Yindapu Co, China; Nie et al., 2018) and the collapse of an ice-cored moraine (Luggye Tsho, Bhutan; Fujita et al., 2008)." Please also see our response to general comment #2 in this regard. Contrary to the reviewer's notion, the goal of our study is not to "identify potential future GLOF sites or so". Our goal is to explore possible predictors of historic GLOFs. We had stressed this issue in the original Abstract: "We use an inventory of 3,390 moraine-dammed lakes and their documented outburst history in the past four decades to test whether elevation, lake area and its rate of change, glacier-mass balance, and monsoonality are useful inputs to a probabilistic classification model"; and: "We find that mostly larger lakes have been more prone to GLOFs in the past four decades, regardless of elevation band in which they occurred", and in many other locations in the manuscript.

General comment #2:

It would be interesting at least discuss how many of documented GLOFs were actually triggered by processes associated with investigated GLOF indicators? This is briefly touched in the introduction (L36-39) or study area section (L108), but I'm convinced that bit deeper and more comprehensive elaboration (e.g. a separate discussion section) would be beneficial for readers. Another example - on L244-245 it is mentioned that 'greater lakes are more likely to having had a GLOF ...'. I wonder what do primary data say about this – what proportion of these 31 GLOF-producing lakes would be classified as large at the time of GLOF and what this proportion is in the population of 3,390 moraine-dammed lakes? And in the other way around - can a speciiňĄc combination of values of GLOF indicators infer about possible (likely) GLOF trigger and mechanism (if not known)?

Reply General comment #2:

The reviewer may acknowledge that the link between mechanistic processes, choice of statistical predictors, and the occurrence of GLOFs invites some interpretation. If we
had sufficient data to run adequately parameterised process-based, numerical models on past GLOFs, we could go beyond the outputs of statistical models of outburst susceptibility. Our motivation, however, is to choose predictors as proxies instead of physical parameters in a deterministic model. Each of these proxies subsumes various physical processes that might be relevant to producing GLOFs. How well these proxies can describe the presence or absence of a GLOF is what our probabilistic models. These models learn from the data directly and inform us about the suitability of our predictors to hindcast historic GLOFs. By design, the outputs are probabilities and less so physical triggers or mechanisms. Still, this probabilistic approach forms a cornerstone in modern hazard and risk analyses. In this context, we are unsure what the reviewer means by "primary data". Our model learns from all the data as stated in the Methods section. We also note that the role of lake area in our model is that of a continuous predictor and not that of a response variable, as the reviewer seems to suspect. The model summarises how the susceptibility to GLOFs changes with lake area rather than vice versa.

General comment #3:

Let me also critically comment on some of the selected GLOF susceptibility indicators (in general, I'm convinced it would be useful presenting these indicators in a separate table with more detailed and comprehensive description than stated in the overview Tab. 1, and in places of the text):

Reply General comment #3:

We appreciate this suggestion and split Table 1 into two new tables: Table 2 now provides more detail on, and motivation for, our selected predictors.

General comment #3 (cont.):

Lake area change: I'm aware this indicator is always tricky to deïňĄne and employ; according to what is written on L134-135, two intervals are used for lake are

TCD
change (1990-2005 and 2005-2018); considering GLOFs occurring throughout the period 1981-2017, it means than these intervals may be pre-GLOF, post-GLOF or the GLOF occurred somewhen during one of these intervals – please comment on how this inconsistency was treated and whether it can explain that no link was observed between lake area change and the occurrence of GLOF.

Reply General comment #3 (cont.):

The data on lake-area changes are not yet resolved on an annual basis for our study area, so that we had to resort to changes averaged over longer periods. However, we used our forecasting model to test whether changes in lake size between two observation periods had a credible effect on PGLOF. Here we explored the weight of relative changes in lake area between 1990 and 2005 to estimate the probability of observing GLOFs that happened in the subsequent period 2005-2018. In other words, we trained the model on GLOF data that pre-date the testing data, and thus offer a realistic and rigorous prediction and validation scenario. We reported in our original manuscript that "The weight of relative lake-area change in the 15 years before is ambiguous ( $\beta A^* =$ -0.04+0.76/-0.67) [...]" (L245-246) and that "In the forecasting model, however, the influence of lake-area change remains negligible even for <50% HDIs." (L350). This indicates with 95% probability that relative lake-area change before the outburst is an inconclusive predictor. To further stress this result, we added the following sentences in the Discussion: "However, in the forecasting model, in which we tested whether differing data observation periods have any credible effects, the influence of lake-area change remains negligible even for <50% HDIs. We thus conclude that relative lakearea change before outburst is an inconclusive predictor. This result contradicts the assumptions made in many previous studies that assumed that rapidly growing lakes are the most prone to sudden outburst (Aggarwal et al., 2016; Bolch et al., 2011; Ives et al., 2010; Mergili and Schneider, 2011; Prakash and Nagarajan, 2017; Rounce et al., 2016; Wang et al., 2012)."

General comment #3 (cont.):

**TCD**
Glacier mass balance: similarly to my comment on lake area change - how can 2000-2016 glacier mass balance be used to explain GLOFs occurring throughout the period 1981-2017? These characteristics (mass balance as well as lake area change) are dynamic in nature and I'm wondering how can a static information from available datasets possibly blur a GLOF signal, especially for pre-2000 GLOFs?

Reply General comment #3 (cont.):

The problem of limited data for lake-area changes is even more pronounced for glaciermass balances in our study area. Again, the data averaged from 2000 to 2016 are among the few regionally consistent data sets that we considered as input for our model. The underlying assumption is that the regional regime of prevalent glacier melting from 2000 to 2016 largely follows a trend dating back to the late 1980s. This is in line with the review of Bolch et al. (2019) who summarized that "glaciers [in High Mountain Asia] have thinned, retreated, and lost mass since the 1970s, except for parts of the Karakoram, eastern Pamir, and western Kunlun" (p. 211). To answer the reviewer's question of "[...] how can a static information from available datasets possibly blur a GLOF signal, especially for pre-2000 GLOFs?", we curtailed our glacier-mass balance model only to those lakes that had outbursts after 2000, and hence that overlap with the study period of Brun et al. (2017). Table R1, which is included in the PDF-version of this Reply Letter, shows the output from this model. We find that: - The parameter estimates at the population level changed only marginally: the weight of catchment area ( $\beta$ C) remains credibly positive and that of lake-area change from 2005 to 2018  $(\beta (A^{(*b)}))$  remains credibly negative. - At the group level, the standard deviation of intercepts of our grouping variable elevation ( $\sigma z$ ) is also comparable to our previous results. - Posterior estimates of  $\sigma r$ , the standard deviation of group level intercepts of glacier-mass balance regions, increase from 0.81+1.60/-0.78 to 1.11+1.77/-1.03, though with much overlap. This further underlines our finding that the glacier-mass balance in a given region credibly affects PGLOF. We now highlight these findings: "On the basis of higher standard deviations, we learn that effects of glaciological regions

TCD
vary more than those of elevation bands ( $\sigma r = 0.81+1.60/-0.78$  and  $\sigma z = 0.48+1.19/-0.47$ ). When training this model on a subset of glacial lakes with documented GLOFs post-2000 (i.e. including only GLOFs which occurred in the time period covered by our glacier-mass balance data), posterior estimates of  $\sigma r$  increase to 1.11+1.77/-1.03, further underlining our result that glacier-mass balance credibly affects PGLOF."

General comment #3 (cont.):

Monsoonality: using climate indicators in GLOF research is promising, but proportion of summer precipitation doesn't tell you about the extremity; for instance, the proportion will be lower in areas where extreme rainfalls occur in summer, but also some precipitation in winter, but will be super-high in generally dry areas with some precipitation during the summer and no precipitation in winter. But process-wise, the first area will have much higher potential to trigger GLOF in my opinion.

Reply General comment #3 (cont.):

We are unsure about whether the reviewer offers an opinion here or whether this statement is supported by data. Our analysis shows that the proportion of summer precipitation is highest in areas with strong monsoonal influence (Fig. 1). We are unaware of any GLOFs that occurred in winter. For example, ice cover on lakes and freezing moraine dams have been thought to make glacier lakes resilient against outbursts, even during strong seismic shaking (Kargel et al., 2015). Most of the heavy rainstorms are tied to the South Asian summer monsoon, and some reported GLOFs were triggered by such storms (Allen et al., 2016; Liu et al., 2014). The drier areas of our study area usually receive higher amounts of precipitation during winter via westerlies (Bolch et al., 2012; Bookhagen and Burbank, 2010), so we think that monsoonality remains a useful predictor.

Detail comment #1:

L11: yes, the approach is quantitative, but selection of GLOF indicators in this study is
also expert judgement-based as the authors are GLOF experts

Reply Detail comment #1:

We rephrased this sentence to "Estimating regional susceptibility of glacial lakes has largely relied on qualitative assessments by experts, thus motivating a more systematic and quantitative appraisal."

Detail comment #2:

L34: see also Cook et al., 2018, Science

Reply Detail comment #2:

We thank the reviewer for suggesting this useful reference, which we added to our manuscript.

Detail comment #3:

L36-37: this needs deeper elaboration in relation to selected GLOF susceptibility indicators (see also my general comment)

Reply Detail comment #3:

We refer the reviewer to our reply on General Comment #3.

Detail comment #4:

L103: I suggest to use 'GLOF susceptibility indicators' instead of 'diagnostics of GLOF potential' or 'diagnostics of GLOF hazard' (L125); similarly, 'controls' and 'predictors' are used throughout the manuscript, please define a difference or unify

Reply Detail comment #4:

We decided to consistently use the term "predictor", in line with the common terminology in multivariate statistics. Our use of "diagnostic" is also appropriate, as the regression model has a bivariate outcome. Yet to make things more clear, we replaced Interactive comment

"diagnostic" by "predictor". We avoid the term "indicator", as it may be confusing in the model context. In regression models, an indicator variable is often a logical binary [0, 1] or dummy variable, whereas we mostly use continuous variables.

Detail comment #5:

L111-112: lake deepening increases hydrostatic pressure, not areal or volumetric growth

Reply Detail comment #5:

This is physically more correct, though we fail to see how any change in lake area or volume could not affect hydrostatic pressure eventually. To be more clear, we rewrote our statement to: "Lake area scales with lake volume and depth (Huggel et al. 2002), and growing lake depths increase the hydrostatic pressure on moraine dams, thus raising the potential of failure (Rounce et al., 2016)."

Detail comment #6:

L115-116: the authors usually argue that larger lakes are more susceptible because large lake areas are more exposed to slope movements potentially triggering GLOFs; large area is also correlated with larger depth (and so hydrostatic pressure acting on a dam)

Reply Detail comment #6:

We added this reasoning to the text: "Larger and growing lakes offer more area for impacts from mass movements originating from adjacent valley slopes such as avalanches, rockfalls, and landslides (Haeberli et al., 2017)."

Detail comment #7:

L130: how is different date of GLOF and input data for model treated? (how possibly different environmental conditions at the time of GLOF and at the time of datasets acquisition can influence your results?) see also my general comments

TCD
Reply Detail comment #7:

We used the reported dates of historic GLOFs where available. We acknowledge that our predictor variables can only approximate the environmental conditions at the time of lake outburst, but this is the point of a statistical predictor in a data-driven model. Please see our replies to General Comments 1-3 in this regard.

Detail comment #8:

L136-139: I suggest to move this to L133

Reply Detail comment #8:

We moved and slightly rephrased this text passage accordingly.

Detail comment #9:

L166: delete 's'

Reply Detail comment #9:

Deleted accordingly.

Detail comment #10:

L176: delete ','

Reply Detail comment #10:

Deleted accordingly.

Detail comment #11:

L207: what is meant by 'common susceptibility'?

Reply Detail comment #11:

For clarification, we rephrased this sentence to "In essence, this varying-intercept model acknowledges that glacial lakes in the same elevation band may have had a
common baseline susceptibility to GLOFs in the past four decades."

Detail comment #12:

L262-263: this step is not clear to me? Please explain

Reply Detail comment #12:

To clarify our approach, we added more details on our predictor catchment area in the new Table 2. We explain our choice of this predictor and why we use it instead of the static lake area A in the glacier-mass balance and monsoonality models: "We also tested the impact of upstream catchment area C ( $m^2$ ) on GLOF susceptibility. A larger upstream catchment area has been associated with an increased susceptibility to GLOFs as runoff from intense precipitation as well as glacier and snow melt can lead to sudden increases in lake volume (Allen et al., 2019; GAPHAZ, 2017)."

Detail comment #13:

L263: please provide details about this correlation

Reply Detail comment #13:

We find that catchment area C has a strong linear correlation with lake area A (Pearson's correlation coefficient of 0.446), such that we preferred C over A in two of our models, as C is constant at the scale of our study.

Detail comment #14:

L272: what is meant by 'average lake'?

Reply Detail comment #14:

The average lake is defined by the combination of all average predictor values.

Detail comment #15:

L352-354: this can be true for a specific period in long-term evolution of a mountain
range (considering gradual glacier retreat and overall shift of all rapid processes including GLOFs to higher elevation zones; i.e. the general shift of morphoclimatic zones)

Reply Detail comment #15:

The point we wanted to make here was that stratifying by elevation hardly helped to inform us more about GLOF susceptibility in this context. In essence, GLOF susceptibility is aptly represented by the pooled model here.

Detail comment #16:

L365: so why not to consider this indicator in your model?

Reply Detail comment #16:

We ran a number of models that used the distance from the parent glacier as a predictor, though obtained no credible posterior weights. However, we found that this distance is most likely the most prone to highly dynamic changes in historic times. We thus added to the text: "However, systematically recorded time series of glacier fronts are even harder to come by when compared to systematic measurements of changes in glacier-lake areas."

Detail comment #17:

L373: are 'minute'? Please check

Reply Detail comment #17:

We clarified this: "[...] deviations from a pooled model for the HKKHN are minute when compared to the other models' spread of posterior group-level intercepts (Fig. 4)."

Detail comment #18:

Tab 2: what is PDF?

Reply Detail comment #18:

TCD
The abbreviation PDF stands for probability density function. We changed the column header in Table 3 (former Table 2) accordingly.

Detail comment #19:

Tab. 4: please also consider presenting false positives and false negatives

Reply Detail comment #19:

We accordingly added false positives and false negatives in a separate column to the table.

Detail comment #20:

Fig. 2: three lake inventories are mentioned (ICIMOD, Veh et al., 2019 and Wang et al., 2020); please make clear how these were integrated; these 3,390 lakes (L131) are from which inventory?

Reply Detail comment #20:

The 3,390 lakes forming our database are a subset of the ICIMOD inventory published by Maharjan et al. (2018). We changed Figure 2 to better show this. We also clarified that "Second, we identified from an independent regional GLOF inventory (Veh et al. 2019) 31 lakes that had at least one outburst between 1981 and 2017 and are listed in the ICIMOD inventory."

Detail comment #21:

Fig. 3: how about green color in Many Models part?

Reply Detail comment #21:

We modified our figure to clarify this and also changed the colour scheme (as requested by referee #2) for improved contrasts.

Detail comment #22:
Fig. 10: please consider highlighting GLOF-producing lakes; switch a-d in the panel (e)

Reply Detail comment #22:

We modified the figure accordingly. Č

References cited in this Reply

Aggarwal, A., Jain, S. K., Lohani, A. K. and Jain, N.: Glacial lake outburst flood risk assessment using combined approaches of remote sensing, GIS and dam break modelling, Geomatics, Nat. Hazards Risk, 7(1), 18–36, doi:10.1080/19475705.2013.862573, 2016.

Allen, S. K., Rastner, P., Arora, M., Huggel, C. and Stoffel, M.: Lake outburst and debris flow disaster at Kedarnath, June 2013: hydrometeorological triggering and topographic predisposition, Landslides, 13(6), 1479–1491, doi:10.1007/s10346-015-0584-3, 2016.

Allen, S. K., Zhang, G., Wang, W., Yao, T. and Bolch, T.: Potentially dangerous glacial lakes across the Tibetan Plateau revealed using a large-scale automated assessment approach, Sci. Bull., (April), doi:10.1016/j.scib.2019.03.011, 2019.

Bolch, T., Peters, J., Yegorov, A., Pradhan, B., Buchroithner, M. and Blagoveshchensky, V.: Identification of potentially dangerous glacial lakes in the northern Tien Shan, Nat. Hazards, 59(3), 1691–1714, doi:10.1007/s11069-011-9860-2, 2011.

Bolch, T., Kulkarni, A., Kääb, A., Huggel, C., Paul, F., Cogley, J. G., Frey, H., Kargel, J. S., Fujita, K., Scheel, M., Bajracharya, S. and Stoffel, M.: The state and fate of himalayan glaciers, Science (80-. )., 336(6079), 310–314, doi:10.1126/science.1215828, 2012.

Bolch, T., Shea, J. M., Liu, S., Azam, F. M., Gao, Y., Gruber, S., Immerzeel, W. W., Kulkarni, A., Li, H., Tahir, A. A., Zhang, G. and Zhang, Y.: Status and Change of the Cryosphere in the Extended Hindu Kush Himalaya Region, in The Hindu Kush Hi-
malaya Assessment: Mountains, Climate Change, Sustainability and People, edited by P. Wester, A. Mishra, A. Mukherji, and A. B. Shrestha, pp. 209–255, Springer International Publishing, Cham., 2019.

Bookhagen, B. and Burbank, D. W.: Toward a complete Himalayan hydrological budget: Spatiotemporal distribution of snowmelt and rainfall and their impact on river discharge, J. Geophys. Res. Earth Surf., 115(3), 1–25, doi:10.1029/2009JF001426, 2010.

Fujita, K., Suzuki, R., Nuimura, T. and Sakai, A.: Performance of ASTER and SRTM DEMs, and their potential for assessing glacial lakes in the Lunana region, Bhutan Himalaya, J. Glaciol., 54(185), 220–228, doi:10.3189/002214308784886162, 2008.

GAPHAZ: Assessment of Glacier and Permafrost Hazards in Mountain Regions: technical Guidance Document. Standing Group on Glacier and Permafrost Hazards in Mountains (GAPHAZ) of the International Association of Cryospheric Sciences (IACS) and the International Per, Zurich, Lima., 2017.

Haeberli, W., Schaub, Y. and Huggel, C.: Increasing risks related to landslides from degrading permafrost into new lakes in de-glaciating mountain ranges, Geomorphology, 293, 405–417, doi:https://doi.org/10.1016/j.geomorph.2016.02.009, 2017.

Huggel, C., Kääb, A., Haeberli, W., Teysseire, P. and Paul, F.: Remote sensing based assessment of hazards from glacier lake outbursts: a case study in the Swiss Alps, Can. Geotech. J., 39(2), 316–330, doi:10.1139/t01-099, 2002.

Huggel, C., Haeberli, W., Kääb, A., Bieri, D. and Richardson, S.: An assessment procedure for glacial hazards in the Swiss Alps, Can. Geotech. J., 41(6), 1068–1083, doi:10.1139/T04-053, 2004.

Ives, J. D., Shrestha, R. B. and Mool, P. K.: Formation of Glacial Lakes in the Hindu Kush-Himalayas and GLOF Risk Assessment, International Centre for Integrated Mountain Development (ICIMOD), Kathmandu., 2010.

Kargel, J. S., Leonard, G. J., Shugar, D. H., Haritashya, U. K., Bevington, A. and Field-

TCD
ing, E. J.: Geomorphic and geologic controls of geohazards induced by Nepal's 2015 Gorkha earthquake., Science, 3(1), 1–10, doi:10.1126/science.aac8353, 2015.

Liu, J. J., Cheng, Z. L. and Su, P. C.: The relationship between air temperature fluctuation and Glacial Lake Outburst Floods in Tibet, China, Quat. Int., 321, 78–87, doi:10.1016/j.quaint.2013.11.023, 2014.

Maharjan, S. B., Mool, P. K., Lizong, W., Xiao, G., Shrestha, F., Shrestha, R. B., Khanal, N. R., Bajracharya, S. R., Joshi, S., Shai, S. and Baral, P.: The Status of Glacial Lakes in the Hindu Kush Himalaya, International Centre for Integrated Mountain Development (ICIMOD), Kathmandu., 2018.

Mergili, M. and Schneider, J. F.: Regional-scale analysis of lake outburst hazards in the southwestern Pamir, Tajikistan, based on remote sensing and GIS, Nat. Hazards Earth Syst. Sci., 11(5), 1447–1462, doi:10.5194/nhess-11-1447-2011, 2011.

Nie, Y., Liu, Q., Wang, J., Zhang, Y., Sheng, Y. and Liu, S.: An inventory of historical glacial lake outburst floods in the Himalayas based on remote sensing observations and geomorphological analysis, Geomorphology, 308(December), 91–106, doi:10.1016/j.geomorph.2018.02.002, 2018.

Prakash, C. and Nagarajan, R.: Outburst susceptibility assessment of morainedammed lakes in Western Himalaya using an analytic hierarchy process, Earth Surf. Process. Landforms, 42(14), 2306–2321, doi:10.1002/esp.4185, 2017.

Richardson, S. D. and Reynolds, J. M.: An overview of glacial hazards in the Himalayas, in Quaternary International, vol. 65–66, pp. 31–47., 2000.

Rounce, D. R., McKinney, D. C., Lala, J. M., Byers, A. C. and Watson, C. S.: A new remote hazard and risk assessment framework for glacial lakes in the Nepal Himalaya, Hydrol. Earth Syst. Sci., 20(9), 3455–3475, doi:10.5194/hess-20-3455-2016, 2016.

Veh, G., Korup, O., Specht, S., Roessner, S. and Walz, A.: Unchanged frequency of moraine-dammed glacial lake outburst floods in the Himalaya, Nat. Clim. Chang.,
2000, 1-5, doi:10.1038/s41558-019-0437-5, 2019.

Wang, W., Yao, T., Gao, Y., Yang, X. and Kattel, D. B.: A First-order Method to Identify Potentially Dangerous Glacial Lakes in a Region of the Southeastern Tibetan Plateau, Mt. Res. Dev., 31(2), 122–130, doi:10.1659/MRD-JOURNAL-D-10-00059.1, 2011.

Wang, X., Liu, S., Ding, Y., Guo, W., Jiang, Z., Lin, J. and Han, Y.: An approach for estimating the breach probabilities of moraine-dammed lakes in the Chinese Himalayas using remote-sensing data, Nat. Hazards Earth Syst. Sci., 12(10), 3109– 3122, doi:10.5194/nhess-12-3109-2012, 2012.

Worni, R., Stoffel, M., Huggel, C., Volz, C., Casteller, A. and Luckman, B.: Analysis and dynamic modeling of a moraine failure and glacier lake outburst flood at Ventisquero Negro, Patagonian Andes (Argentina), J. Hydrol., 444–445, 134–145, doi:10.1016/j.jhydrol.2012.04.013, 2012.

Worni, R., Huggel, C. and Stoffel, M.: Glacial lakes in the Indian Himalayas - From an area-wide glacial lake inventory to on-site and modeling based risk assessment of critical glacial lakes, Sci. Total Environ., 468–469, S71–S84, doi:10.1016/j.scitotenv.2012.11.043, 2013.

Please also note the supplement to this comment: https://tc.copernicus.org/preprints/tc-2020-327/tc-2020-327-AC1-supplement.pdf

TCD

---

## Author Comment (AC2) · 16 Mar 2021

Please see the supplementary PDF-document for a formatted version of this Reply Letter.

General comment #1:

Hazard concept: The article is strictly focusing on GLOF susceptibility and using this term consistently throughout the manuscript. Nevertheless, I think regarding some aspects of the study, concepts and terminologies are mixed at some places. According to international standards from UNISDR, IPCC etc., hazard is a function of probability (of occurrence) and intensity (or magnitude). Susceptibility in turn 'is a relative measure

of the likelihood (or probability) that a hazard will occur or initiate from a given site, based on intrinsic properties and dynamic characteristics of that site' and 'has an inverse relationship with stability' (GAPHAZ, 2017). I.e., susceptibility can be considered as probability of occurrence and is one factor of hazard. It is determined by conditioning factors (=inherent and more or less static factors) on the one hand, and triggering factors (=factors that directly initiate an outburst) on the other. The factors (predictors) analyzed in this paper, are limited (for good reasons!) to conditioning factors. In other words, the result of an analyses based on the parameters used in the present study, is mainly a lake stability assessment.

Reply General comment #1:

We thank the reviewer for this observation and these definitions. However, we would have hoped for some guidance as to where exactly we might have mixed concepts and terminologies in this regard. The specific comments below seem not to pick up this issue. Perhaps some of the confusion arises from both qualitative and quantitative uses of the term "hazard". We echo the reviewer's comments on hazard and susceptibility in principle. Yet we wish to stress that our model is far from a stability assessment of moraine dams. The reviewer may concede that such appraisals frequently hinge on hard classes such as "stable" or "unstable". Even if such an appraisal would be (geo-)technically feasible, we would have needed to collate many more parameters on the internal structure and geometry of moraine dams, including grain-size distribution, presence and size of ice cores, the volume, width, height, and slope of moraine dams, pore water pressure in the dam, armouring of the outlet channel, presence and opening of tension cracks, rates of subsidence, and many others. Such parameters have been likely prone to change during our study period and are difficult to obtain for a single lake, and so even less feasible for the size of our regional study. To avoid an elusive feeling of stability, and hence, safety, we refrain to call our approach a dam-stability assessment. Strictly speaking, our analysis estimates the probability of correctly detecting historic lake outbursts from a set of predictors. The referee may acknowledge

that this probability is indeed a likelihood of GLOF outburst conditioned on reporting. In this sense one could see this metric as a "relative measure of the likelihood (or probability) that a hazard will occur or initiate from a given site" like the referee suggests. Our forecasting model in particular addresses this scenario.

General comment #1 (cont.):

In contrast to this, most of the mentioned regional glacial lake assessment approaches with more expert based, and probably subjective, parameter weightings, follow a hazard assessment approach, rather than a stability/susceptibility assessment.

Reply General comment #1 (cont.):

From the literature that we compiled in this and our previous work on GLOFs, we infer that very few, if any, of these so-called hazard assessments offer probabilistic metrics that satisfy the formal quantitative definition of hazard, as the referee clearly points out. During our review, we found that most of these studies deal with hazard in a qualitative or semi-quantitative way.

General comment #1 (cont.):

The reasons why these other studies consider factors like lake area or volume, or regional glacier mass balance, is not mainly because these factors directly influence lake stability, but because they have an impact on hazard potentials. Larger lake volumes (area is often used as a proxy for volume) and lake growth imply higher potential flood volumes, and therefore increase the hazard due to higher intensities, without affecting GLOF susceptibility. For similar reasons glacier masse balance is included in such models: Negative regional mass balances lead to glacier retreat and the formation of new and growth of existing lakes. Both processes increase the GLOF hazard potential in a region, but only have minor effects on GLOF susceptibility of individual lakes.

Reply General comment #1 (cont.):

We agree with the referee in principle here. Yet we found it difficult to trace objectively any increases (or even changes) in hazard in the literature due a distinct lack of the necessary probabilistic metrics.

General comment #1 (cont.):

Further, unfavorable conditioning factors do not lead to a lake outburst immediately. It of course increases GLOF susceptibility, but requires still a triggering event to initiate an outburst. Clague and Evans (2000) and Emmer et al. (2020) present concepts about the timing of the causal chain of climate change, glacier retreat, glacial lake formation, and glacial lake outburst and conclude, based on empirical data from British Columbia and the Cordillera Blanca, that there is a lag between lake formation and outburst of up to several decades. The fact that a lake did not have an outburst event in the periods investigated in this study, does not automatically imply that the lake has a low GLOF susceptibility. It is indeed possible, that the lake is actually unstable (i.e. has a high susceptibility) but an outburst simply has not been triggered (yet).

Reply General comment #1 (cont.):

We reiterate our point above and state that our method is set out to detect reported GLOFs. The reviewer's comment on dam stability is important and should be considered in geo-technical assessments, but is tangential to our objectives. Nowhere did we state that we wanted to quantify or estimate the stability of moraine dams. We also do acknowledge the concept of lag times between lake formation and outburst: every lake has a life span, but the question is whether an outburst needs to end it. This concept of lag time is thought provoking, but hinges on data similar to the models that we present here. One major advantage in our models is that we can fully capture the underlying uncertainties, something that we have so far yet to see for any lag-time model.

General comment #2:

Used data and parameters: Data availability for the entire study region is of course

an important criterion for the selection of predicting parameters. But in addition to the parameters investigated in this study, there are candidates for other parameters which are often and successfully applied in other regional assessment approaches cited in the study, such as the Steep Lake front Area (SLA) developed by Fujita et al. (2013) and used by Rounce et al. (2016), or the topographic potential for rock or ice avalanches (cf. Allen et al., 2019), one of the most frequent GLOF triggers in High Mountain Asia. Considering this, I suggest to include more details about the selection of the predicting parameters.

Reply General comment #2:

We wish to refer the reviewer to the changes that we have made following the suggestions of referee #1. We rewrote large parts of Section 2.1 to make clear how we selected each predictor. For example, we note that: - "Larger and growing lakes offer more area for impacts from mass flows originating from adjacent valley slopes such as avalanches, rockfalls, and landslides (Haeberli et al., 2017)."; - "A larger upstream catchment area has been associated with an increased susceptibility to GLOFs as more runoff from intense precipitation, together with glacier and snow melt, can lead to sudden increases in lake volume (Allen et al., 2019; GAPHAZ, 2017; Worni et al., 2012). "; - "These readily available data on regional glacier-mass balances are proxies for other, less accessible, physical controls on GLOF susceptibility such as glacial melt-water input, either directly from the parent glacier or from glaciers upstream, as well as permafrost decay in slopes fringing the lake."; - "Meteorological drivers entered previous qualitative GLOF hazard appraisals mostly as (the probability of) extreme monsoonal precipitation events: the Kedarnath GLOF disaster, for example, was triggered by intense surface runoff (Huggel et al., 2004; Prakash and Nagarajan, 2017). [. . .] Elevated lake levels during the monsoon season also raise the hydrostatic pressure acting onto moraine dams (Richardson and Reynolds, 2000)."

Fujita et al. (2013) used the SLA approach to derive Potential Flood Volumes (PFVs) of Himalayan lakes as a proxy for GLOF susceptibility. We assume that PFVs are

largely represented by our predictor lake area, given that larger lakes should produce larger floods. A major critique of the SLA concept is that lakes can have zero PFV despite large lake volumes. This issue was observed, for example, at Imja Lake in the Mt. Everest region, Nepal, that stores 78.4 × 106 m3 of water surrounded by steep slopes (Haritashya et al., 2018). Fujita et al. (2013) also point towards an issue that the reviewer had cautioned against above (p. 1834): "PFVs were simply calculated from the topography surrounding the moraine-dammed lakes and thus the robustness of the dam could not be evaluated. As the existence of ice within the damming moraine may alter the dam's vulnerability, understanding the distribution and degradation of permafrost will be an important factor for the further assessment of GLOF probability". Furthermore, SLA depends on user-defined cutoffs, for example a 1-km search radius for steep slopes around lakes (Fujita et al., 2013), or a minimum slope threshold (Rounce et al., 2016). Such thresholds introduce additional subjectivity that we wished to avoid in our appraisal. Finally, errors in digital elevation models in high mountains remain unaccounted for in these slope-based metrics, though have been acknowledged for many years (Fujita et al., 2008). For example, Mukul et al., (2017) reported that "vertical accuracy of the data decreases with increase in slope and elevation due to presence of large outliers and voids. Therefore, studies using SRTM data "as is", especially in regions like the Himalaya, are not statistically meaningful". In summary, these findings motivated us to keep the influence of potentially error-prone model inputs at a minimum.

General comment #2 (cont.):

Then, the influence of overlapping time periods of the different data sets used should be discussed in more detail, as also mentioned in the review of A. Emmer (Emmer, 2021). In particular the fact that the lake area change period overlaps the period which is investigated for GLOF occurrence, in my view disqualifies this parameter to be considered, as actually discussed in L340-344.

Reply General comment #2 (cont.):

We again wish to refer the referee to our reply to referee #1: The data on lake-area changes are not yet resolved on an annual basis for our study area, so that we had to resort to changes averaged over longer periods. However, we used our forecasting model to test whether changes in lake size between two observation periods had a credible effect on PGLOF. Here we explored the weight of relative changes in lake area between 1990 and 2005 to estimate the probability of observing GLOFs that happened in the subsequent period 2005-2018. In other words, we trained the model on GLOF data that pre-date the testing data, and thus offer a realistic and rigorous prediction and validation scenario. We reported in our original manuscript that "The weight of relative lake-area change in the 15 years before is ambiguous ($\beta$A* = $-0.04+0.76/-0.67$) [...]" (L245-246) and that "In the forecasting model, however, the influence of lake-area change remains negligible even for <50% HDIs." (L350). This indicates with 95% probability that relative lake-area change before the outburst is an inconclusive predictor. To further stress this result, we added the following sentences in the Discussion: "However, in the forecasting model, in which we tested whether differing data observation periods have any credible effects, the influence of lake-area change remains negligible even for <50% HDIs. We thus conclude that relative lake-area change before outburst is an inconclusive predictor. This result contradicts the assumptions made in many previous studies that assumed that rapidly growing lakes are the most prone to sudden outburst (Aggarwal et al., 2016; Bolch et al., 2011; Ives et al., 2010; Mergili and Schneider, 2011; Prakash and Nagarajan, 2017; Rounce et al., 2016; Wang et al., 2012)."

General comment #3:

Statistical significance: Bayesian approaches are certainly most suitable for this type of research question where a large number of lakes (3,390) had relatively few (31) GLOF events. But still this is a very limited data basis, in particular since for the Forecasting, the Glacier-mass balance, and the Monsoonality models, only 11 GLOF events were recorded in the relevant 2005 to 2018 period. Even more, these 11 events are split

over four to seven groups, depending on elevation, region, or monsoonality. Over the western half of the study region, only 3 GLOFs are found. This leads to very few (often only 1 or 2) or even zero GLOF events per subgroup (cf. boxes for Hindu Kush, Karakoram and Western Himalayas in Fig. 7). I wonder, how any predictor weights can be found in these cases.

Reply General comment #3:

The problem of highly imbalanced data (few GLOF reports out of several thousand lakes) was a major motivation for us to use Bayesian models. The low prior probability of detecting a reported GLOF can be compared directly with the posterior probability, a strategy that we showed in our original Fig. 9. Classical rare-events logistic regression penalises the model likelihood, and this step is done naturally via the prior distributions in the Bayesian setting. The low number of data points in some groups is even less of an issue in a hierarchical model, as this always draws strength across each group and the pooled model of all data taken together. The high posterior uncertainties tied to some model groups clearly underline the effect of fewer data points. To further underline these points, we added the following statement: "The small number of reported GLOFs introduces strong imbalance to our data, given that some regions, and hence levels, had few or no reported GLOFs. Although this would be problematic in most other modelling approaches, Bayesian multi-level models are particularly well suited for this kind of imbalanced training data (Gelman and Hill, 2007; Shor et al., 2007; Stegmueller, 2013)."

General comment #3 (cont.):

A very recent study from Zheng et al. (2021) on a slightly larger study region found evidence for a total of 215 GLOF events that presumable have happened since 1900, 176 thereof so far unreported. This does not contradict any of the data used here, but offers at least a potential alternative of a database with much more GLOF evidences (in turn posing challenges on the predictor data of course).

Reply General comment #3 (cont.):

We checked the study by Zheng et al. (2021) but found mostly GLOFs without times-tamps that are difficult to reconcile with our predictors, some of which are averaged over specified time periods. Please also see our reply to Referee #1 with respect to the validity of predictors that change over time.

Detail comment #1:

L11: I suggest to include something like 'regional-scale' (hazard estimations), because at the level of individual lakes, there are many quantitative assessments available, including numerical modeling, geophysical measurements etc.

Reply Detail comment #1:

We rephrased this sentence accordingly to: "Estimating regional susceptibility of glacial lakes [. . .]."

Detail comment #2:

L21: Maybe change 'with respect to' to 'compared to'?

Reply Detail comment #2:

We changed the phrasing as requested.

Detail comment #3:

L81: Indicate the version number of the RGI

Reply Detail comment #3:

We added the version number 6.0.

Detail comment #4:

L112: Hydrostatic pressure acting on the dam depends mainly on lake depth, not area.

Reply Detail comment #4:

We acknowledge that this is physically more correct, though we fail to see how any change in lake area or volume could not affect hydrostatic pressure eventually. We accordingly rephrased our statement to: "Lake area scales with lake volume and depth (Huggel et al., 2002), and growing lake depths increase the hydrostatic pressure acting on moraine dams, thus raising the potential of failure (Rounce et al., 2016)."

Detail comment #5:

L263: The statement that upstream catchment area is well correlated with lake area is not clear to me. This needs further explanations of references. Also I do not understand why lake area is replaced by upstream catchment area in these model (Glacier-mass balance and Monsoonality), but not in others. This requires some more explanation.

Reply Detail comment #5:

We thank the reviewer for this suggestion and added more details on our predictor catchment area. We refer to our replies to referee #1's detail comments #12 and #13: To clarify our approach, we added more details on our predictor catchment area in the new Table 2. We explain our choice of this predictor and why we use it instead of the static lake area A in the glacier-mass balance and monsoonality models: "We also tested the impact of upstream catchment area C ($m^2$) on GLOF susceptibility. A larger upstream catchment area has been associated with an increased susceptibility to GLOFs as runoff from intense precipitation as well as glacier and snow melt can lead to sudden increases in lake volume (Allen et al., 2019; GAPHAZ, 2017)." We find that catchment area C has a strong linear correlation with lake area A (Pearson's correlation coefficient of 0.446), such that we preferred C over A in two of our models, as C is constant at the scale of our study.

Detail comment #6:

L285: In the Forecasting and Glacier-mass balance models, A* represents lake-area

change between 2005 and 2018. Is A* here also referring to this period (and not 1990 – 2018, as written)? If so, please correct, if not, another symbol should be used (△A?).

Reply Detail comment #6:

Our predictor relative lake-area change A* (not to be confused with net lake-area change △A) is calculated for three different time windows: 1990 to 2005 in the forecasting model, 2005 to 2018 in the glacier-mass balance model, and 1990 to 2018 in the monsoonality model. In order to avoid confusion and correctly refer to each respective time interval of relative lake-area change, we now assigned each with its own symbol: relative lake-area change between 1990 to 2005 is A*a , relative lake-area change between 2005 and 2018 is A*b , and relative lake-area change between 1990 and 2018 is A*c. We explain this notation in our new Table 2 and in the respective model descriptions in the Results section 3.

Detail comment #7:

L321/Fig. 9: Why are the log-odds ratios negative for the first (few) lakes? Would be interesting to describe in the text.

Reply Detail comment #7:

The negative log-odds ratios indicate lakes for which the posterior probability of a reported GLOF is lower than the prior probability. To further clarify this, we rephrased this to: "A positive log-odds ratio means that we obtain a higher posterior probability of attributing a historic GLOF to a given lake compared to a random draw. Negative log-odd ratios indicate lakes for which the posterior probability of a reported GLOF is lower than the prior probability."

Detail comment #8:

L323/324 (Caption Fig. 9): '...in the past four decades' applies only to the lakes in the x-axes of (a) and (c), for the other panels it's 2005-2018. I suggest to replace this with 'in the period 1981 – 2018 (a and c) and 2005 – 2018 (b and d-h). (Or change panel

letters, see suggestion below).

Reply Detail comment #8:

We changed panel labelling and added information on used time periods for lake subsets on the x-axis to the figure caption accordingly.

Detail comment #9:

Table 1: This is a pretty large table for only presenting the 6 predictor parameter selected for this study. I suggest to present the 6 parameters used here in separate table, giving some more details as well. (By the way, I think dam type could be ticked as well, at least a tick in brackets. As only moraine-dammed lakes are investigated here, this criteria is inherently considered). If the authors wish to keep having a table with other potentially relevant parameters for GLOF hazard assessment, this could be done in a more compressed format. But in this case, further geotechnical and geomorphic parameters would need to be included, such as permafrost conditions, lithology, seismicity, etc. The annex tables of the GAPHAZ guidelines (GAPHAZ, 2017) might give some indications for this.

Reply Detail comment #9:

To meet the requests from both referees, we split our former Table 1 into two separate tables with an overview table listing the predictors for HKKHN lakes described in the literature (now labelled as Table 1) and a comprehensive table listing our predictor choices (now labelled as Table 2). We added a number of additional parameters to Table 1 and more details (used notation and selection reasoning) to Table 2. Now the indicator "dam type" is also ticked in Table 1.

Detail comment #10:

Fig. 1: According to the caption, white triangles represent GLOFs since 1935. But as the study only deals with GLOFs that have occurred on the periods 1981-2018 and 2005-2018, respectively, only these should be shown here. Preferably with two colors,

one for 1981-2005 and another for 2005-2018 to discriminate these to reference data sets. Please also indicate the spacing of the lake bubbles.

Reply Detail comment #10:

We thank the reviewer for this suggestion and changed the figure accordingly.

Detail comment #11:

Fig. 3: Blue-green combinations are hardly readable in the bubbles. I can see it in the text, slightly see it in the middle ('mulit-level') bubble, but do not see any green in the right ('many models'). Colors to be adjusted.

Reply Detail comment #11:

We modified Fig. 3 to make the principles of multi-level modelling more clear. We also chose a green-purple-yellow colour combination to improve contrast.

Detail comment #12:

Figure 4: I suggest to sort sub-groups from highest (top) to lowest (bottom) in (a) and (b), West to East (or East-West) in (c) and highest monsoonality on top to lowest monsoonality in (d). (Same as ordering in Figs. 5-8).

Reply Detail comment #12:

We appreciate this suggestion. However, the point of the figure is to highlight better the ranked deviations of the group-level coefficients from the pooled mean (at the bottom of the stack). We believe that this ranking allows a better visual assessment of which groups deviate most from the pooled means.

Detail comment #13:

Figs. 5-8 (general): In none of the figures I can see the middle (blueish) line. I only see the purple and orange lines. Similar for the color shades, I guess I only see the purple and the orange and the overlap of the two. Is this middle line represented in the

Figures? If so, please adjust coloring, if not please add (or remove from the legend). For all figures it would be very nice to also have a panel for the pooled data, similar to Fig. 4.

Reply Detail comment #13:

The middle line is grey in the original Figs. 5-8 and lies between the orange and purple lines. We acknowledge that this may be hard to decipher and changed the colour scheme to provide more contrast. Adding a panel for pooled data is a good suggestion and we are to amend the figures as requested.

Detail comment #14:

Figs. 5 and 6: It would be helpful to indicated elevation bands in m a.s.l.

Reply Detail comment #14:

This is a good suggestion and we are to amend the figures as requested.

Detail comment #15:

Fig. 9: To me it would make more sense to number the TP a-d and the TN e-h.

Reply Detail comment #15:

We thank the reviewer for this suggestion and changed the figure accordingly.

Detail comment #16:

Fig. 10: In the legend (e) (the letter e is not needed in my view) swap ordering, that a is on top and d at the bottom, as in the main panels. Ad '%' to the numbers at the bottom. In the panels it would be helpful to include the locations with a recorded GLOF (for 1990-2018 in (a) and 2005-2018 in (b), (c) and (d)).

Reply Detail comment #16:

We thank the reviewer for this suggestion and changed the figure accordingly.

References cited in this Reply

Allen, S. K., Zhang, G., Wang, W., Yao, T. and Bolch, T.: Potentially dangerous glacial lakes across the Tibetan Plateau revealed using a large-scale automated assessment approach, Sci. Bull., (April), doi:10.1016/j.scib.2019.03.011, 2019.

Fujita, K., Suzuki, R., Nuimura, T. and Sakai, A.: Performance of ASTER and SRTM DEMs, and their potential for assessing glacial lakes in the Lunana region, Bhutan Himalaya, J. Glaciol., 54(185), 220–228, doi:10.3189/002214308784886162, 2008.

Fujita, K., Sakai, A., Takenaka, S., Nuimura, T., Surazakov, A. B., Sawagaki, T. and Yamanokuchi, T.: Potential flood volume of Himalayan glacial lakes, Nat. Hazards Earth Syst. Sci., 13(7), 1827–1839, doi:10.5194/nhess-13-1827-2013, 2013.

GAPHAZ: Assessment of Glacier and Permafrost Hazards in Mountain Regions: technical Guidance Document. Standing Group on Glacier and Permafrost Hazards in Mountains (GAPHAZ) of the International Association of Cryospheric Sciences (IACS) and the International Per, Zurich, Lima., 2017.

Gelman, A. and Hill, J.: Data Analysis using Regression and Multilevel/Hierarchical Models, Cambridge University Press, New York., 2007.

Haritashya, U. K., Kargel, J. S., Shugar, D. H., Leonard, G. J., Strattman, K., Watson, C. S., Shean, D., Harrison, S., Mandli, K. T. and Regmi, D.: Evolution and Controls of Large Glacial Lakes in the Nepal Himalaya, Remote Sens. , 10(5), doi:10.3390/rs10050798, 2018.

Huggel, C., Kääb, A., Haeberli, W., Teysseire, P. and Paul, F.: Remote sensing based assessment of hazards from glacier lake outbursts: a case study in the Swiss Alps, Can. Geotech. J., 39(2), 316–330, doi:10.1139/t01-099, 2002.

Mukul, M., Srivastava, V., Jade, S. and Mukul, M.: Uncertainties in the Shuttle Radar Topography Mission (SRTM) Heights: Insights from the Indian Himalaya and Peninsula, Sci. Rep., 7(February), 1–10, doi:10.1038/srep41672, 2017.

Rounce, D. R., McKinney, D. C., Lala, J. M., Byers, A. C. and Watson, C. S.: A new remote hazard and risk assessment framework for glacial lakes in the Nepal Himalaya, Hydrol. Earth Syst. Sci., 20(9), 3455–3475, doi:10.5194/hess-20-3455-2016, 2016.

Shor, B., Bafumi, J., Keele, L. and Park, D.: A Bayesian multilevel modeling approach to time-series cross-sectional data, Polit. Anal., 15(2), 165–181, doi:10.1093/pan/mpm006, 2007.

Stegmueller, D.: How many countries for multilevel modeling? A comparison of frequentist and bayesian approaches, Am. J. Pol. Sci., 57(3), 748–761, doi:10.1111/ajps.12001, 2013.

Zheng, G., Bao, A., Allen, S., Ballesteros-Cánovas, J. A., Yuan, Y., Jiapaer, G. and Stoffel, M.: Numerous unreported glacial lake outburst floods in the Third Pole revealed by high-resolution satellite data and geomorphological evidence, Sci. Bull., 2021.

Please also note the supplement to this comment:
https://tc.copernicus.org/preprints/tc-2020-327/tc-2020-327-AC2-supplement.pdf

---

## Author Response (AR1)

Institute for Environmental Science
and Geography

Karl-Liebknecht-Straße 24-25 Haus 1
14476 Potsdam - Golm
melaniefischer@uni-potsdam.de

Dr. Tobias Bolch
The Cryosphere

16th April, 2021

**Revision of manuscript "Controls of outbursts of moraine-dammed lakes in the greater Himalayan region"**

Dear Dr. Bolch,

thank you for the quick and transparent review process. Please find our point-by-point responses to the referees below. Changes to our manuscript are cited with line numbers which are based on the revised manuscript file without track changes.

We also thank you for your remark concerning potential data sources of glacier-mass balances prior to 2000. Although we decided to not include these in our analysis at this point we will gladly consider them for future studies.

Best regards,

Melanie Fischer
On behalf of all co-authors

**Reply to Referee #1 (RC1)**

*General comment #1:*

> *The authors present interesting results, some of which are novel in a sense that contradict assumptions of previous GLOF hazard assessment studies (e.g. the assumption that fast-growing lakes are more susceptible to GLOF), but this is only one part of the story (so far pretty much model-oriented) in my opinion. If the overall aim is enhanced identification of potential future GLOF sites or so, stronger linkages of investigated GLOF indicators to physical processes behind as well as (at least brief) characterization of documented GLOFs (in terms of triggers, mechanisms) are missing. For instance, how (process-wise) is the EDW, glacier-mass balance or lake (catchment) area linked to documented GLOF? What are triggers of historic GLOFs considered in this study? In fact, I'd expect this to be taken into consideration in the very first step – selection and justification of GLOF indicators.*

We appreciate the reviewer's comment and elaborated in detail the choice of our predictors in a new table (now Table 2). We rewrote large parts of Section 2.1 to make clear how we selected each predictor. For example, we note that:
- "Larger and growing lakes offer more area for impacts from mass flows such as avalanches, rockfalls, and landslides originating from adjacent valley slopes (Haeberli et al., 2017)." (L123-124);
- "A larger upstream catchment area has been associated with an increased susceptibility to GLOFs as more runoff from intense precipitation, together with glacier and snow melt, can lead to sudden increases in lake volume (Allen et al., 2019; GAPHAZ, 2017)." (L126-128);
- "These readily available data on regional glacier-mass balances are proxies for other, less accessible, physical controls on GLOF susceptibility such as glacial meltwater input, either directly from the parent glacier or from glaciers upstream, as well as permafrost decay in slopes fringing the lake." (L133-136);
- "Meteorological drivers entered previous qualitative GLOF hazard appraisals mostly as (the probability of) extreme monsoonal precipitation events: the Kedarnath GLOF disaster, for example, was triggered by intense surface runoff (Huggel et al., 2004; Prakash and Nagarajan, 2017). Heavy rainfall may also trigger landslides or debris flows from adjacent hillslopes followed by displacement waves that overtop moraine dams (Huggel et al., 2004; Prakash and Nagarajan, 2017). Elevated lake levels during the monsoon season also raise the hydrostatic pressure acting onto moraine dams (Richardson and Reynolds, 2000). Furthermore, different precipitation regimes and climatic preconditions may also influence moraine dam failure mechanics (Wang et al., 2012)." (L137-143)

Background information on triggers is scant or conjectural for most GLOFs in our study region. We now offer a more thorough discussion on possible triggers: "The triggering mechanism of these studied GLOFs is reported in only seven cases, four of which are attributed to ice avalanches entering the lake (e.g. Tam Pokhari, Nepal or Kongyangmi La Tsho, India; Ives et al., 2010; Nie et al., 2018). Other triggers of the GLOFs studied here include piping (Yindapu Co, China; Nie et al., 2018) and the collapse of an ice-cored moraine (Luggye Tsho, Bhutan; Fujita et al., 2008)." (L157-160). Please also see our response to general comment #2 in this regard.

Contrary to the reviewer's notion, the goal of our study is not to *"identify potential future GLOF sites or so"*. Our goal is to explore possible predictors of **historic** GLOFs. We had stressed this issue in the original Abstract: "We use a comprehensive inventory of 3,390 morainedammed lakes and their documented outburst history in the past four decades to test whether elevation, lake area and its rate of change, glacier-mass balance, and monsoonality are useful inputs to a probabilistic classification model" (L13-15 of orig. manuscript); and: "We find that mostly larger lakes have been more prone to GLOFs in the past four decades, regardless of elevation band in which they occurred"(L17-18 of orig. manuscript), and in many other locations in the manuscript.

*General comment #2:*

> *It would be interesting at least discuss how many of documented GLOFs were actually triggered by processes associated with investigated GLOF indicators? This is briefly touched in the introduction (L36-39) or study area section (L108), but I'm convinced that bit deeper and more comprehensive elaboration (e.g. a separate discussion section) would be beneficial for readers. Another example - on L244-245 it is mentioned that 'greater lakes are more likely to having had a GLOF ...'. I wonder what do primary data say about this – what proportion of these 31 GLOF-producing lakes would be classified as large at the time of GLOF and what this proportion is in the population of 3,390 moraine-dammed lakes? And in the other way around - can a specific combination of values of GLOF indicators infer about possible (likely) GLOF trigger and mechanism (if not known)?*

The reviewer may acknowledge that the link between mechanistic processes, choice of statistical predictors, and the occurrence of GLOFs invites some interpretation. If we had sufficient data to run adequately parameterised process-based, numerical models on past GLOFs, we could go beyond the outputs of statistical models of outburst susceptibility. Our motivation, however, is to choose predictors as **proxies** instead of physical parameters in a deterministic model. Each of these proxies subsumes various physical processes that might be relevant to producing GLOFs (now further stressed in our new Table 2 in L147 and in the additions we made to Section 2.1 in L123-143). How well these proxies can describe the presence or absence of a GLOF is what our probabilistic models predict. These models learn from the data directly and inform us about the suitability of our predictors to hindcast historic GLOFs. By design, the outputs are probabilities and less so physical triggers or mechanisms. This probabilistic approach forms a cornerstone in modern hazard and risk analyses.

In this context, we are unsure what the reviewer means by "*primary data*". Our model learns from all the data as stated in the Methods section. We also note that the role of lake area in our model is that of a **continuous** predictor and not that of a response variable, as the reviewer seems to suspect. The model summarises how the susceptibility to GLOFs changes with lake area rather than vice versa.

*General comment #3:*

> *Let me also critically comment on some of the selected GLOF susceptibility indicators (in general, I'm convinced it would be useful presenting these indicators in a separate table with more detailed and comprehensive description than stated in the overview Tab. 1, and in places of the text):*

We appreciate this suggestion and split Table 1 into two new tables: Table 2 (L147) now provides more detail on, and motivation for, our selected predictors.

> *Lake area change: I'm aware this indicator is always tricky to define and employ; according to what is written on L134-135, two intervals are used for lake are change (1990-2005 and 2005-2018); considering GLOFs occurring throughout the period 1981-2017, it means than these intervals may be pre-GLOF, post-*

*GLOF or the GLOF occurred somewhen during one of these intervals – please comment on how this inconsistency was treated and whether it can explain that no link was observed between lake area change and the occurrence of GLOF.*

The data on lake-area changes are not yet resolved on an annual basis for our study area, so that we had to resort to changes averaged over longer periods. However, we used our forecasting model to test whether changes in lake size between two observation periods had a credible effect on $P_{\text{GLOF}}$. Here we explored the weight of relative changes in lake area between 1990 and 2005 to estimate the probability of observing GLOFs that happened in the subsequent period 2005-2018. In other words, we trained the model on GLOF data that **pre-date the testing data**, and thus offer a realistic and rigorous prediction and validation scenario. We reported in our original manuscript, that "The weight of relative lake-area change in the 15 years before is ambiguous ($\beta_{A^*} = -0.04^{+0.76}/_{-0.67}$) […]" (L245-246 of orig. manuscript) and that "In the forecasting model, however, the influence of lake-area change remains negligible even for <50% HDIs." (L350 of orig. manuscript). This indicates with 95% probability that relative lake-area change **before** the outburst is an inconclusive predictor. To further stress this result, we added the following sentences in the Discussion: "However, in the forecasting model, in which we tested whether differing data observation periods have any credible effects, the influence of lake-area change remains negligible even for <50% HDIs. We thus conclude that relative lake-area change before outburst is an inconclusive predictor. This result contradicts the assumptions made in many previous studies that argued that rapidly growing lakes are the most prone to sudden outburst (Aggarwal et al., 2016; Bolch et al., 2011; Ives et al., 2010; Mergili and Schneider, 2011; Prakash and Nagarajan, 2017; Rounce et al., 2016; Wang et al., 2012)." (L382-387).

*Glacier mass balance: similarly to my comment on lake area change - how can 2000-2016 glacier mass balance be used to explain GLOFs occurring throughout the period 1981-2017? These characteristics (mass balance as well as lake area change) are dynamic in nature and I'm wondering how can a static information from available datasets possibly blur a GLOF signal, especially for pre-2000 GLOFs?*

The problem of limited data for lake-area changes is even more pronounced for glacier-mass balances in our study area. Again, the data averaged from 2000 to 2016 are among the few regionally consistent data sets that we considered as input for our model. The underlying assumption is that the regional regime of prevalent glacier melting from 2000 to 2016 largely follows a trend dating back to the late 1980s. This is in line with the review of Bolch et al. (2019) who summarized that *"glaciers* [in High Mountain Asia] *have thinned, retreated, and lost mass since the 1970s, except for parts of the Karakoram, eastern Pamir, and western Kunlun"* (p. 211).

To answer the reviewer's question of "[…] *how can a static information from available datasets possibly blur a GLOF signal, especially for pre-2000 GLOFs?"*, we curtailed our glacier-mass balance model only to those lakes that had documented outbursts **after 2000,** and hence that overlap with the study period of Brun et al. (2017). Table R1 shows the output from this model:

*Table R1: Posterior model parameter estimates for the glacier-mass balance model trained on a post-2000 GLOF subset.*

| Model parameter | Estimate | Estimation error | Lower 95% CI boundary | Upper 95% CI boundary |
|---|---|---|---|---|
| $\alpha_{z,r}$ | -7.32 | 1.36 | -10.40 | -5.02 |
| $\beta_{A^* b}$ (2005 to 2018) | -0.62 | 0.29 | -1.19 | -0.06 |
| $\beta_C$ | 0.88 | 0.22 | 0.44 | 1.32 |
| $\gamma_r$ | -2.87 | 3.00 | -9.64 | 2.13 |

| | | | | |
|---|---|---|---|---|
| $\sigma_z$ | 0.48 | 0.42 | 0.02 | 1.53 |
| $\sigma_r$ | 1.11 | 0.72 | 0.08 | 2.88 |

We find that:
- The parameter estimates at the population level changed only minutely: the weight of catchment area ($\beta_C$) remains credibly positive and that of lake-area change from 2005 to 2018 $\left(\beta_{A*b}\right)$ remains credibly negative.
- At the group level, the standard deviation of intercepts of our grouping variable elevation ($\sigma_z$) is also similar to our previous results.
- Posterior estimates of $\sigma_r$, the standard deviation of group level intercepts of glacier-mass balance regions, increase from $0.81^{+1.60}/_{-0.78}$ to $1.11^{+1.77}/_{-1.03}$, **though with much overlap**. This further underlines our finding that the glacier-mass balance in a given region credibly affects $P_{GLOF}$.

We now highlight these findings in our results: "On the basis of higher standard deviations, we learn that effects of glaciological regions vary more than those of elevation bands ($\sigma_r = 0.81^{+1.60}/_{-0.78}$ and $\sigma_z = 0.48^{+1.19}/_{-0.47}$). When training this model on a subset of glacial lakes with documented GLOFs that happened after 2000 (i.e. including only those in the interval covered by glacier-mass balance data), posterior estimates of $\sigma_r$ increase to $1.11^{+1.77}/_{-1.03}$, further underlining our result that glacier-mass balance credibly affects $P_{GLOF}$." (L304-308).

> *Monsoonality: using climate indicators in GLOF research is promising, but proportion of summer precipitation doesn't tell you about the extremity; for instance, the proportion will be lower in areas where extreme rainfalls occur in summer, but also some precipitation in winter, but will be super-high in generally dry areas with some precipitation during the summer and no precipitation in winter. But process-wise, the first area will have much higher potential to trigger GLOF in my opinion.*

We are unsure about whether the reviewer offers an opinion here or whether their statement is supported by data. Our analysis shows that the proportion of summer precipitation is highest in areas with strong monsoonal influence (Fig. 1). We are unaware of any GLOFs that have been reported in winter. For example, ice cover on lakes and freezing moraine dams have been thought to make glacier lakes resilient against outbursts, even during strong seismic shaking (Kargel et al., 2015). Most of the heavy rainstorms are tied to the South Asian summer monsoon, and some reported GLOFs were triggered by such storms (Allen et al., 2016; Liu et al., 2014). The drier areas of our study area usually receive higher amounts of precipitation during winter via westerlies (Bolch et al., 2012; Bookhagen and Burbank, 2010), so we think that monsoonality remains a useful predictor and did not change our manuscript in this regard.

*Detail comment #1:*
> *L11: yes, the approach is quantitative, but selection of GLOF indicators in this study is also expert judgement-based as the authors are GLOF experts*

We rephrased this sentence to "Estimating regional susceptibility of glacial lakes has largely relied on qualitative assessments by experts, thus motivating a more systematic and quantitative appraisal." (L10-12).

*Detail comment #2:*
> *L34: see also Cook et al., 2018, Science*

We thank the reviewer for suggesting this useful reference, which we added to our manuscript (L35).

*Detail comment #3:*
> *L36-37: this needs deeper elaboration in relation to selected GLOF susceptibility indicators (see also my general comment)*

We refer the reviewer to our reply on General Comment #3.

*Detail comment #4:*
> *L103: I suggest to use 'GLOF susceptibility indicators' instead of 'diagnostics of GLOF potential' or 'diagnostics of GLOF hazard' (L125); similarly, 'controls' and 'predictors' are used throughout the manuscript, please define a difference or unify*

We decided to now consistently use the term "predictor", in line with the common terminology in the statistics literature. Our use of "diagnostic" is also appropriate, as the regression model has a bivariate outcome. Yet to make things more clear, we replaced "diagnostic" by "predictor" throughout the manuscript. We avoid the term "indicator", as it may be confusing in the model context. In regression models, an indicator variable is often a logical binary [0, 1] or dummy variable, whereas we mostly use continuous variables.

*Detail comment #5:*
> *L111-112: lake deepening increases hydrostatic pressure, not areal or volumetric growth*

This is physically more correct, though we fail to see how any change in lake area or volume could not affect hydrostatic pressure eventually. To be more clear, we rewrote our statement to: "Lake area scales with lake volume and depth (Huggel et al. 2002), and growing lake depths increase the hydrostatic pressure on moraine dams, thus raising the potential of failure (Rounce et al., 2016)." (L119-121).

*Detail comment #6:*
> *L115-116: the authors usually argue that larger lakes are more susceptible because large lake areas are more exposed to slope movements potentially triggering GLOFs; large area is also correlated with larger depth (and so hydrostatic pressure acting on a dam)*

We added this reasoning to the text: "Larger and growing lakes offer more area for impacts from mass flows such as avalanches, rockfalls, and landslides originating from adjacent valley slopes (Haeberli et al., 2017)." (L123-124).

*Detail comment #7:*
> *L130: how is different date of GLOF and input data for model treated? (how possibly different environmental conditions at the time of GLOF and at the time of datasets acquisition can influence your results?) see also my general comments*

We used the reported dates of historic GLOFs where available. We acknowledge that our predictor variables can only approximate the environmental conditions at the time of lake outburst, but this is the point of a statistical predictor in a data-driven model. Please see our replies to General Comments #1-3 in this regard.

*Detail comment #8:*
    *L136-139: I suggest to move this to L133*

We moved and slightly rephrased this text passage accordingly: "These variables are correlated with elevation because of the same underlying interpolation technique so that we limited our models to those with poorly correlated predictors. This meant omitting other predictors such as mean annual temperature, annual precipitation totals and annual temperature and precipitation variability." (L163-165).

*Detail comment #9:*
    *L166: delete 's'*

Deleted accordingly.

*Detail comment #10:*
    *L176: delete ','*

Deleted accordingly.

*Detail comment #11:*
    *L207: what is meant by 'common susceptibility'?*

For clarification, we rephrased this sentence to "In essence, this varying-intercept model acknowledges that glacial lakes in the same elevation band may have had a common baseline susceptibility to GLOFs in the past four decades." (L239-240).

*Detail comment #12:*
    *L262-263: this step is not clear to me? Please explain*

To clarify our approach, we added more details on our predictor catchment area in the new Table 2 (L147). We explain our choice of this predictor and why we use it instead of the static lake area *A* in the glacier-mass balance and monsoonality models: "We also tested the impact of upstream **catchment area** *C* (m²) on GLOF susceptibility. A larger upstream catchment area has been associated with an increased susceptibility to GLOFs as runoff from intense precipitation as well as glacier and snow melt can lead to sudden increases in lake volume (Allen et al., 2019; GAPHAZ, 2017)." (L126-128).

*Detail comment #13:*
    *L263: please provide details about this correlation*

We find that catchment area *C* has a strong linear correlation with lake area *A* (Pearson's correlation coefficient of 0.446), such that we preferred *C* over *A* in two of our models, as *C* is invariant at the timescale of our study. We also added this to the text: "We find that catchment area *C* correlates with lake area *A* (Pearson's $\rho = 0.45$) and, thus preferred *C* over *A* in two of our models, as *C* is invariant at the timescale of our study." (L128-130).

*Detail comment #14:*
    *L272: what is meant by 'average lake'?*

The average lake is defined by the combination of all average predictor values. We added this definition to the text: "This model has the highest values of $P_{GLOF}$ for average lakes (i.e. all

average predictor values combined) in the Nyainqentanglha Mountains and the Eastern Himalaya (Fig. 4)." (L209-311).

*Detail comment #15:*
> *L352-354: this can be true for a specific period in long-term evolution of a mountain range (considering gradual glacier retreat and overall shift of all rapid processes including GLOFs to higher elevation zones; i.e. the general shift of morphoclimatic zones)*

The point we wanted to make here was that stratifying by elevation hardly helped to inform us more about GLOF susceptibility in this context. In essence, GLOF susceptibility is aptly represented by the pooled model here. We did not change our manuscript in this regard.

*Detail comment #16:*
> *L365: so why not to consider this indicator in your model?*

We ran a number of models that used the distance from the parent glacier as a predictor, though obtained no credible posterior weights. However, we found that this distance is most likely the most prone to highly dynamic changes in historic times. We thus added to the text: "However, systematically recorded time series of glacier fronts are even harder to come by when compared to systematic measurements of changes in glacial-lake areas." (L403-404).

*Detail comment #17:*
> *L373: are 'minute'? Please check*

We clarified this: "[…] deviations from a pooled model for the HKKHN are minute when compared to the spread of posterior group-level intercepts in the other models (Fig. 4)." (L411-412).

*Detail comment #18:*
> *Tab 2: what is PDF?*

The abbreviation PDF stands for probability density function. We changed the column header in Table 3 (former Table 2) accordingly (L227).

*Detail comment #19:*
> *Tab. 4: please also consider presenting false positives and false negatives*

We accordingly added false positives and false negatives in a separate column to the table (now Table 5, L364).

*Detail comment #20:*
> *Fig. 2: three lake inventories are mentioned (ICIMOD, Veh et al., 2019 and Wang et al., 2020); please make clear how these were integrated; these 3,390 lakes (L131) are from which inventory?*

The 3,390 lakes forming our database are a subset of the ICIMOD inventory published by Maharjan et al. (2018). We changed Figure 2 to better show this. We also clarified in the text that "Second, we identified from an independent regional GLOF inventory (Veh et al. 2019) 31 lakes that had at least one outburst between 1981 and 2017 and that are listed in the ICIMOD inventory." (L155-157).

*Detail comment #21:*

>*Fig. 3: how about green color in Many Models part?*

We modified our figure to clarify this and also changed the colour scheme (as requested by referee #2) for improved contrasts.

*Detail comment #22:*

>*Fig. 10: please consider highlighting GLOF-producing lakes; switch a-d in the panel (e)*

We modified the figure accordingly.

**Reply to Referee #2 (RC2)**

*General comment #1:*

> *Hazard concept: The article is strictly focusing on GLOF susceptibility and using this term consistently throughout the manuscript. Nevertheless, I think regarding some aspects of the study, concepts and terminologies are mixed at some places. According to international standards from UNISDR, IPCC etc., hazard is a function of probability (of occurrence) and intensity (or magnitude). Susceptibility in turn 'is a relative measure of the likelihood (or probability) that a hazard will occur or initiate from a given site, based on intrinsic properties and dynamic characteristics of that site' and 'has an inverse relationship with stability' (GAPHAZ, 2017). I.e., susceptibility can be considered as probability of occurrence and is one factor of hazard. It is determined by conditioning factors (=inherent and more or less static factors) on the one hand, and triggering factors (=factors that directly initiate an outburst) on the other. The factors (predictors) analyzed in this paper, are limited (for good reasons!) to conditioning factors. In other words, the result of an analyses based on the parameters used in the present study, is mainly a lake stability assessment.*

We thank the reviewer for this observation and these definitions. However, we would have hoped for some guidance as to where exactly we might have mixed concepts and terminologies in this regard. The referee's specific comments do not pick up this issue as promised. Perhaps some of the confusion arises from both qualitative and quantitative uses of the term "hazard". We echo the reviewer's comments on hazard and susceptibility in principle. Yet we wish to stress that our model is far from a stability assessment of moraine dams. The reviewer may agree that such appraisals frequently hinge on hard classes such as "stable" or "unstable". Even if such a geotechnical or engineering geological appraisal would be feasible, it would require data on the internal structure and geometry of moraine dams, including grain-size distribution, presence and size of ice cores, the volume, width, height, and slope of moraine dams, pore water pressure in the dam, armouring of the outlet channel, presence and opening of tension cracks, rates of subsidence, and many others. Such parameters have been likely prone to change during our study period and are difficult to obtain for a single lake, and so even less feasible for the size of our regional study. To avoid an elusive feeling of stability, and hence, safety, we refrain to call our approach a dam-stability assessment.

Strictly speaking, our analysis estimates the **probability of correctly detecting historic lake outbursts from a set of predictors**. The referee may acknowledge that this probability is indeed a likelihood of GLOF outburst conditioned on reporting. In this sense one could see this metric as a "relative measure of the likelihood (or probability) that a hazard will occur or initiate from a given site" as the referee suggests. Our forecasting model in particular addresses this scenario. To further stress this point, we added the following definitions and statements to our introduction: "Our method estimates the probability of correctly detecting historic GLOFs from a set of predictors, which act as proxies subsuming various physical processes described as being relevant to GLOFs. Triggering mechanisms of these GLOFs are rarely reported, however. Thus, we discuss what we can learn more about how these historic GLOFs were linked to readily available measures of topography, monsoonality, and glaciological changes. Our model results provide a posterior probability of outburst conditioned on detection, and this may be used as a relative metric of GLOF release from a given lake. Therefore, our approach is an alternative to a formal assessment of moraine-dam stability, which is (geo)technically feasible only at selected sites and at scales much finer than our regional and decadal focus." (L75-81).

*In contrast to this, most of the mentioned regional glacial lake assessment approaches with more expert based, and probably subjective, parameter weightings, follow a hazard assessment approach, rather than a stability/susceptibility assessment.*

From the literature that we compiled in this and our previous work on GLOFs, we infer that very few, if any, of these so-called hazard assessments offer probabilistic metrics that satisfy the formal quantitative definition of hazard, as the referee clearly points out. During our review, we found that most of these studies deal with hazard in a qualitative or semi-quantitative way. We now acknowledge this in our manuscript: "Specifically, we tested how well some of the more widely used predictors of GLOF susceptibility and hazard fare in a multi-level logistic regression that is informed more by data rather than by expert opinion." (L72-73).

*The reasons why these other studies consider factors like lake area or volume, or regional glacier mass balance, is not mainly because these factors directly influence lake stability, but because they have an impact on hazard potentials. Larger lake volumes (area is often used as a proxy for volume) and lake growth imply higher potential flood volumes, and therefore increase the hazard due to higher intensities, without affecting GLOF susceptibility. For similar reasons glacier masse balance is included in such models: Negative regional mass balances lead to glacier retreat and the formation of new and growth of existing lakes. Both processes increase the GLOF hazard potential in a region, but only have minor effects on GLOF susceptibility of individual lakes.*

We agree with the referee in principle here. Yet we found it difficult to trace objectively any increases (or even changes) in hazard in the literature due a distinct lack of the necessary probabilistic metrics.

*Further, unfavorable conditioning factors do not lead to a lake outburst immediately. It of course increases GLOF susceptibility, but requires still a triggering event to initiate an outburst. Clague and Evans (2000) and Emmer et al. (2020) present concepts about the timing of the causal chain of climate change, glacier retreat, glacial lake formation, and glacial lake outburst and conclude, based on empirical data from British Columbia and the Cordillera Blanca, that there is a lag between lake formation and outburst of up to several decades. The fact that a lake did not have an outburst event in the periods investigated in this study, does not automatically imply that the lake has a low GLOF susceptibility. It is indeed possible, that the lake is actually unstable (i.e. has a high susceptibility) but an outburst simply has not been triggered (yet).*

We reiterate our point above and state that our method is set out to detect reported GLOFs, as now further stressed in the added introductory definitions in L75-81. The referee's comment on dam stability is important and should be considered in geotechnical assessments, but is tangential to our objectives. Nowhere did we state that we wanted to quantify or estimate the stability of moraine dams. We also do acknowledge the concept of lag times between lake formation and outburst: every lake has a life span, but the question is whether an outburst needs to end it. This concept of lag time is thought provoking, but hinges on data similar to the models that we present here. One major advantage in our models is that we can fully capture the underlying uncertainties, something that we have so far yet to see for any lag-time model.

*General comment #2:*

*Used data and parameters: Data availability for the entire study region is of course an important criterion for the selection of predicting parameters. But in addition to the parameters investigated in this study, there are candidates for other parameters which are often and successfully applied in other regional assessment approaches cited in the study, such as the Steep Lake front Area (SLA) developed by Fujita et al. (2013) and used by Rounce et al. (2016), or the topographic potential for rock or ice avalanches (cf. Allen et al., 2019), one of the most frequent GLOF triggers in High Mountain Asia. Considering this, I suggest to include more details about the selection of the predicting parameters.*

We wish to refer the reviewer to the changes that we have made following the suggestions of referee #1. We rewrote large parts of Section 2.1 to make clear how we selected each predictor. For example, we note that:
- "Larger and growing lakes offer more area for impacts from mass flows such as avalanches, rockfalls, and landslides originating from adjacent valley slopes (Haeberli et al., 2017)." (L123-124);
- "A larger upstream catchment area has been associated with an increased susceptibility to GLOFs as more runoff from intense precipitation, together with glacier and snow melt, can lead to sudden increases in lake volume (Allen et al., 2019; GAPHAZ, 2017)." (L126-128);
- "These readily available data on regional glacier-mass balances are proxies for other, less accessible, physical controls on GLOF susceptibility such as glacial meltwater input, either directly from the parent glacier or from glaciers upstream, as well as permafrost decay in slopes fringing the lake." (L133-136);
- "Meteorological drivers entered previous qualitative GLOF hazard appraisals mostly as (the probability of) extreme monsoonal precipitation events: the Kedarnath GLOF disaster, for example, was triggered by intense surface runoff (Huggel et al., 2004; Prakash and Nagarajan, 2017). Heavy rainfall may also trigger landslides or debris flows from adjacent hillslopes followed by displacement waves that overtop moraine dams (Huggel et al., 2004; Prakash and Nagarajan, 2017). Elevated lake levels during the monsoon season also raise the hydrostatic pressure acting onto moraine dams (Richardson and Reynolds, 2000). Furthermore, different precipitation regimes and climatic preconditions may also influence moraine dam failure mechanics (Wang et al., 2012)." (L137-143)

Fujita et al. (2013) used the SLA approach to derive Potential Flood Volumes (PFVs) of Himalayan lakes as a proxy for GLOF susceptibility. We assume that PFVs are largely represented by our predictor lake area, given that larger lakes should produce larger floods. A major critique of the SLA concept is that lakes can have zero PFV despite large lake volumes. This issue was observed, for example, at Imja Lake in the Mt. Everest region, Nepal, that stores $78.4 \times 10^6$ m$^3$ of water surrounded by steep slopes (Haritashya et al., 2018). Fujita et al. (2013) also point towards an issue that the reviewer had cautioned against above (p. 1834): *"PFVs were simply calculated from the topography surrounding the moraine-dammed lakes and thus the robustness of the dam could not be evaluated. As the existence of ice within the damming moraine may alter the dam's vulnerability, understanding the distribution and degradation of permafrost will be an important factor for the further assessment of GLOF probability".* Furthermore, SLA depends on user-defined cutoffs, for example a 1-km search radius for steep slopes around lakes (Fujita et al., 2013), or a minimum slope threshold (Rounce et al., 2016). Such thresholds introduce additional subjective bias that we wished to avoid in our appraisal. Finally, errors in digital elevation models in high mountains remain unaccounted for in these slope-based metrics, though have been acknowledged for many years (Fujita et al., 2008). For example, Mukul et al. (2017) reported that *"vertical accuracy of the data decreases with increase in slope and elevation due to presence of large outliers and voids. Therefore, studies*

*using SRTM data "as is", especially in regions like the Himalaya, are not statistically meaningful".* In summary, these findings motivated us to keep the influence of potentially error-prone model inputs at a minimum.

> *Then, the influence of overlapping time periods of the different data sets used should be discussed in more detail, as also mentioned in the review of A. Emmer (Emmer, 2021). In particular the fact that the lake area change period overlaps the period which is investigated for GLOF occurrence, in my view disqualifies this parameter to be considered, as actually discussed in L340-344.*

We again wish to refer the referee to our reply to referee #1:

The data on lake-area changes are not yet resolved on an annual basis for our study area, so that we had to resort to changes averaged over longer periods. However, we used our forecasting model to test whether changes in lake size between two observation periods had a credible effect on $P_{GLOF}$. Here we explored the weight of relative changes in lake area between 1990 and 2005 to estimate the probability of observing GLOFs that happened in the subsequent period 2005-2018. In other words, we trained the model on GLOF data that **pre-date the testing data**, and thus offer a realistic and rigorous prediction and validation scenario. We reported in our original manuscript that "The weight of relative lake-area change in the 15 years before is ambiguous ($\beta_{A^*} = -0.04^{+0.76}/_{-0.67}$) [...]" (L245-246 of orig. manuscript) and that "In the forecasting model, however, the influence of lake-area change remains negligible even for <50% HDIs." (L350 of orig. manuscript). This indicates with 95% probability that relative lake-area change **before** the outburst is an inconclusive predictor.

To further stress this result, we added the following sentences in the Discussion: "However, in the forecasting model, in which we tested whether differing data observation periods have any credible effects, the influence of lake-area change remains negligible even for <50% HDIs. We thus conclude that relative lake-area change before outburst is an inconclusive predictor. This result contradicts the assumptions made in many previous studies that argued that rapidly growing lakes are the most prone to sudden outburst (Aggarwal et al., 2016; Bolch et al., 2011; Ives et al., 2010; Mergili and Schneider, 2011; Prakash and Nagarajan, 2017; Rounce et al., 2016; Wang et al., 2012)." (L382-387).

*General comment #3:*
> *Statistical significance: Bayesian approaches are certainly most suitable for this type of research question where a large number of lakes (3,390) had relatively few (31) GLOF events. But still this is a very limited data basis, in particular since for the Forecasting, the Glacier-mass balance, and the Monsoonality models, only 11 GLOF events were recorded in the relevant 2005 to 2018 period. Even more, these 11 events are split over four to seven groups, depending on elevation, region, or monsoonality. Over the western half of the study region, only 3 GLOFs are found. This leads to very few (often only 1 or 2) or even zero GLOF events per subgroup (cf. boxes for Hindu Kush, Karakoram and Western Himalayas in Fig. 7). I wonder, how any predictor weights can be found in these cases.*

The problem of highly imbalanced data (few GLOF reports out of several thousand lakes) was a major motivation for us to use Bayesian models. The low prior probability of detecting a reported GLOF can be compared directly with the posterior probability, a strategy that we showed in our original Fig. 9. Classical rare-events logistic regression penalises the model

likelihood, and this step is done naturally via the prior distributions in the Bayesian setting. The low number of data points in some groups is even less of an issue in a hierarchical model, as this always draws strength across each group and the pooled model of all data taken together. The high posterior uncertainties tied to some model groups clearly underline the effect of fewer data points. To further emphasise these points, we added that: "The small number of reported GLOFs introduces strong imbalance to our data, given that some regions, and hence levels, had few or no reported GLOFs. Although this would be problematic in most other modelling approaches, Bayesian multi-level models are well suited for this kind of imbalanced training data (Gelman and Hill, 2007; Shor et al., 2007; Stegmueller, 2013)." (L208-211) and in the Discussion that: "In the forecasting model, however, the contributions of lake elevation to $P_{GLOF}$ are devoid of any systematic pattern and likely reflect several, potentially combined, drivers (Fig. 4). This model was trained on fewer GLOFs and the imbalance in the data introduces more uncertainties in terms of broad 95% HDIs." (L391-393).

> *A very recent study from Zheng et al. (2021) on a slightly larger study region found evidence for a total of 215 GLOF events that presumable have happened since 1900, 176 thereof so far unreported. This does not contradict any of the data used here, but offers at least a potential alternative of a database with much more GLOF evidences (in turn posing challenges on the predictor data of course).*

We checked the study by Zheng et al. (2021) but found mostly GLOFs without timestamps that are difficult to align with our predictors, some of which are averaged over specified time periods. We, thus, decided not to add this source to our study. Please also see our reply to referee #1 with respect to the validity of predictors that change over time.

*Detail comment #1:*
> *L11: I suggest to include something like 'regional-scale' (hazard estimations), because at the level of individual lakes, there are many quantitative assessments available, including numerical modeling, geophysical measurements etc.*

We rephrased this sentence accordingly to: "Estimating regional susceptibility of glacial lakes […]." (L11-13).

*Detail comment #2:*
> *L21: Maybe change 'with respect to' to 'compared to'?*

We changed the phrasing as requested (L21).

*Detail comment #3:*
> *L81: Indicate the version number of the RGI*

We added the version number 6.0 (L89).

*Detail comment #4:*
> *L112: Hydrostatic pressure acting on the dam depends mainly on lake depth, not area.*

We acknowledge that this is physically more correct, though we fail to see how any change in lake area or volume could not affect hydrostatic pressure eventually. We accordingly rephrased our statement to: "Lake area scales with lake volume and depth (Huggel et al.,

2002), and growing lake depths increase the hydrostatic pressure acting on moraine dams, thus raising the potential of failure (Rounce et al., 2016)." (L119-121).

*Detail comment #5:*
> *L263: The statement that upstream catchment area is well correlated with lake area is not clear to me. This needs further explanations of references. Also I do not understand why lake area is replaced by upstream catchment area in these model (Glacier-mass balance and Monsoonality), but not in others. This requires some more explanation.*

We thank the reviewer for this suggestion and added more details on our predictor catchment area. We refer to our replies to referee #1's detail comments #12 and #13:

To clarify our approach, we added more details on our predictor catchment area in the new Table 2 (L147). We explain our choice of this predictor and why we use it instead of the static lake area $A$ in the glacier-mass balance and monsoonality models: "We also tested the impact of upstream **catchment area** $C$ (m²) on GLOF susceptibility. A larger upstream catchment area has been associated with an increased susceptibility to GLOFs as runoff from intense precipitation as well as glacier and snow melt can lead to sudden increases in lake volume (Allen et al., 2019; GAPHAZ, 2017)." (L126-128).

We find that catchment area $C$ has a strong linear correlation with lake area $A$ (Pearson's correlation coefficient of 0.446), such that we preferred $C$ over $A$ in two of our models, as $C$ is invariant at the timescale of our study. We also added this to the text: "We find that catchment area $C$ correlates with lake area $A$ (Pearson's $\rho$ = 0.45) and, thus preferred $C$ over $A$ in two of our models, as $C$ is invariant at the timescale of our study." (L128-130).

*Detail comment #6:*
> *L285: In the Forecasting and Glacier-mass balance models, A\* represents lake-area change between 2005 and 2018. Is A\* here also referring to this period (and not 1990 – 2018, as written)? If so, please correct, if not, another symbol should be used (∆A?).*

Our predictor relative lake-area change $A^*$ (not to be confused with net lake-area change $\Delta A$) is calculated for three different time windows: 1990 to 2005 in the forecasting model, 2005 to 2018 in the glacier-mass balance model, and 1990 to 2018 in the monsoonality model. In order to avoid confusion and correctly refer to each respective time interval of relative lake-area change, we now assigned each with its own superscript: relative lake-area change between 1990 to 2005 is $A^{*a}$, relative lake-area change between 2005 and 2018 is $A^{*b}$, and relative lake-area change between 1990 and 2018 is $A^{*c}$. We explain this notation in our new Table 2 and in the respective model descriptions in the Results section 3 (L269-340).

*Detail comment #7:*
> *L321/Fig. 9: Why are the log-odds ratios negative for the first (few) lakes? Would be interesting to describe in the text.*

The negative log-odds ratios indicate lakes for which the posterior probability of a reported GLOF is lower than the prior probability. To further clarify this, we rephrased this in the text to: "A positive log-odds ratio means that we obtain a higher posterior probability of attributing a historic GLOF to a given lake compared to a random draw. Negative log-odd ratios indicate lakes for which the posterior probability of a reported GLOF is lower than the prior probability." (L347-349).

*Detail comment #8:*

*L323/324 (Caption Fig. 9): '...in the past four decades' applies only to the lakes in the x-axes of (a) and (c), for the other panels it's 2005-2018. I suggest to replace this with 'in the period 1981 – 2018 (a and c) and 2005 – 2018 (b and d-h). (Or change panel letters, see suggestion below).*

We changed panel labelling and added information on used time periods for lake subsets on the x-axis to the figure caption accordingly (L356-361).

*Detail comment #9:*

*Table 1: This is a pretty large table for only presenting the 6 predictor parameter selected for this study. I suggest to present the 6 parameters used here in separate table, giving some more details as well. (By the way, I think dam type could be ticked as well, at least a tick in brackets. As only moraine-dammed lakes are investigated here, this criteria is inherently considered). If the authors wish to keep having a table with other potentially relevant parameters for GLOF hazard assessment, this could be done in a more compressed format. But in this case, further geotechnical and geomorphic parameters would need to be included, such as permafrost conditions, lithology, seismicity, etc. The annex tables of the GAPHAZ guidelines (GAPHAZ, 2017) might give some indications for this.*

To meet the requests from both referees, we split our former Table 1 into two separate tables with an overview table listing the predictors for HKKHN lakes described in the literature (now labelled as Table 1, L83) and a comprehensive table listing our predictor choices (now labelled as Table 2, L147). We added a number of additional parameters to Table 1 and more details (used notation and selection reasoning) to Table 2. Now the indicator "dam type" is also ticked in Table 1.

*Detail comment #10:*

*Fig. 1: According to the caption, white triangles represent GLOFs since 1935. But as the study only deals with GLOFs that have occurred on the periods 1981-2018 and 2005-2018, respectively, only these should be shown here. Preferably with two colors, one for 1981-2005 and another for 2005-2018 to discriminate these to reference data sets. Please also indicate the spacing of the lake bubbles.*

We thank the reviewer for this suggestion and changed the figure accordingly.

*Detail comment #11:*

*Fig. 3: Blue-green combinations are hardly readable in the bubbles. I can see it in the text, slightly see it in the middle ('mulit-level') bubble, but do not see any green in the right ('many models'). Colors to be adjusted.*

We modified Fig. 3 to make the principles of multi-level modelling more clear. We also chose a green-purple-yellow colour combination to improve contrast.

*Detail comment #12:*

*Figure 4: I suggest to sort sub-groups from highest (top) to lowest (bottom) in (a) and (b), West to East (or East-West) in (c) and highest monsoonality on top to lowest monsoonality in (d). (Same as ordering in Figs. 5-8).*

We appreciate this suggestion. However, the point of the figure is to highlight better the ranked deviations of the group-level coefficients from the pooled mean (at the bottom of the stack). We believe that this ranking allows a better visual assessment of which groups deviate most from the pooled means and, thus, made no changes to Figure 4.

*Detail comment #13:*
> *Figs. 5-8 (general): In none of the figures I can see the middle (blueish) line. I only see the purple and orange lines. Similar for the color shades, I guess I only see the purple and the orange and the overlap of the two. Is this middle line represented in the Figures? If so, please adjust coloring, if not please add (or remove from the legend). For all figures it would be very nice to also have a panel for the pooled data, similar to Fig. 4.*

The middle line is grey in the original Figs. 5-8 and lies between the orange and purple lines. We acknowledge that this may be hard to decipher and changed the colour scheme to provide more contrast. Adding a panel for pooled data is a good suggestion and we altered the figures as requested.

*Detail comment #14:*
> *Figs. 5 and 6: It would be helpful to indicated elevation bands in m a.s.l.*

This is a good suggestion and we modified the figures as requested.

*Detail comment #15:*
> *Fig. 9: To me it would make more sense to number the TP a-d and the TN e-h.*

We thank the reviewer for this suggestion and changed the figure accordingly.

*Detail comment #16:*
> *Fig. 10: In the legend (e) (the letter e is not needed in my view) swap ordering, that a is on top and d at the bottom, as in the main panels. Ad '%' to the numbers at the bottom. In the panels it would be helpful to include the locations with a recorded GLOF (for 1990-2018 in (a) and 2005-2018 in (b), (c) and (d)).*

We thank the reviewer for this suggestion and changed the figure accordingly.

References cited in this Author's Reply

[revised manuscript text omitted]

---

## Author Response (AR2)

**Response to Referee Report #2:**

*Minor comment #1:*

> *My first general comment in the initial review was referring to the terminologies around hazard and susceptibility. Related adjustments in the new manuscript regarding this point are good and helpful, namely the consistent renaming of what was initially called "GLOF hazard parameters" or "diagnostics of GLOF potential" to now "predictors of GLOF susceptibility" is a clear improvement. The only point where I am not yet convinced is the statement in the conclusions (L481-482): "In any case, the widely adapted notion that (rapid) lake growth may be a predictor of impending outburst remains poorly supported by our model results". In my view lake growth is not mainly considered as an outburst predictor in these assessments (i.e. a proxy for a high outburst susceptibility), but rather as a predictor of increased downstream hazard potential due to increased potential flood volume and thus hazard magnitude. I might be wrong, but since this statement is also used in the abstract, I would ask the authors to more clearly elaborate from the cited publications about these assessment schemes, if any why lake area changes are really considered as predictors of lake outburst susceptibility (rather than indicators of increased potential flood intensities downstream). This could be done, for instance, around L122 where it now says "and many studies have emphasised the role of growing lakes on GLOF susceptibility (Bolch et al., 2011; Prakash and Nagarajan, 2017; Rounce et al., 2016)".*

We thank the reviewer for giving us the opportunity to elaborate more on the roles of lake area and lake area growth in previous appraisals on GLOF susceptibility. We agree that some studies included these predictors to estimate GLOF hazard and risk downstream. However, other studies emphasize that a larger lake area could make lakes more prone both to external triggers, such as mass flows entering the lake, and to internal triggers, such as an increasing hydrostatic pressure on the moraine dam. For example, GAPHAZ (2017) reported that "large lakes obviously can produce potentially greater flood magnitudes, but larger lakes also are more susceptible to impacts from rock and ice." Rounce et al. (2016) noted that "lake growth is crucial to incorporate into hazard assessments as the expansion of a glacial lake may greatly alter the lake's proximity to potential hazards and increase the volume of water likely released in a GLOF." Prakash and Nagarajan (2017) argued that "Characteristics of the lake and its mother glacier, such as the area of the lake, […] are important factors affecting the outburst probability of a glacial lake (McKillop and Clague, 2007; Bolch et al., 2011)." and that "large expanding lakes are more susceptible to outburst, as the greater surface area is exposed to potential mass movement impact". Iribarren Anacona et al. (2014) reported that "lake dimensions have been directly related to outburst volume, peak discharge and flood damage potential (Costa and Schuster, 1988). Accordingly, larger lakes are considered to be more hazardous than small lakes. Furthermore, lakes with larger areas are generally deeper (see e.g. the database by Diaz et al., 2007), and may exert higher hydrostatic pressures over the dams making them more susceptible to failure (Richardson and Reynolds, 2000b). Larger lakes also have a greater surface area potentially exposed to mass movement and ice avalanche impacts, increasing their outburst susceptibility." Finally, Mergili and Schneider (2011) directly used "lake area development" as one of their four key parameters in their qualitative rating for the susceptibility to lake outburst by internal forces (dam failure).

We agree with the referee that the citation of Bolch et al. (2011) within this context is less suitable in L122 and therefore removed it. We now rewrote this text passage as follows (LL122-132):

"Since 1990, lake areas have grown largest in the Central Himalayas (+23%), and lowest in the northwestern Himalayas (+5.0%) (Nie et al., 2017), and many studies have emphasised the role of growing lakes on GLOF susceptibility (e.g. GAPHAZ, 2017; Prakash and Nagarajan, 2017; Rounce et al., 2016). Many previous GLOF assessment schemes included lake area or lake area growth as a proxy for the volume of water that could be potentially released by an outburst and, thus, the resulting downstream hazard (e.g. Allen et al., 2019; Bolch et al., 2011). However, a number of studies also stress that lake area and its growth define the exposure to external and internal triggers of moraine dam breach: larger and growing lakes offer more area for impacts from mass flows such as avalanches, rockfalls, and landslides originating from adjacent valley slopes (GAPHAZ, 2017; Haeberli et al., 2017; Prakash and Nagarajan, 2017; Rounce et al., 2016). Some authors also link growing lake areas to an increase in hydrostatic pressure acting on its moraine dam, thus, making the letter more susceptible to sudden failure (Iribarren Anacona et al., 2014; Mergili and Schneider, 2011)."

*Minor comment #2:*

> *The second aspect where I am not yet 100% convinced is the point on using catchment area instead of lake area. The authors provide convincing arguments in their response letter and include this also in the revised manuscript (catchment area representing the potential of intense rainfall runoff and snow melt, as well as being invariant at the investigated time scales). But then I wonder why catchment area has not been used in all the models instead of lake area. Or why is this aspect more important for the glacier-mass balance model and the monsoonality model, an less for the elevation dependent and the forecasting model? In my view, a few more words on that would be needed, probably in the paragraph ending on L130.*

From a physical process point of view, we argue that catchment area is more suitable in our glacier-mass balance and monsoonality models than glacial lake area. We anticipate that larger catchments can collect more run-off from glacier melt or heavy monsoon precipitation. The choice of this predictor is supported by Allen et al. (2019) who noted that the "total watershed area upstream of the lake [*recognises*] the potential for runoff from heavy rainfall or glacier/snow melt to drain into and overwhelm a glacial lake [4,41]." We now address this issue in our revised manuscript (LL135-138): "We find that catchment area $C$ correlates with lake area A (Pearson's $\rho = 0.45$) and we, thus, preferred $C$ over $A$ in two of our models, as $C$ is invariant at the timescale of our study and we use these two models to explicitly test whether runoff by glacier melt or monsoonal precipitation had an effect on GLOFs in our study area." We reiterate our point that the strong correlation between the static predictors glacial lake area and catchment area in our database makes them, from a data-driven point of view, interchangeable in our models. One motivation for our study was to test and present alternative model variants, and the replacement of glacial lake area by catchment area may be useful in areas with limited data on either predictor.

*Minor comment #3:*

> *Then also the reasoning of selecting "lake-area change" as a predictor (Table 2) as well as the response to my detailed comment #4 (lake depth, not area is determining the hydrostatic pressure on the dam) do not convince me yet. Most glacial lakes, once they have a certain size, mainly increase their area (and thus volume) but not their depth while the grow further. It is true that there are depth and volume estimation rules based on lake area, but the uncertainties of these approaches are*

*huge, and they are based on empirical relationships (of static conditions) and do not involve any dynamic considerations, not any physical processes. If I think of any larger moraine dammed lake in the Himalayas that is growing due to glacier retreat at its upstream end (and thus increasing its area and volume), I would not expect a significant increase of lake depth. L119-121 and the reasoning in Table 2 should be reconsidered in my view.*

We find it difficult to comment on the reviewer's remark that a glacial lake requires a "certain size" to chiefly increase its area instead of depth. We are unsure whether this is a hypothesis or a result from a published study. The reviewer may concede, though, that reliable field data on lake-depth changes are few, hence our choice of a data-driven probabilistic model instead of a physical one. We do acknowledge that empirical relationships between lake area, depth, and volume have high noise (depending on model choice). Yet some of this noise arises from the different stages of lake development (or age) that the data are based on. Hence the dynamics of lake growth are implicit in these empirical relationships, contrary to what the reviewer suggests. In this context, we recall the purpose of a statistical predictor: we use changes in lake area because we lack detailed measurements of changes in lake depth at the scale of our study. Our objective is to offer probabilistic estimates of a GLOF history based on these predictors instead of deriving a physical-based model that ties lake-depth changes to past outbursts.

We point out that a number of studies have stressed that the relationship between lake area, lake volume, and lake depth might be assumed for a majority of glacial lakes and have subsequently used this approach (Huggel et al., 2002; Iribarren Anacona et al., 2014; Mergili et al., 2011; Prakash and Nagarajan, 2017). These studies are – like our own assessment – predominantly remote-sensing-based and are, due to the lack of bathymetric data on a regional scale, limited to the more readily available metric of lake area. To our knowledge, (measured) lake depth has not been applied in any GLOF assessment of comparable scale in the greater Himalayan region. Our objective is to test whether commonly applied predictors of GLOF susceptibility are credibly informative in this regard, and glacial lake area is indeed so. In other words, we can learn more from the data by including this metric. In reflection to this comment, we rephrased this in LL119-122 as followed: "Due to a general lack in available bathymetric data on a regional scale, a number of studies used the frequently observed phenomenon that lake area scales with lake volume and depth (Huggel et al., 2002; Iribarren Anacona et al., 2014). Growing lake depths increase the hydrostatic pressure acting on moraine dams, thus raising the potential of failure (Iribarren Anacona et al., 2014; Rounce et al., 2016)." We also now list the bullet point "increasing lake area commonly reported as scaling with increasing lake depth" in our Table 2.

*Minor comment #4:*

> *Finally two more details on Table 1:*
> *In the new Table 1, the predictors "glacial lake area" "lake-area change" should be moved from "lake characteristics and dynamics" to the "potential triggering mechanisms" group in my view. As the authors mention in the text and the response, lake area (increase) is a proxy for (increased) probability of the lake being impacted by an upstream mass movement.*

In our reply to Minor Comment #1 and in our revised manuscript, we argue that glacial lake area and its change may affect the susceptibility both to external and internal GLOF triggering mechanisms. In Table 1, the group "potential triggering mechanisms (geomorphic)"

predominantly focusses on external triggering mechanisms, so that we prefer to keep these in their dedicated "lake characteristics and dynamics" group.

*Minor comment #5:*

> *Further, I think it is very unfortunate no reference is given for the "summer precipitation" predictor (last line in Table 1). Not that I could make a good suggestion, but having no reference here contradicts the study description (L72-73) "[…]we tested how well some of the more widely adopted predictors of GLOF susceptibility and hazard fare in a multi-level logistic regression […]". Either some reference(s) should be added to this last line in Tab. 1, or a statement on the summer precipitation predictor should be made around this statement in the introduction.*

Liu et al. (2014) studied the relationship between air temperature and GLOFs on the Tibetan Plateau and emphasize that warm and moist conditions during the Asian summer monsoon may have played a role in historic GLOFs in the Himalayas. Wang et al., (2012) considered the "climatic predisposition" of glacial lakes as categorical value in their assessment of moraine-dam breach probabilities in the Chinese Himalayas. Using this variable, they defined a category "warm-wet", derived from high annual precipitation and high daily average temperatures during the summer months at a given lake location. In their assessment of breach probabilities for the Longbasaba and Pida glacial lakes, Wang et al. (2008) also assessed the climatic setting of lakes. They defined the "combination of high temperature–wetness" as one of the "value(s) of most likely breaching" in their assessment. In both of these studies these "warm-wet" and "high temperature-wetness" values were used as indicators of the effects of summer monsoon. However, both studies used meteorological station data instead of CHELSA model data and we are – to our knowledge – the first to apply these data in a regional GLOF assessment for the HKKHN. As a result, the estimated summer precipitation or the precipitation of the warmest quarter (variable Bio18; Karger et al., 2017), has not yet been explored in other GLOF studies. We use this variable to estimate the annual proportion of summer precipitation and thus as a proxy of monsoonality. To acknowledge the approach by Wang et al. (2008, 2012), we changed the description of this predictor to "Summer precipitation or proxy of monsoonality".

Cited references:

Allen, S. K., Zhang, G., Wang, W., Yao, T. and Bolch, T.: Potentially dangerous glacial lakes across the Tibetan Plateau revealed using a large-scale automated assessment approach, Sci. Bull., (April), doi:10.1016/j.scib.2019.03.011, 2019.

Bolch, T., Peters, J., Yegorov, A., Pradhan, B., Buchroithner, M. and Blagoveshchensky, V.: Identification of potentially dangerous glacial lakes in the northern Tien Shan, Nat. Hazards, 59(3), 1691–1714, doi:10.1007/s11069-011-9860-2, 2011.

GAPHAZ: Assessment of Glacier and Permafrost Hazards in Mountain Regions: technical Guidance Document. Standing Group on Glacier and Permafrost Hazards in Mountains (GAPHAZ) of the International Association of Cryospheric Sciences (IACS) and the International Per, Zurich, Lima., 2017.

Haeberli, W., Schaub, Y. and Huggel, C.: Increasing risks related to landslides from degrading permafrost into new lakes in de-glaciating mountain ranges, Geomorphology, 293,

405–417, doi:https://doi.org/10.1016/j.geomorph.2016.02.009, 2017.

Huggel, C., Kääb, A., Haeberli, W., Teysseire, P. and Paul, F.: Remote sensing based assessment of hazards from glacier lake outbursts: a case study in the Swiss Alps, Can. Geotech. J., 39(2), 316–330, doi:10.1139/t01-099, 2002.

Iribarren Anacona, P., Norton, K. P. and Mackintosh, A.: Moraine-dammed lake failures in Patagonia and assessment of outburst susceptibility in the Baker Basin, Nat. Hazards Earth Syst. Sci., 14(12), 3243–3259, doi:10.5194/nhess-14-3243-2014, 2014.

Karger, D. N., Conrad, O., Böhner, J., Kawohl, T., Kreft, H., Soria-Auza, R. W., Zimmermann, N. E., Linder, H. P. and Kessler, M.: Climatologies at high resolution for the earth's land surface areas, Sci. Data, 4, 1–20, doi:10.1038/sdata.2017.122, 2017.

Liu, J. J., Cheng, Z. L. and Su, P. C.: The relationship between air temperature fluctuation and Glacial Lake Outburst Floods in Tibet, China, Quat. Int., 321, 78–87, doi:10.1016/j.quaint.2013.11.023, 2014.

Mergili, M. and Schneider, J. F.: Regional-scale analysis of lake outburst hazards in the southwestern Pamir, Tajikistan, based on remote sensing and GIS, Nat. Hazards Earth Syst. Sci., 11(5), 1447–1462, doi:10.5194/nhess-11-1447-2011, 2011.

Mergili, M., Schneider, D., Worni, R. and Schneider, J. F.: Glacial lake outburst floods in the Pamir of Tajikistan: Challenges in prediction and modelling, Int. Conf. Debris-Flow Hazards Mitig. Mech. Predict. Assessment, Proc., 973–982, doi:10.4408/IJEGE.2011-03.B-106, 2011.

Nie, Y., Sheng, Y., Liu, Q., Liu, L., Liu, S., Zhang, Y. and Song, C.: A regional-scale assessment of Himalayan glacial lake changes using satellite observations from 1990 to 2015, Remote Sens. Environ., 189, 1–13, doi:10.1016/j.rse.2016.11.008, 2017.

Prakash, C. and Nagarajan, R.: Outburst susceptibility assessment of moraine-dammed lakes in Western Himalaya using an analytic hierarchy process, Earth Surf. Process. Landforms, 42(14), 2306–2321, doi:10.1002/esp.4185, 2017.

Rounce, D. R., McKinney, D. C., Lala, J. M., Byers, A. C. and Watson, C. S.: A new remote hazard and risk assessment framework for glacial lakes in the Nepal Himalaya, Hydrol. Earth Syst. Sci., 20(9), 3455–3475, doi:10.5194/hess-20-3455-2016, 2016.

Wang, X., Liu, S., Guo, W. and Xu, J.: Assessment and simulation of glacier lake outburst floods for Longbasaba and Pida lakes, China, Mt. Res. Dev., 28(3–4), 310–317, doi:10.1659/mrd.0894, 2008.

Wang, X., Liu, S., Ding, Y., Guo, W., Jiang, Z., Lin, J. and Han, Y.: An approach for estimating the breach probabilities of moraine-dammed lakes in the Chinese Himalayas using remote-sensing data, Nat. Hazards Earth Syst. Sci., 12(10), 3109–3122, doi:10.5194/nhess-12-3109-2012, 2012.

---

## Author Response (AR3)

**Reply to Editor Comments**

EC #1: L. 36/37: Provide a reference for the statement.

> We added Richardson and Reynolds (2000) and Rounce et al. (2016) as references to our statement.

EC #2: L 48ff: You may also cite King et al. (2019) here as they specifically show that lake-terminating glaciers have higher mass loss in comparison to land terminating ones which has important implications for this study.

> We thank the editor for this suggestion and added the following statement in LL50f: "Parts of this meltwater have been trapped in glacial lakes that have expanded by approximately 14% between 1990 and 2015 (Nie et al., 2017). King et al. (2019) found that Himalayan glaciers terminating in lakes had higher rates of mass loss since the 1970s than those not in direct contact to a glacial lake."

EC #3: L. 89: Please add the correct citation for the RGI 6.0 (RGI Consortium, 2017. Randolph Glacier Inventory (RGI) – A dataset of global glacier outlines: version 6.0, Technical Report., Boulder, Colorado, USA.)

> Thank you, we changed the citation accordingly.

EC #4: L. 90f and Fig. 1, Fig. 10: Please provide more details about the subdivision. It would be good to have consistency amongst the publications to allow comparisons. Did you use existing accepted subdivisions (e.g. as defined by Bolch et al. (2019) for the Hindukush-Himalayan Assessment report) or defined you own? The Nyainqentanglha region does not include the Eastern Nyainqentanglha. Adjust the name.

> We use the subdivision established by Veh et al. (2020). To provide more detail on our study area subdivision, we rephrased the text passage L89ff as followed: "Following the outlines of glacier regions in High Mountain Asia used in the Randolph Glacier Inventory version 6.0 (RGI Consortium, 2017) and those defined by Brun et al. (2017), Veh et al. (2020) subdivided our study area into seven mountain ranges: the Hindu Kush, the Karakoram, the Western Himalaya, the Central Himalaya, the Eastern Himalaya, the Nyainqentanglha, and the Hengduan Shan."

EC #5: L. 122f: There are more publications addressing the changes of the glacial lake in the Himalaya and beyond, e.g. Wang et al. (2020) or Chen et al. (2021). Do the they come to similar findings than the study by Nie et al. (2017)?

> In essence, most multi-temporal lake inventories show enhanced growth in the Southern part of the Himalayas (Central Himalayas, Eastern Himalayas, and Nyainqentanglha). For an instructive comparison of previous work, we wish to refer the Editor to Supplementary Table 1 in Wang et al. (2020), listing 34 lake inventories in our study region. They note that "it is difficult to evaluate any discrepancies comprehensively because different extents of glacial lake distribution were examined and inconsistent thresholds of minimum lake area were used." We now rephrased the sentence starting in L123 to reflect the recent findings of Chen et al.

(2021) but wish to also maintain the findings of Nie et al. (2017), though we are aware of potential discrepancies with other studies largely because of a different methodological setup. "In the past decades, lake areas have grown largest in the Central Himalayas (+23% in 1990-2015; Nie et al., 2017) and Nyainqentanglha Mountains but lowest in the northwestern Himalayas (Chen et al., 2021; Nie et al., 2017), and many studies have emphasised the role of growing lakes on GLOF susceptibility (e.g. GAPHAZ, 2017; Prakash and Nagarajan, 2017; Rounce et al., 2016)."

EC #6: L163: Provide the information about how you extracted the moraine-dammed lakes.

We rephrased this text passage LL162ff for better clarification: "First, we used the ICIMOD database of 25,614 lakes manually mapped from Landsat imagery acquired in 2005 ± two years (Maharjan et al., 2018), from which we extracted 7,284 lakes dammed by moraines (classes m(l), m(e), and m(o) in Maharjan et al., 2018)."

EC #7: L. 410: You may want to consider that lake-terminating lakes have more negative mass balances and higher retreat rates than land terminating ones (King et al. 2019). Hence, the lakes are also growing faster.

We thank the author for this suggestion and included the finding from King et al. (2019) in the paragraph starting from L409. "Most moraine-dammed lakes in the HKKHN, however, are fed by glaciers with negative mass balances that likely help to elevate GLOF potential through increased meltwater input and glacier-tongue calving rates (Emmer, 2017; Richardson and Reynolds, 2000). This is also supported by the findings of King et al. (2019), which imply that higher rates of mass loss of lake-terminating glaciers since the 1970s might have also led to increased meltwater input into lakes adjacent to their termini. More than 70% of all lakes that burst out in the past four decades were in contact to their parent glaciers (Veh et al., 2019)."

References cited in this reply:

Brun, F., Berthier, E., Wagnon, P., Kääb, A. and Treichler, D.: A spatially resolved estimate of High Mountain Asia glacier mass balances from 2000 to 2016, Nat. Geosci., 10(9), 668–673, doi:10.1038/ngeo2999, 2017.

Chen, F., Zhang, M., Guo, H., Allen, S., Kargel, J. S., Haritashya, U. K. and Scott Watson, C.: Annual 30 m dataset for glacial lakes in high mountain asia from 2008 to 2017, Earth Syst. Sci. Data, 13(2), 741–766, doi:10.5194/essd-13-741-2021, 2021.

Emmer, A.: Glacier Retreat and Glacial Lake Outburst Floods (GLOFs), in Oxford Research Encyclopedia of Natural Hazard Science, pp. 1–37, Oxford University Press USA., 2017.

GAPHAZ: Assessment of Glacier and Permafrost Hazards in Mountain Regions: technical Guidance Document. Standing Group on Glacier and Permafrost Hazards in Mountains (GAPHAZ) of the International Association of Cryospheric Sciences (IACS) and the International Per, Zurich, Lima., 2017.

King, O., Bhattacharya, A., Bhambri, R. and Bolch, T.: Glacial lakes exacerbate Himalayan glacier mass loss, Sci. Rep., 9(1), 1–9, doi:10.1038/s41598-019-53733-x, 2019.

Maharjan, S. B., Mool, P. K., Lizong, W., Xiao, G., Shrestha, F., Shrestha, R. B., Khanal, N. R., Bajracharya, S. R., Joshi, S., Shai, S. and Baral, P.: The Status of Glacial Lakes in the Hindu Kush Himalaya, International Centre for Integrated Mountain Development (ICIMOD), Kathmandu., 2018.

Nie, Y., Sheng, Y., Liu, Q., Liu, L., Liu, S., Zhang, Y. and Song, C.: A regional-scale assessment of Himalayan glacial lake changes using satellite observations from 1990 to 2015, Remote Sens. Environ., 189, 1–13, doi:10.1016/j.rse.2016.11.008, 2017.

Prakash, C. and Nagarajan, R.: Outburst susceptibility assessment of moraine-dammed lakes in Western Himalaya using an analytic hierarchy process, Earth Surf. Process. Landforms, 42(14), 2306–2321, doi:10.1002/esp.4185, 2017.

RGI Consortium: Randolph Glacier Inventory – A Dataset of Global Glacier Outlines: Version 6.0: Technical Report, , doi:https://doi.org/10.7265/N5-RGI-60, 2017.

Richardson, S. D. and Reynolds, J. M.: An overview of glacial hazards in the Himalayas, in Quaterny International, vol. 65–66, pp. 31–47., 2000.

Rounce, D. R., McKinney, D. C., Lala, J. M., Byers, A. C. and Watson, C. S.: A new remote hazard and risk assessment framework for glacial lakes in the Nepal Himalaya, Hydrol. Earth Syst. Sci., 20(9), 3455–3475, doi:10.5194/hess-20-3455-2016, 2016.

Veh, G., Korup, O., Specht, S., Roessner, S. and Walz, A.: Unchanged frequency of moraine-dammed glacial lake outburst floods in the Himalaya, Nat. Clim. Chang., 2000, 1–5, doi:10.1038/s41558-019-0437-5, 2019.

Veh, G., Korup, O. and Walz, A.: Hazard from Himalayan glacier lake outburst floods, Proc. Natl. Acad. Sci. U. S. A., 117(2), 907–912, doi:10.1073/pnas.1914898117, 2020.

Wang, X., Guo, X., Yang, C., Liu, Q., Wei, J., Zhang, Y., Liu, S., Zhang, Y., Jiang, Z. and Tang, Z.: Glacial lake inventory of high-mountain Asia in 1990 and 2018 derived from Landsat images, Earth Syst. Sci. Data, 12, 2169–2182, doi:10.5194/essd-12-2169-2020, 2020.